# Sample-Conditioned Hypothesis Stability Sharpens Information-Theoretic Generalization Bounds

**Ziqiao Wang**
University of Ottawa
zwang286@uottawa.ca

**Yongyi Mao**
University of Ottawa
ymao@uottawa.ca

## Abstract

We present new information-theoretic generalization guarantees through the a novel construction of the "neighboring-hypothesis" matrix and a new family of stability notions termed sample-conditioned hypothesis (SCH) stability. Our approach yields sharper bounds that improve upon previous information-theoretic bounds in various learning scenarios. Notably, these bounds address the limitations of existing information-theoretic bounds in the context of stochastic convex optimization (SCO) problems, as explored in the recent work by Haghifam et al. (2023).

## 1 Introduction

Information-theoretic upper bounds for generalization error have recently been developed since the seminal works of [52, 67]. On the one hand, these bounds have attracted increasing interest due to their distribution dependence and algorithm dependence, making them highly suitable for reasoning the generalization behavior of modern deep learning. In fact, subsequent studies have demonstrated that information-theoretic bounds can effectively track the dynamics of generalization error in deep neural networks [40, 60, 23, 61, 62, 25, 64]. On the other hand, recent studies [58, 19, 23, 20, 25] have revealed the expressive nature of information-theoretic bounds in the distribution-free setting. Notably, the conditional mutual information (CMI) framework proposed by [58] has shown great promise by establishing connections to VC theory [59] and matching minimax rates for the binary classification [19, 20]. Additionally, in the case of the $0 - 1$ loss and the realizable setting, where an interpolating algorithm attains zero empirical risk, information-theoretic bounds give an exact characterization of the generalization error [20, 64], thereby providing the tightest possible generalization bound.

Nonetheless, information-theoretic bounds have been extensively discussed due to their two main deficiencies. The first deficiency concerns the unbounded nature of the original input-output mutual information (IOMI) bounds [67, 4, 32, 31]. To address this issue, several techniques have been developed, including the chaining method [2], the individual technique [9] or random subset technique [40], and the Gaussian noise perturbation technique [41]. Notably, the CMI bound [58] stands out as it has a finite upper bound for any learning scenario due to its supersample construction. These techniques are now often applied jointly for analyzing generalization [18, 49, 70, 63, 23, 25, 64]. The second deficiency concerns the sub-optimal convergence rate of information-theoretic bounds. Specifically, when the information-theoretic quantities, whether IOMI or CMI, are bounded by constants, the bounds exhibit a decaying rate in the order of $\mathcal{O}(1/\sqrt{n})$, where $n$ is the sample size. In contrast, generalization errors in practice may decay at a faster rate, e.g., $\mathcal{O}(1/n)$. To address this limitation, several works, inspired by some PAC-Bayesian literature, have proposed fast-rate information-theoretic bounds [24, 25, 64], demonstrating a good characterization in some instances of non-convex settings such as deep learning. More recently, [65, 66, 71] present IOMI bounds for the Gaussian mean estimation problem, that achieve optimal convergence rates, contrasting previous bounds [9, 70].

The issue of slow convergence in information-theoretic bounds has recently been amplified by the observation that these bounds may not even vanish. [21] highlight this limitation in the context of stochastic convex optimization (SCO) problems [55]. Specifically, [21] shows that all existing information-theoretic bounds, including the CMI bound [58], the Gaussian noise perturbed IOMI bound [41], the individual IOMI bound [9], the evaluated CMI (e-CMI) bound [58] and the functional CMI ($f$-CMI) bound [23], fail to vanish in at least one of the counterexamples they constructed. These failures stem from the dimension-dependent nature of current information-theoretic quantities [31], appearing an intrinsic barrier for overcoming these limitations. However, if we dissect the process by which these bounds are derived, opportunities do exist. Specifically, recall that all these information-theoretic bounds are built upon the Donsker-Varadhan (DV) variational representation of the KL divergence [44, Theorem 3.5] (see Lemma A.2 in the Appendix). Using this representation, a generalization upper bound is derived in terms of information-theoretic quantity and a cumulant-generating function (CGF), as illustrated below.

$$\text{Generalization Error} \leq \inf_{\text{Para.}>0} \frac{\text{IOMI or CMI} + \text{CGF}}{\text{Para.}}.$$

Particularly note that the CGF depends on certain choice of the auxiliary function in the DV formula. Then, except for the IOMI or CMI itself, the tightness of the generalization bound hinges on two key factors: the selection of the DV auxiliary function and the approach used to bound the CGF, the latter of which often requires additional assumptions specific to the chosen DV auxiliary function. The most common choices for the DV auxiliary function involve making assumptions such as sub-Gaussian loss or bounded loss. For instance, in the case of IOMI bounds, the DV auxiliary function is typically chosen as the single loss or average loss, and the sub-Gaussian assumption (or boundedness assumption) is utilized. In CMI bounds, the DV auxiliary function is defined as the difference in loss between a training sample and a "ghost" sample, and the boundedness property is employed. We now note that exploiting bounded loss is the fundamental reason behind the failures in SCO problems. Although [21] does not explicitly rely on boundedness, they do make use of the product of the Lipschitz constant and the diameter of the hypothesis domain, which essentially serves as an upper bound for the loss. The product does not vanish with $n$, thereby neutralizing the potential to include another decaying factor in the bound. Arguably, at least for these SCO problems, the selection of the DV auxiliary function may not be optimal, for example, the resulting CGF$= \mathcal{O}(1/n)$, giving vacuous generalization guarantees. This analysis inspires us to explore alternative DV auxiliary function and to adopt different assumptions for bounding the CGF. For instance, we may devise appropriate stability assumptions and create opportunities to upper bound the CGF using a term in $\mathcal{O}(\beta^2/n)$, where $\beta$ is the stability parameter that decays as $n$. This is promising in light of the capability of the stability-based framework in explaining the generalization of SCO problems [6, 56, 22, 5].

In this paper, we combine information-theoretic analysis with stability notions to develop IOMI bounds and CMI bounds that improve upon previous information-theoretic bounds for stable learning algorithms, achieving faster convergence rates. The main contributions of our paper are summarized below.

- We introduce our new notions of algorithmic stability, referred to as sample-conditioned hypothesis (SCH) stability. We also present a novel construction of a sample-dependent hypothesis matrix, where each column is a neighborhood pair of hypotheses obtained from two training samples that differ in only one element, inspired by the supersample setting of CMI [58].

- We present new IOMI bounds, which explicitly include the SCH stability parameters and are shown superior to previous bounds for stable learning algorithms.

- We show that the sample-dependent hypothesis matrix, similar to the supersample matrix, enjoys a symmetry property. Exploiting this symmetry, we establish novel CMI bounds. Specifically, we present hypotheses-conditioned CMI bounds that are analogous to the previous supersample-conditioned CMI bounds. Additionally, we derive sample-conditioned CMI bounds exploiting other assumptions. Notably, these bounds introduce novel CMI quantities and include SCH stability parameters. In particular, the new CMI quantities remain boundedness as the original CMI. Consequently, these CMI bounds vanish no slower than their stability parameters. In addition, we also obtain a second-moment generalization bound that matches the tightest known bound in the literature [14] under the same condition.

- We apply our new bounds to a convex-Lipschitz-bounded (CLB) example in which previous information-theoretic bounds fail to explain generalization [21], and show that the new IOMI

and CMI bounds vanish, benefiting from SCH stability. Additionally, we discuss another CLB example where the uniform stability parameter is non-vanishing or has a slow convergence rate, but our information-theoretic bounds remain tight up to a constant.

- We extend our analysis to derive information-theoretic generalization bounds based on Bernstein condition, which leverages a connection between stability and Bernstein condition. We also show that stability can be incorporated into generalization bounds to obtain stronger results using alternative information-theoretic quantities such as the loss difference based CMI, e-CMI and $f$-CMI. Furthermore, we illustrate the expressiveness of our new CMI notions under the distribution-free setting.

## 2 Preliminaries

**Probability and Information Theory Notation**   Unless otherwise noted, a random variable will be denoted by a capitalized letter, and its realization by the corresponding lower-case letter. The distribution of a random variable $X$ is denoted by $P_X$, and the conditional distribution of $X$ given $Y$ is denoted by $P_{X|Y}$. When conditioning on a specific realization $y$, we use the shorthand $P_{X|Y=y}$ or simply $P_{X|y}$. Denote by $\mathbb{E}_X$ expectation over $X \sim P_X$, and by $\mathbb{E}_{X|Y=y}$ (or $\mathbb{E}_{X|y}$) expectation over $X \sim P_{X|Y=y}$. The entropy of a random variable $X$ is denoted by $H(X)$, and the KL divergence of probability distribution $P$ with respect to $Q$ is denoted by $\mathrm{D}_{\mathrm{KL}}(P||Q)$. The mutual information (MI) between random variables $X$ and $Y$ is denoted by $I(X;Y)$, and the conditional mutual information between $X$ and $Y$ given $Z$ is denoted by $I(X;Y|Z)$. We also define the disintegrated mutual information as $I^z(X;Y) \triangleq \mathrm{D}_{\mathrm{KL}}(P_{X,Y|Z=z}||P_{X|Z=z}P_{Y|Z=z})$, following the notation in [40]. Note that $I(X;Y|Z) = \mathbb{E}_Z[I^Z(X;Y)]$.

**Generalization Error and Uniform Stability**   We consider the supervised learning setting, where we have a domain of instances $\mathcal{Z} = \mathcal{X} \times \mathcal{Y}$, with input and label spaces denoted by $\mathcal{X}$ and $\mathcal{Y}$ respectively. The distribution of an instance is given by $\mu$, and we have a training sample $S = \{Z_i\}_{i=1}^n \sim \mu^n$. Let $R \in \mathcal{R}$ be a source of randomness (a random variable independent of $S$ over an appropriate space $\mathcal{R}$), from which a learning algorithm $\mathcal{A} : \mathcal{Z}^n \times \mathcal{R} \to \mathcal{W}$ takes the training sample $S$ and $R$ as input, and outputs a hypothesis $W = \mathcal{A}(S, R) \in \mathcal{W}$. To evaluate the quality of the output hypothesis $W$, we use a loss function $\ell : \mathcal{W} \times \mathcal{Z} \to \mathbb{R}_0^+$. Given a fixed $w$, we define the population risk $L_\mu(w) \triangleq \mathbb{E}_{Z'}[\ell(w, Z')]$, where $Z' \sim \mu$ is a testing instance. The quantity $L_\mu = \mathbb{E}_W[L_\mu(W)]$ is then the expected population risk. For a fixed $w \in \mathcal{W}$, the empirical risk on $S$ is defined as $L_S(w) \triangleq \frac{1}{n}\sum_{i=1}^n \ell(w, Z_i)$. Similarly, we define the expected empirical risk as $\hat{L}_n = \mathbb{E}_{W,S}[L_S(W)]$. Thus, the expected generalization error is given by $\mathcal{E}_\mu(\mathcal{A}) \triangleq L_\mu - \hat{L}_n$.

We now give two notions of uniform stability [6, 13], where $s \simeq s^i$ denotes two training sets $s$ and $s^i$ that differ only at the $i$th element. We say a learning algorithm $\mathcal{A}$ is $\beta_1$-weakly uniformly stable if

$$\sup_{s \simeq s^i, z} \mathbb{E}_R \left| \ell(\mathcal{A}(s, R), z) - \ell(\mathcal{A}(s^i, R), z) \right| \leq \beta_1,$$

and $\beta_2$-strongly uniformly stable if

$$\sup_{s \simeq s^i, z} \sup_r \left| \ell(\mathcal{A}(s, r), z) - \ell(\mathcal{A}(s^i, r), z) \right| \leq \beta_2.$$

**Remark 2.1.** *Notably the weak uniform stability above is the standard in the literature. It is evident that if $\mathcal{A}$ is $\beta$-strongly uniformly stable, it must also be $\beta$-weakly uniformly stable. It is also worth noting that for deterministic algorithms (e.g., GD or fixed permutation SGD), the two notions are identical. We note that in this paper, when speaking of uniform stability, we refer to the strong notion.*

Let $S' = \{Z'_i\}_{i=1}^n \sim \mu^n$ be an independent copy of $S$, and let $S^{\setminus i} = S \setminus \{Z_i\}$ so $S^i = S^{\setminus i} \cup \{Z'_i\}$.

The following well-known result (e.g., [6, Lemma 7], [54, Thm. 13.2]) is frequently used in this paper.

**Lemma 2.1.** *For any algorithm $\mathcal{A}$, we have $L_\mu = \mathbb{E}_{S,S',R} \left[ \frac{1}{n}\sum_{i=1}^n \ell(\mathcal{A}(S^i, R), Z_i) \right]$, and*

$$\mathcal{E}_\mu(\mathcal{A}) = \mathbb{E}_{S,S'} \left[ \frac{1}{n}\sum_{i=1}^n \left[ \mathbb{E}_{\mathcal{A}(S^i,R)|S,Z'_i} \ell(\mathcal{A}(S^i, R), Z_i) - \mathbb{E}_{\mathcal{A}(S,R)|S} \ell(\mathcal{A}(S, R), Z_i) \right] \right]. \quad (1)$$

**Supersample and Sample-Conditioned Hypothesis Stability**  Following [58], let $\widetilde{Z} \in \mathcal{Z}^{n \times 2}$ be a supersample matrix with $n$ rows and 2 columns, each entry drawn independently from $\mu$. We index the columns of $\widetilde{Z}$ by $0, 1$ and denote the $i$th row of $\widetilde{Z}$ by $\widetilde{Z}_i$, with entries $(\widetilde{Z}_{i,0}, \widetilde{Z}_{i,1})$. We often use the superscripts $+$ and $-$ to respectively replace the subscripts $0$ and $1$, i.e., writing $\widetilde{Z}_i^+$ for $\widetilde{Z}_{i,0}$ and $\widetilde{Z}_i^-$ for $\widetilde{Z}_{i,1}$. Correspondingly, $\widetilde{Z}_{[n]}^+$ and $\widetilde{Z}_{[n]}^-$ denote the first and second columns, respectively. Additionally, we will use $\widetilde{Z}_{[n]\sim i}^+$ to denote $\widetilde{Z}_{[n]}^+$ in which the $i$th element $\widetilde{Z}_i^+$ is replaced with the corresponding element $\widetilde{Z}_i^-$ in the second column. Let $\widetilde{W}^+ = \mathcal{A}(\widetilde{Z}_{[n]}^+, R)$, and $\widetilde{W}_i^- = \mathcal{A}(\widetilde{Z}_{[n]\sim i}^+, R)$ for each $i \in [n]$. That is, $\widetilde{W}^+$ and each $\widetilde{W}_i^-$ are obtained by two "neighboring" training samples, namely those differ only on one instance (the $i$th instance). We then construct matrix $\widetilde{W} \in \mathcal{W}^{n \times 2}$, where the $i$th row $\widetilde{W}_i = (\widetilde{W}_i^+, \widetilde{W}_i^-) = (\widetilde{W}_{i,0}, \widetilde{W}_{i,1})$ and $\widetilde{W}_i^+ = \widetilde{W}^+$ for all $i \in [n]$, as shown in Table 1. Unlike the supersample matrix, elements in $\widetilde{W}$ are identically distributed but not independent. In this case, Eq. (1) can be rewritten as

| $\widetilde{Z}_1^+$ | $\widetilde{Z}_1^-$ |
|---|---|
| $\widetilde{Z}_2^+$ | $\widetilde{Z}_2^-$ |
| $\vdots$ | $\vdots$ |
| $\widetilde{Z}_n^+$ | $\widetilde{Z}_n^-$ |

$\overset{\mathcal{A}}{\Longrightarrow}$

| $\widetilde{W}^+$ | $\widetilde{W}_1^-$ |
|---|---|
| $\widetilde{W}^+$ | $\widetilde{W}_2^-$ |
| $\vdots$ | $\vdots$ |
| $\widetilde{W}^+$ | $\widetilde{W}_n^-$ |

Table 1: Supersample (Left) and the induced hypotheses (Right).

$$\mathcal{E}_\mu(\mathcal{A}) = \frac{1}{n} \sum_{i=1}^{n} \mathbb{E}_{\widetilde{Z}_i^+, \widetilde{W}_i} \left[ \ell(\widetilde{W}_i^-, \widetilde{Z}_i^+) - \ell(\widetilde{W}_i^+, \widetilde{Z}_i^+) \right]. \tag{2}$$

The summand in Eq (2) exhibits an interesting "symmetry": for any $i$,

$$\mathbb{E}_{\widetilde{Z}_i^+, \widetilde{W}_i} \left[ \ell(\widetilde{W}_i^-, \widetilde{Z}_i^+) - \ell(\widetilde{W}_i^+, \widetilde{Z}_i^+) \right] = \mathbb{E}_{\widetilde{Z}_i^-, \widetilde{W}_i} \left[ \ell(\widetilde{W}_i^+, \widetilde{Z}_i^-) - \ell(\widetilde{W}_i^-, \widetilde{Z}_i^-) \right]. \tag{3}$$

That is, the $+/-$ superscripts in the summand of Eq (2) can be flipped for any $i$. We may then use $n$ binary ($\{0,1\}$-valued) variables $(U_1, U_2, \ldots, U_n) := U$ to govern whether we choose to flip the signs for each of the $n$ terms, where $U_i = 0$ indicates "no flipping". Let $\overline{U}_i \triangleq 1 - U_i$, we have

$$\begin{aligned} \mathcal{E}_\mu(\mathcal{A}) =& \frac{1}{n} \sum_{i=1}^{n} \mathbb{E}_{\widetilde{Z}_i, \widetilde{W}_i, U_i} \left[ \ell(\widetilde{W}_{i,\overline{U}_i}, \widetilde{Z}_{i,U_i}) - \ell(\widetilde{W}_{i,U_i}, \widetilde{Z}_{i,U_i}) \right] \\ =& \frac{1}{n} \sum_{i=1}^{n} \mathbb{E}_{\widetilde{Z}_i, \widetilde{W}_i, U_i} \left[ (-1)^{U_i} \left( \ell(\widetilde{W}_i^-, \widetilde{Z}_{i,U_i}) - \ell(\widetilde{W}_i^+, \widetilde{Z}_{i,U_i}) \right) \right] \end{aligned} \tag{4}$$

$$=\mathbb{E}_{\widetilde{W}, E, U} \left[ \frac{1}{n} \sum_{i=1}^{n} (-1)^{U_i} \left( \ell(\widetilde{W}_i^-, \widehat{Z}_i) - \ell(\widetilde{W}_i^+, \widehat{Z}_i) \right) \right], \tag{5}$$

where in Eq. (4) we have chosen $U$ to be an i.i.d. Bernoulli-($\frac{1}{2}$) sequence, and in Eq. (5) we have renamed $\widetilde{Z}_{i,U_i}$ as $\widehat{Z}_i$ and denoted $E \triangleq (\widehat{Z}_1, \widehat{Z}_2, \ldots, \widehat{Z}_n)$. Note that $E$, induced by $U$ from $\widetilde{Z}$, contains $n$ instances, each serving to evaluate the loss difference between a pair of hypotheses $(\widetilde{W}_i^-, \widetilde{W}^+)$. The polarities of these evaluations are governed by $U$.

We now define some new notions of stability.

**Definition 2.1** (Sample-Conditioned Hypothesis (SCH) Stability). *Let $S \sim \mu^n$, $W$ and $W^i$ generated via $W = \mathcal{A}(S, R)$ and $W^i = \mathcal{A}(S^i, R)$. Let $\mathcal{Z}_{w,w^i}$ denote the support of the conditional distribution $P_{Z_i|w,w^i}$, and $\mathcal{W}_s$ denote the support of the conditional distribution $P_{W|s}$. We introduce four types of SCH stability, referred to as types A, B, C, D. Specifically, a learning algorithm $\mathcal{A}$ is*

*a)* $\gamma_1$-*SCH-A stable if* $\forall i \in [n]$, $\displaystyle \sup_{w \in \cup_{s \in \mathcal{Z}^n} \mathcal{W}_s} \sup_{z \in \mathcal{Z}} \left| \ell(w, z) - \mathbb{E}_{W^i|w} \left[ \ell(W^i, z) \right] \right| \leq \gamma_1$, (6)

*b)* $\gamma_2$-*SCH-B stable if* $\forall i \in [n]$, $\displaystyle \mathbb{E}_{S,R,Z'} \left[ \left( \ell(W, Z') - \mathbb{E}_{W^i|W} \left[ \ell(W^i, Z') \right] \right)^2 \right] \leq \gamma_2^2$, (7)

*c)* $\gamma_3$-*SCH-C stable if* $\forall i \in [n]$, $\displaystyle \mathbb{E}_{W,W^i} \left[ \sup_{z_i \in \mathcal{Z}_{W,W^i}} \left| \ell(W, z_i) - \ell(W^i, z_i) \right| \right] \leq \gamma_3$, (8)

*d)* $\gamma_4$-*SCH-D stable if* $\forall i \in [n]$, $\displaystyle \mathbb{E}_{S,Z_i',R} \left[ \left( \ell(W, Z_i) - \ell(W^i, Z_i) \right)^2 \right] \leq \gamma_4^2$, (9)

where $Z'$ in Eq. (7) is an independent instance drawn from $\mu$.

**Remark 2.2.** *By definition, we can see $\gamma_2 \leq \gamma_1$, and it is expected that $\gamma_4 \leq \gamma_3$. Note that all of them are smaller than $\beta_2$. In addition, it is expected that $\gamma_2$, $\gamma_3$ and $\gamma_4$ are smaller than $\beta_1$, although the relationship between $\gamma_1$ and $\beta_1$ is uncertain. Moreover, it is also expected that $\gamma_4$ is larger than $\gamma_2$ due to the independence of $Z'$ in Eq. (7). We emphasize that supreme $\sup_w$ in Eq. (6) is taken over the sample-dependent hypothesis space $\cup_{s \in \mathcal{Z}^n} \mathcal{W}_s$, not the whole hypothesis space $\mathcal{W}$. Notably, the notion of "hypothesis set stability" in [16] is closely related to our SCH stability. In fact, $\gamma_1$-SCH-A stable implies $\gamma_1$-hypothesis set stable. Due to space constraints, we provide further elaboration on the definition of SCH stability in Appendix B.*

## 3 IOMI Bounds for Stable Algorithms

We are now in a position to give the IOMI bounds for stable learning algorithms.

**Theorem 3.1.** *If the learning algorithm $\mathcal{A}$ is $\gamma_1$-SCH-A stable, then*

$$|\mathcal{E}_\mu(\mathcal{A})| \leq \frac{\sqrt{2}\gamma_1}{n} \sum_{i=1}^n \sqrt{I(\widetilde{W}^+; \widetilde{Z}_i^+)} \leq \frac{\sqrt{2}\gamma_1}{n} \sum_{i=1}^n \sqrt{I(\widetilde{W}^+; \widetilde{Z}_i^+ | \widetilde{W}_i^-)}.$$

We note that $I(\widetilde{W}^+; \widetilde{Z}_i^+) = I(\widetilde{W}_i^-; \widetilde{Z}_i^-) = I(W; Z_i)$, the only difference between the first bound in Theorem 3.1 and the previous individual IOMI bound in [9] is that the sub-Gaussian variance proxy is replaced by the stability parameter $\gamma_1$ in our bound. In fact, while we use $\widetilde{W}$ to better understand the appearance of $\gamma_1$ in the IOMI bounds, the first bound in Theorem 3.1 itself does not necessarily rely on the supersample and the construction of $\widetilde{W}$.

In addition, if $\mathcal{A}$ is a deterministic algorithm, the term $I(\widetilde{W}^+; \widetilde{Z}_i^+ | \widetilde{W}_i^-)$ in the second bound is tighter than the mutual information stability or erasure information studied in [46, 23], that is, $I(\widetilde{W}^+; \widetilde{Z}_i^+ | \widetilde{W}_i^-) \leq I\left(\widetilde{W}^+; \widetilde{Z}_i^+ | \widetilde{Z}_{[n] \setminus i}^+\right)$ (see Remark C.1 in the Appendix for an explanation).

*Proof Sketch of Theorem 3.1.* Motivated by Lemma 2.1 and Eq. (2), the main innovation in this proof is to let the auxiliary function in DV be the "relative loss" instead of the single loss, namely we let $g(\tilde{w}_i^+, \tilde{z}_i^+) = \mathbb{E}_{\widetilde{W}_i^- | \tilde{w}^+} \left[ \ell(\widetilde{W}_i^-, \tilde{z}_i^+) \right] - \ell(\tilde{w}^+, \tilde{z}_i^+)$ and let the auxiliary function $f = t \cdot g$ for $t > 0$ in Lemma A.2. This enables us to utilize Eq. (6) to bound the CGF. The remaining steps are routine. The complete proof can be found in Appendix C.1. $\square$

The following corollary, immediately following from $\gamma_1 \leq \beta_2$, suggests a clear improvement over the previous IOMI bound for the uniformly stable deterministic algorithm with vanishing $\beta_2$.

**Corollary 3.1.** *If $\mathcal{A}$ is $\beta_2$-uniform stable, then $|\mathcal{E}_\mu(\mathcal{A})| \leq \frac{\sqrt{2}\beta_2}{n} \sum_{i=1}^n \sqrt{I(\widetilde{W}^+; \widetilde{Z}_i^+)}$.*

A key message conveyed in this paper, as shown in the proof of Theorem 3.1, is the importance of carefully selecting the DV auxiliary function and appropriate assumptions for various learning scenarios. Corollary 3.1 also suggests a potential enhancement for uniform stability in the certain deterministic setting, provided that $I(W; Z_i)$ also vanishes appropriately with $n$.

Similar to [40, 49, 23, 25], we also give an $R$-conditioned IOMI bound below.

**Theorem 3.2.** *If $\mathcal{A}$ is $\beta_2$-uniform stable, we have $|\mathcal{E}_\mu(\mathcal{A})| \leq \frac{\sqrt{2}\beta_2}{n} \sum_{i=1}^n \mathbb{E}_R \sqrt{I^R(\widetilde{W}^+; \widetilde{Z}_i^+)}$.*

This disintegrated IOMI bound is not directly comparable to the bounds in Theorem 3.1, but it is equivalent to Corollary 3.1 for deterministic algorithms. With additional assumptions, we may remove the square-root in Theorem 3.1, accelerating the decay of IOMI as shown in Theorem 3.3.

**Theorem 3.3.** *Under the same conditions in Theorem 3.1, if $\mathcal{A}$ is further $\gamma_2$-SCH-B stable, then*

$$|\mathcal{E}_\mu(\mathcal{A})| \leq \frac{\gamma_1}{n} \sum_{i=1}^n I(\widetilde{W}^+; \widetilde{Z}_i^+) + 0.72 \frac{\gamma_2^2}{\gamma_1}.$$

Notice that $\gamma_2^2/\gamma_1 \leq \gamma_1^2/\gamma_1 = \gamma_1$. If $\gamma_2^2/\gamma_1^2$ decays faster than $\frac{1}{n} \sum_{i=1}^n \sqrt{I(\widetilde{W}^+; \widetilde{Z}_i^+)}$, then Theorem 3.3 is qualitatively strictly stronger than Theorem 3.1.

# 4 CMI Bounds for Stable Algorithms

**Hypotheses-Conditioned CMI Bounds** In this section, we give a handful of CMI bounds based on a new information-theoretic quantity.

**Theorem 4.1.** *Suppose that there exists* $\Delta_1 : \mathcal{W}^2 \to \mathbb{R}$ *such that for every* $\tilde{w}_i = (\tilde{w}_i^+, \tilde{w}_i^-)$, $\sup_{z_i \in \mathcal{Z}_{\tilde{w}_i^+, \tilde{w}_i^-}} \left| \ell(\tilde{w}_i^+, z_i) - \ell(\tilde{w}_i^-, z_i) \right| \le \Delta_1(\tilde{w}_i)$. *Then,*

$$|\mathcal{E}_\mu(\mathcal{A})| \le \frac{\sqrt{2}}{n} \sum_{i=1}^n \min \left\{ \mathbb{E}_{\widetilde{W}_i} \left[ \Delta_1(\widetilde{W}_i) \sqrt{I^{\widetilde{W}_i}(\widehat{Z}_i; U_i)} \right], \sqrt{\mathbb{E}_{\widetilde{W}_i} \left[ \Delta_1(\widetilde{W}_i)^2 \right] I(\widehat{Z}_i; U_i | \widetilde{W}_i)} \right\}.$$

*Furthermore, if* $\mathcal{A}$ *is* $\gamma_3$*-SCH-C stable, then such a* $\Delta_1$ *exists and*

$$|\mathcal{E}_\mu(\mathcal{A})| \le \frac{\sqrt{2}\gamma_3}{n} \sum_{i=1}^n \sqrt{\sup_{\tilde{w}_i} I^{\tilde{w}_i}(\widehat{Z}_i; U_i)}.$$

Compared to the standard supersample-conditioned CMI bound [58] in terms of $I(W; U | \widetilde{Z})$, which assesses how well one can infer the "training-set membership" from the output hypothesis, our new CMI quantity, namely $I(\widehat{Z}_i; U_i | \widetilde{W}_i)$, measures our ability to decide if an instance $\widehat{Z}_i$ contributes to the training of $\widetilde{W}_i^+$ or to the training of $\widetilde{W}_i^-$ when we know it contributes to only one of them. When $\widetilde{W}_i^+$ and $\widetilde{W}_i^-$ are similar, this decision (i.e., determining $U_i$) is difficult, giving rise to small $I(\widehat{Z}_i; U_i | \widetilde{W}_i)$. Additionally, for uniformly stable algorithms, $\sup_{w_i} \Delta_1(\tilde{w}_i) \le \beta_2$ and it is also expected that $\mathbb{E}_{\widetilde{W}_i} \left[ \Delta_1(\widetilde{W}_i)^2 \right] \le \beta_1^2$. Moreover, if we simply replace $\gamma_3$ by an upper bound of $\ell$, the second bound of Theorem 4.1 becomes a *uniform convergence* bound if $I^{\tilde{w}}(\widehat{Z}_i; U_i)$ vanishes with $n$.

*Proof Sketch of Theorem 4.1.* Again we find motivation in Lemma 2.1. Additionally, the symmetry exhibited in Eq. (5), analogous to the symmetry between $\widetilde{Z}_i^+$ and $\widetilde{Z}_i^-$ used in deriving the standard CMI bounds, allows a similar development. Specifically, letting $g(\tilde{w}_i, \hat{z}_i, u_i) = (-1)^{u_i} \left( \ell(\tilde{w}_i^-, \hat{z}_i) - \ell(\tilde{w}_i^+, \hat{z}_i) \right)$ and $f = t \cdot g$ for $t > 0$ in Lemma A.2 enables the bounding of the CGF via invoking the assumptions in the theorem. The complete proof is given in Appendix D.1. $\square$

Notably, our new CMI quantity in the bound preserves the boundedness of the original CMI in [58], that is, $I(\widehat{Z}_i; U_i | \widetilde{W}) \le H(U_i) = \log 2$. Furthermore, parallel to supersample-conditioned CMI being smaller than IOMI [18], a similar result for hypotheses-conditioned CMI is given below.

**Theorem 4.2.** *For any* $\mathcal{A}$ *and* $\mu$, *we have* $I(\widehat{Z}_i; U_i | \widetilde{W}_i) \le I(W; Z_i)$.

Similar to Theorem 3.3, we present a CMI bound without square-root, which could be much stronger in certain regimes.

**Theorem 4.3.** *Let* $\Delta_1$ *be defined in the same way as in Theorem 4.1, and we let* $\Lambda(\tilde{w}_i) = \mathbb{E}_{\widehat{Z}_i | \tilde{w}_i} \left[ \left( \ell(\tilde{w}_i^-, \widehat{Z}_i) - \ell(\tilde{w}_i^+, \widehat{Z}_i) \right)^2 \right] \Big/ \Delta_1(\tilde{w}_i)^2$, *then*

$$|\mathcal{E}_\mu(\mathcal{A})| \le \frac{1}{n} \sum_{i=1}^n \mathbb{E}_{\widetilde{W}_i} \left[ \Delta_1(\widetilde{W}_i) \left( I^{\widetilde{W}_i}(\widehat{Z}_i; U_i) + 0.72 \Lambda(\widetilde{W}_i) \right) \right]. \tag{10}$$

*If* $\mathcal{A}$ *is further* $\beta_2$*-uniform stable and* $\gamma_4$*-SCH-D stable, then*

$$|\mathcal{E}_\mu(\mathcal{A})| \le \frac{\beta_2}{n} \sum_{i=1}^n I(\widehat{Z}_i; U_i | \widetilde{W}_i) + 0.72 \frac{\gamma_4^2}{\beta_2}. \tag{11}$$

Note that it is valid to set $\gamma_3 = \mathbb{E}_{\widetilde{W}_i} \left[ \Delta_1(\widetilde{W}_i) \right]$, which can be viewed as a generalization bound in its own right. Additionally, it can be verified that $\Lambda(\tilde{w}_i) \le 1$ for any $\tilde{w}_i$. As $I^{\widetilde{W}_i}(\widehat{Z}_i; U_i) \le \log 2 \approx 0.69$, Eq. (10) can be further upper bounded by $1.41\gamma_3$. This ensures that Eq. (10) will decay no slower than $\mathcal{O}(\gamma_3)$.

We also present a second moment generalization bound.

**Theorem 4.4.** *Assume that $\mathcal{A}$ is $\beta_2$-uniform stable and symmetric with respect to $S$, i.e. it does not depend on the order of the elements in $S$. Let $\ell(\cdot, \cdot) \in [0, 1]$, then*

$$\mathbb{E}_{W,S}\left[(L_\mu(W) - L_S(W))^2\right] \leq 4\beta_2^2\left(\frac{1.5I(E; U|\widetilde{W}) + 0.82}{n} + 1\right) + \frac{1}{n}.$$

Since $I(E; U|\widetilde{W}) \leq \mathcal{O}(n)$, the bound can be further upper bounded by $\mathcal{O}(\beta_2^2 + 1/n)$, which matches the previous tight bound for the second moment generalization error in [14, Thm. 1.2]. Notice that [14, Thm. 1.2] only holds for the deterministic setting, while our bound also holds for randomized algorithms, and we also give a stronger result in Appendix D.4 based on our SCH stability notions.

**Supersample-Conditioned CMI Bounds** It is also possible to give $\widetilde{Z}$-conditioned CMI bounds that explicitly contain the stability parameters. Let $W_i = \widetilde{W}_{i,U_i}$ and $\overline{W}_i = \widetilde{W}_{i,\overline{U}_i}$, we have the following results.

**Theorem 4.5.** *Suppose there exists $\Delta_2 : \mathcal{Z} \to \mathbb{R}$ such that $\sup_{w,w^i \in \mathcal{W}_{z_i}^2} \left|\ell(w, z_i) - \ell(w^i, z_i)\right| \leq \Delta_2(z_i)$ for every $z_i$, where $\mathcal{W}_{z_i}^2$ is the support of the conditional distribution $P_{W,W^i|z_i}$. Then,*

$$|\mathcal{E}_\mu(\mathcal{A})| \leq \frac{\sqrt{2}}{n}\sum_{i=1}^n \min\left\{\mathbb{E}_{\widetilde{Z}_i^+}\left[\Delta_2(\widetilde{Z}_i^+)\sqrt{I^{\widetilde{Z}_i^+}(W_i, \overline{W}_i; U_i)}\right], \sqrt{\mathbb{E}_{\widetilde{Z}_i^+}\left[\Delta_2(\widetilde{Z}_i^+)^2\right]I(W_i, \overline{W}_i; U_i|\widetilde{Z}_i^+)}\right\}.$$

The proof is deferred to Appendix D.5. The interpretation of the CMI quantity $I(W_i, \overline{W}_i; U_i|\widetilde{Z}_i^+)$ *appears* identical to our earlier CMI quantity $I(\widehat{Z}_i; U_i|\widetilde{W}_i)$. A closer look in fact reveals that the two quantities are mathematically equal. To see this, first note $I(\widehat{Z}_i; U_i|\widetilde{W}_i) = H(U_i) - H(U_i|\widehat{Z}_i, \widetilde{W}_i)$ and $I(W_i, \overline{W}_i; U_i|\widetilde{Z}_i^+) = H(U_i) - H(U_i|\widetilde{Z}_i^+, W_i, \overline{W}_i)$. Let random variable $A_1 = (\widehat{Z}_i, \widetilde{W}_i^+, \widetilde{W}_i^-)$ and let $A_2 = (\widetilde{Z}_i^+, W_i, \overline{W}_i)$, these two random variables are identically distribution (since $P_{\widetilde{W}_i^+} = P_{\widetilde{W}_i^-}$, $P_{\widetilde{Z}_i^+} = P_{\widetilde{Z}_i^-}$ and $U_i \sim$ Bernoulli-$(\frac{1}{2})$ and also we have $P_{A_1|U_i} = P_{A_2|U_i}$ so $P_{A_1,U_i} = P_{A_2,U_i}$, which gives us $H(U_i|A_1) = H(U_i|A_2)$. This indicates that $I(\widehat{Z}_i; U_i|\widetilde{W}_i) = I(W_i, \overline{W}_i; U_i|\widetilde{Z}_i^+)$.

In Theorem 4.5, a data-dependent hypothesis space $\mathcal{W}_{z_i}^2$ is defined. A similar concept has been utilized in the hypothesis set cross-validation (CV) stability studied in [16]. Furthermore, [16] derives some bounds based on either their transductive Rademacher complexity or their hypothesis set CV stability. They show that these two notions dominate in different learning scenarios. Given the close relationship between the supersample construction and the Rademacher complexity [58, 64], and the inspiration behind our $\widetilde{W}$ construction, our framework is likely to have a fundamental connection to [16]. Additionally, obtaining the similar results of $\widetilde{Z}$-conditioned CMI as in Theorem 4.3-4.4 is warranted, which may require some stability notions analogous to the average CV-stability in [16].

## 5 Convex–Lipschitz–Bounded (CLB) Problems

We now discuss two examples of the convex-Lipschitz-bounded (CLB) problem, a subclass of SCO problems.

The first example is previously given in [21, Thm. 17], in which nearly all previous information-theoretic bounds are non-vanishing. We will demonstrate that our CMI bounds are non-vacuous in this example.

**Example 1.** *Let $d \in \mathbb{N}$ and $\mathcal{Z} = \{e(i) : i \in [d]\}$ where $e(i)$ is a one-hot vector with 1 at the $i$-th coordinate. Let $\mu = \text{Unif}(\mathcal{Z})$. Given a sample $S = \{Z_i\}_{i=1}^n$ drawn i.i.d. from $\mu$, we choose the 1-Lipschitz convex loss function $\ell(w, z) = -\langle w, z\rangle$ and use GD to select a hypothesis $w$ from $\mathcal{W} = \{w \in \mathbb{R}^d : ||w|| \leq 1\}$. Let the number of GD iterations be $T = n^2$ and let the learning rate be $\eta = \frac{1}{n\sqrt{n}}$.*

Let $\hat{\mu} = \frac{1}{n}\sum_{i=1}^n z_i$ be the sample mean. In this deterministic setting, it's easy to see that

$$w_t = \begin{cases} \eta t\hat{\mu} & \text{if } \eta t||\hat{\mu}|| \leq 1, \\ \eta t\hat{\mu}/||\eta t\hat{\mu}|| & \text{otherwise.} \end{cases}$$

Let $\hat{\mu}^i$ be the sample mean of $s^i$ and let $w_t^i$ be its corresponding hypothesis at time $t$. Notice that Euclidean projection does not increase the distance between projected points, namely non-expansive [22, Lemma 4.6]. Hence, whether $w_t = \eta t \hat{\mu}$ or its truncated version $\eta t \hat{\mu}/||\eta t \hat{\mu}||$ limited within the unit ball, we have $||w_t - w_t^i|| \leq ||\eta t \hat{\mu} - \eta t \hat{\mu}^i|| \leq \mathcal{O}(\eta t/n)$. Recall that the loss function is 1-Lipschitz, we have $|\mathcal{E}_\mu(\mathcal{A})| \leq \beta_2 \leq \mathcal{O}(\eta t/n)$. In this example, $\eta T/n = 1/\sqrt{n}$ so $\beta_2 \in \mathcal{O}(1/\sqrt{n})$. One can also directly obtain this rate from [22, 5].

Now following the same setting in [21], if we let $d = 2n^2$, we can find that $I(W_T; Z_i) \in \Omega(1)$ (see [21, Thm. 17] or Appendix F). Thus, IOMI itself could not explain the generalization of GD in this problem. Furthermore, all our CMI quantities including those in Section 6 also have the order of $\Omega(1)$, that is, they fail to vanish as $n \to \infty$ (see Appendix F for more elaboration).

Therefore, the stability parameter $\beta_2$ should not be replaced by some constant (e.g., the upper bound of the loss function) in the IOMI or CMI bound. In fact, for our CMI bounds, due to their boundedness property, we have the following corollary.

**Corollary 5.1.** *If $\mathcal{A}$ is $\beta_2$-uniform stable, we have $\frac{\beta_2}{n} \sum_{i=1}^n \sqrt{I(\widehat{Z}_i; U_i|\widetilde{W}_i)} \leq \mathcal{O}(\beta_2)$.*

Corollary 5.1 provides a solution to the the non-vanishing limitation of the previous information-theoretic bounds in Example 1 (and also the counterexample in [21, Thm. 4]), as it can explain generalization as long as the stability-based bound is sufficient. Thus, the shortfalls of information-theoretic bounds in analyzing deterministic algorithms for CLB problems are tempered by the stability-based framework.

We now show another CLB Example from [21, Thm. 3] where $\mathcal{A}$ is not uniformly stable, and we will see information-theoretic bounds in this paper are tight up to a constant. This example is also studied in [42, Sec. 5].

**Example 2.** *Let $\mathcal{W} \in \mathbb{R}^d$ be a ball with radius $R_0$, and let the input space be $\mathcal{Z} = \{z_0/R_0, -z_0/R_0\}$ where $z_0 \in \mathcal{W}$ such that $||z_0|| = R_0$. Let $\mu = \mathrm{Unif}(\mathcal{Z})$. Consider a convex and L-Lipschiz loss function $\ell(w, z) = -L\langle w, z\rangle$. In addition, $\mathcal{A}$ is any empirical risk minimization (ERM) algorithm.*

In this example, $\beta_2 = 2LR_0$ is a constant. [21] has shown that $|\mathcal{E}_\mu(\mathcal{A})| \geq \frac{LR_0}{\sqrt{2n}}$ and $I(W; S) \leq 1$ (see [21, Thm. 3] or Appendix F for an explanation). This gives us $\frac{2LR_0}{n} \sum_{i=1}^n \sqrt{I(W; Z_i)} \leq 2LR_0 \sqrt{\frac{I(W;S)}{n}} \leq \frac{2LR_0}{\sqrt{n}}$. Thus, the distribution-dependent property of IOMI can improve the stability-based bound in this case. In addition, we know that $I(\widehat{Z}_i; U_i|\widetilde{W}_i) \leq I(W; Z_i)$ from Theorem 4.2, our new CMI bounds are also tight (up to a constant) in this example.

Notably, we can construct an additional example building upon Example 2, where $\mathcal{A}$ is uniformly stable but the uniform stability itself results in a slow convergence rate for generalization error. Specifically, let $R_0 = \frac{1}{d}$ and let $d = \sqrt{n}$, then $\beta_2 \in \mathcal{O}(1/\sqrt{n})$ while $|\mathcal{E}_\mu(\mathcal{A})| \geq \frac{L}{\sqrt{2}n}$. Note that the information-theoretic bounds in this paper can still provide a tight rate, namely $\mathcal{O}(1/n)$.

These examples demonstrate that our bounds can improve both the stability-based bound and information-theoretic bounds in some learning scenarios.

Additional applications of our bounds are discussed in Appendix G.

# 6 Extensions

**Connection with Bernstein Condition** The Bernstein condition is commonly used to derive fast-rate generalization bound for both PAC-Bayes bounds [69, 10, 36, 17] and stability-based bounds [28], then it is natural to explore the relationship between our fast-rate bounds and the Bernstein condition, formally defined below.

**Definition 6.1** (Bernstein Condition)**.** *Assume that $w^* = \arg\min_{w \in \mathcal{W}} L_\mu(w)$ is a risk minimizer. We say that the Bernstein assumption is satisfied with some $B > 0$ and $\kappa \in [1, +\infty)$ if for any $w \in \mathcal{W}$,*

$$\mathbb{E}_Z \left[ (\ell(w, Z) - \ell(w^*, Z))^2 \right] \leq B \left( L_\mu(w) - L_\mu(w^*) \right)^{\frac{1}{\kappa}}.$$

This condition can be easily satisfied in many common situations [1, 28]. In the following proposition, we can see that the Bernstein condition implies the $\gamma_2$-SCH-B stability.

**Proposition 1.** *If the Bernstein condition is satisfied with some $B$ and $\kappa$, then $\mathcal{A}$ is $\gamma_2$-SCH-B stable, where $\gamma_2^2 = 4B\mathbb{E}_W\left[(L_\mu(W) - L_\mu(w^*))^{\frac{1}{\kappa}}\right]$.*

Therefore, invoking the Bernstein condition, we can obtain the fast-rate bound as presented in Theorem 3.3, where we need to assume the loss is bounded, as shown below as a by-product.

**Corollary 6.1.** *If the Bernstein condition is satisfied with $\kappa = 1$ and $\ell \in [0, C]$, then*

$$|\mathcal{E}_\mu(\mathcal{A})| \leq \frac{C}{n}\sum_{i=1}^n I(W; Z_i) + \frac{2.88B}{C}\left(L_\mu - L_\mu(w^*)\right).$$

Recently, [65, 66] use the unexpected excess risk as the DV auxiliary function and invoke the $(\eta, c)$-central condition to establish some optimal-rate bounds for specific learning problems, e.g., Gaussian mean estimation. This again highlights the significance of selecting appropriate DV auxiliary functions and corresponding assumptions tailored to different learning problems. It is worth mentioning that the Bernstein condition also implies their $(\eta, c)$-central condition. Therefore, unifying these conditions can be considered as a potential avenue for future research.

**Loss Difference, Evaluated, and Functional CMI for Stable Algorithms**  Similar to [64], we derive some tighter bounds based on the loss difference.

**Theorem 6.1.** *Let $\Delta L_i = \ell(W_i, \widetilde{Z}_i^+) - \ell(\overline{W}_i, \widetilde{Z}_i^+)$. If $\mathcal{A}$ is $\beta_2$-uniform stable, then*

$$|\mathcal{E}_\mu(\mathcal{A})| \leq \frac{\sqrt{2}\beta_2}{n}\sum_{i=1}^n \min\left\{\sqrt{I(\Delta L_i; U_i)}, \mathbb{E}_{\widetilde{Z}_i^+}\sqrt{I^{\widetilde{Z}_i^+}(\Delta L_i; U_i)}\right\} \leq \frac{\sqrt{2}\beta_2}{n}\sum_{i=1}^n\sqrt{I(\Delta L_i; U_i|\widetilde{Z}_i^+)}.$$

We note that while it is feasible to replace $\beta_2$ in the (disintegrated) CMI bounds above with certain sample-conditioned hypothesis stability, it is not possible to apply the same substitution for the unconditional MI bound in Theorem 6.1.

Furthermore, notice that $\Delta L_i - (L_i, \bar{L}_i) - (F_i, \overline{F}_i) - (W_i, \overline{W}_i)$ forms a Markov chain given $\widetilde{Z}_i^+$, wherein $(L_i, \bar{L}_i)$ are the loss pair evaluated at $\widetilde{Z}_i^+$ using $(W_i, \overline{W}_i)$, and $(F_i, \overline{F}_i)$ are label predictions of $\widetilde{Z}_i^+$ using $(W_i, \overline{W}_i)$. By the data-processing inequality, one can obtain e-CMI bound [58, 25], $f$-CMI bound [23] and recover $I(W_i, \overline{W}_i; U_i|\widetilde{Z}_i^+)$ based bound from Theorem 6.1: $I(\Delta L_i; U_i|\widetilde{Z}_i^+) \leq I(L_i, \bar{L}_i; U_i|\widetilde{Z}_i^+) \leq I(F_i, \overline{F}_i; U_i|\widetilde{Z}_i^+) \leq I(W_i, \overline{W}_i; U_i|\widetilde{Z}_i^+)$. Additionally, notice that we can also apply the similar technique for the hypotheses-conditioned CMI, which should give the same results.

**Expressiveness of New CMI Notions Under Distribution-Free Setting**  Previous works [58, 19, 20, 23, 25] have demonstrated that the CMI framework is expressive enough to establish connections with VC theory in the distribution-free learning setting. Here, we further illustrate the expressiveness of the sample-conditioned CMI discussed in this work.

**Theorem 6.2.** *Let $\mathcal{Z} = \mathcal{X} \times \{0, 1\}$, and let $\mathcal{F} = \{f_w : \mathcal{X} \to \{0, 1\}|w \in \mathcal{W}\}$ be a functional hypothesis class with finite VC dimension $d$. Let $n > d + 1$, for any algorithm $\mathcal{A}$, we have*

$$\frac{1}{n}\sum_{i=1}^n \sqrt{I(F_i, \bar{F}_i; U_i|\widetilde{Z}_i^+)} \leq \mathcal{O}\left(\sqrt{\frac{d}{n}\log\left(\frac{n}{d}\right)}\right).$$

Like the previous works, this bound matches the classic result of the uniform convergence bound [59]. Notice that the result could be extended to multi-class classification with finite Natarajan dimension [39] by proceeding similarly to [25, Thm. 8].

We invoke Theorem 6.2 to demonstrate that our new information-theoretic quantities have the same expressive power as standard CMI quantities. The expressiveness result for the (functional) hypotheses-conditioned CMI is expected to align with Theorem 6.2. This alignment is due to the equivalence between hypotheses-conditioned CMI and supersample-conditioned CMI, as discussed in Theorem 4.5.

## 7  Related Works and Additional Discussions

**Stability-Based Framework vs. Information-Theoretic Framework**  Using stability methods to analyze generalization errors can be traced back to several seminal works, such as [51, 11, 12,

34, 27]. It is worth noting that stability arguments have proven particularly effective in analyzing the learnability of SCO problems [56], where traditional uniform convergence bounds may not be sufficient to explain the generalization behavior. The application of stability approaches has gained popularity for providing high-probability guarantees since the work of [6]. Recent advancements have further sharpened the convergence rates of high-probability generalization upper bounds for uniformly stable algorithms in a series of works [14, 15, 8, 28]. While information-theoretic bounds are commonly used to analyze the in-expectation generalization, it is expected that combining them with the stability framework will yield sharper high-probability bounds.

Additionally, we note that in the realizable setting, where $\mathcal{A}$ is an interpolating algorithm, information-theoretic bounds exhibit greater power than stability-based bounds. For instance, in the case of the $0-1$ loss, information-theoretic bounds can achieve the known optimal minimax rates [19] and even provide exact characterizations of the generalization error [20, 64]. However, it should be noted that due to the inherent fitting-stability tradeoff property [54, Sec. 13.4], interpolating algorithms tend to be unstable, rendering stability arguments inapplicable. Given the prevalence of zero empirical risk in modern deep learning [68], it is natural to question whether information-theoretic bounds require the stability-based approach for analyzing non-convex (and potentially non-smooth and non-Lipschitz continuous) learning scenarios.

**Connection Between Two Frameworks in Previous Works**   The connection between information-theoretic bounds, including some PAC-Bayes bounds, and algorithmic stability has been explored in previous literature [46, 33, 48, 29, 58, 23, 3, 47]. These works primarily focus on either regarding the information-theoretic quantities as notions of distributional stability [46, 67, 29, 58, 3] and/or converting information-theoretic quantities to some other algorithmic stability notions [58, 23, 47]. The later often relies on the addition of Gaussian noise to the hypotheses (or assuming that the prior and posterior distributions are Gaussian in PAC-Bayes). In [33], the authors also combine the DV formula with stability assumptions, where they derive some PAC-Bayes bounds. However, comparing these bounds with others is challenging in general due to the presence of their hyperparameter stability.

**Comparison with Standard CMI**   While our IOMI quantity aligns with the previous work [9], the new CMI quantity $I(\widehat{Z}_i; U_i | \widetilde{W}_i)$ (or equivalently $I(W_i, \overline{W}_i; U_i | \widetilde{Z}_i^+)$) may not be directly comparable to the standard individual CMI $I(W; U_i | \widetilde{Z}_i)$ in [49, 70]. Specifically, we have $I(\widehat{Z}_i; U_i | \widetilde{W}_i) = H(U_i) - H(U_i | \widehat{Z}_i, \widetilde{W}_i)$ and $I(W; U_i | \widetilde{Z}_i) = H(U_i) - H(U_i | W, \widetilde{Z}_i)$. The relationship between $H(U_i | \widehat{Z}_i, \widetilde{W}_i)$ and $H(U_i | W, \widetilde{Z}_i)$ is not trivial. Exploring and quantitatively comparing these CMI measures would be an intriguing research direction.

**Leave-One-Out CMI**   Our construction of $\widetilde{W}$ bears resemblance to the leave-one-out (LOO) setting, where there are also $n+1$ hypotheses. It is worth noting that LOO-CMI has been recently proposed in concurrent works [20, 47]. In LOO-CMI, the supersample $\widetilde{Z} = Z_{[n+1]}$ consists of $n+1$ instances, and $U$ is an index uniformly drawn from $[n+1]$ to select one hold-out instance. Consequently, $Z_U$ represents the testing data, while $Z_{[n+1]\setminus U}$ serves as the training sample. In this context, LOO-CMI can be defined as $I(W; U | \widetilde{Z})$. Notice that this quantity still fails to explain the generalization in Example 1. The LOO setting is often associated with the stability-based framework [6], and it is expected that the $n+1$-supersample induced $\widetilde{W}$ could yield new CMI bounds that also contain the SCH stability notions. Nevertheless, in this paper, we do not adopt the LOO setting because in that case, $H(U) = \log(n+1)$, the LOO-CMI bound is no longer upper bounded by a constant independent of $n$ (note that the LOO-CMI bound for general setting in [20, Thm. 2.5] does not contain the $1/\sqrt{n}$ factor).

# 8   Concluding Remarks

We propose a novel construction of the hypothesis matrix and a new family of stability notions called sample-conditioned hypothesis stability. Leveraging these concepts, we derive sharper information-theoretic bounds for stable learning algorithms. Several promising avenues for future research include comparing our new CMI quantities with the standard CMI in a quantitative manner, analyzing the generalization of gradient-based optimization algorithms like SGD using our bounds, and establishing new high-probability generalization guarantees. Further discussions can be found in Appendix H.

## Acknowledgements

This work is supported partly by an NSERC Discovery grant. Ziqiao Wang is also supported in part by the NSERC CREATE program through the Interdisciplinary Math and Artificial Intelligence (INTER-MATH-AI) project. The authors would like to thank the anonymous AC and reviewers for their careful reading and valuable suggestions.

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

# Appendices

## A   Some Useful Lemmas

In this paper, there are some equivalent forms of the generalization error we will study, e.g., Eq. (2) and Eq. (5) in the main text, which are presented in the following lemma.

**Lemma A.1.** *Let* $W_i = \widetilde{W}_{i,U_i}$ *and* $\overline{W}_i = \widetilde{W}_{i,\overline{U}_i}$. *For any learning algorithm* $\mathcal{A}$, *the following equations hold*

$$\mathcal{E}_\mu(\mathcal{A}) = \frac{1}{n}\sum_{i=1}^{n} \mathbb{E}_{\widetilde{Z}_i^+,\widetilde{W}_i}\left[\ell(\widetilde{W}_i^-,\widetilde{Z}_i^+) - \ell(\widetilde{W}_i^+,\widetilde{Z}_i^+)\right], \tag{12}$$

$$= \frac{1}{n}\sum_{i=1}^{n} \mathbb{E}_{\widetilde{W}_i}\left[\mathbb{E}_{\widehat{Z}_i,U_i|\widetilde{W}_i}\left[(-1)^{U_i}\left(\ell(\widetilde{W}_i^-,\widehat{Z}_i) - \ell(\widetilde{W}_i^+,\widehat{Z}_i)\right)\right]\right], \tag{13}$$

$$= \frac{1}{n}\sum_{i=1}^{n} \mathbb{E}_{\widetilde{Z}_i^+}\left[\mathbb{E}_{W_i,\overline{W}_i,U_i|\widetilde{Z}_i^+}\left[(-1)^{U_i}\left(\ell(\overline{W}_i,\widetilde{Z}_i^+) - \ell(W_i,\widetilde{Z}_i^+)\right)\right]\right]. \tag{14}$$

*Proof.* This lemma is a consequence of Lemma 2.1, with further utilizing some symmetric properties. Recall Eq. (1) in Lemma 2.1,

$$\mathcal{E}_\mu(\mathcal{A}) = \mathbb{E}_{\widetilde{Z}_{[n]}^+,\widetilde{Z}_{[n]}^-}\left[\frac{1}{n}\sum_{i=1}^{n}\left[\mathbb{E}_{\widetilde{W}_i^-|\widetilde{Z}_{[n]}^+,\widetilde{Z}_i^-}\ell(\widetilde{W}_i^-,\widetilde{Z}_i^+) - \mathbb{E}_{\widetilde{W}^+|\widetilde{Z}_{[n]}^+}\ell(\widetilde{W}^+,\widetilde{Z}_i^+)\right]\right],$$

$$= \mathbb{E}_{\widetilde{Z}_{[n]}^+,\widetilde{W}}\left[\frac{1}{n}\sum_{i=1}^{n}\left[\ell(\widetilde{W}_i^-,\widetilde{Z}_i^+) - \ell(\widetilde{W}^+,\widetilde{Z}_i^+)\right]\right],$$

$$= \frac{1}{n}\sum_{i=1}^{n} \mathbb{E}_{\widetilde{Z}_i^+,\widetilde{W}_i}\left[\ell(\widetilde{W}_i^-,\widetilde{Z}_i^+) - \ell(\widetilde{W}^+,\widetilde{Z}_i^+)\right].$$

Note that Eq. (2) in the main text is from the second equation above, which is used to derive individual IOMI bounds in Section 3.

Similar to the standard setting for CMI bounds, where the role of each $\widetilde{Z}_i^+$ and $\widetilde{Z}_i^-$ can be exchanged, a key observation here is that for each $i$, $\widetilde{W}_i^+$ and $\widetilde{W}_i^-$ can also be exchanged arbitrarily. That is to say,

$$\mathcal{E}_\mu(\mathcal{A}) = \frac{1}{n}\sum_{i=1}^{n} \mathbb{E}_{\widetilde{Z}_i^-,\widetilde{W}_i}\left[\ell(\widetilde{W}_i^+,\widetilde{Z}_i^-) - \ell(\widetilde{W}_i^-,\widetilde{Z}_i^-)\right] \tag{15}$$

also holds true. Notice that we do not change the definitions of any the random variable, e.g., $\widetilde{W}^+ = \mathcal{A}(\widetilde{Z}_{[n]}^+, R)$ and $\widetilde{W}_i^- = \mathcal{A}(\widetilde{Z}_{[n]\sim i}^+, R)$.

What differs from the standard CMI is that the roles of the whole sequences $\widetilde{Z}_{[n]}^+$ and $\widetilde{Z}_{[n]}^-$ are not exchangeable with each other. Here, when we exchange each $\widetilde{Z}_i^+$ and $\widetilde{Z}_i^-$, we need to keep the other positions in $S$ unchanged.

By introducing $U_i \sim \text{Unif}(\{0,1\})$, we have

$$\mathcal{E}_\mu(\mathcal{A}) = \frac{1}{n}\sum_{i=1}^{n} \mathbb{E}_{\widetilde{Z}_i^+,\widetilde{W}_i}\left[\ell(\widetilde{W}_i^-,\widetilde{Z}_i^+) - \ell(\widetilde{W}_i^+,\widetilde{Z}_i^+)\right],$$

$$= \frac{1}{n}\sum_{i=1}^{n} \mathbb{E}_{\widetilde{Z}_i,\widetilde{W}_i,U_i}\left[\ell(\widetilde{W}_{i,\overline{U}_i},\widetilde{Z}_{i,U_i}) - \ell(\widetilde{W}_{i,U_i},\widetilde{Z}_{i,U_i})\right].$$

To obtain Eq. (13), notice that $\widehat{Z}_i = \widetilde{Z}_{i,U_i}$, we have

$$\mathcal{E}_\mu(\mathcal{A}) = \frac{1}{n} \sum_{i=1}^n \mathbb{E}_{\widehat{Z}_i, \widetilde{W}_i, U_i} \left[ \ell(\widetilde{W}_{i,\overline{U}_i}, \widehat{Z}_i) - \ell(\widetilde{W}_{i,U_i}, \widehat{Z}_i) \right]$$

$$= \frac{1}{n} \sum_{i=1}^n \mathbb{E}_{\widehat{Z}_i, \widetilde{W}_i, U_i} \left[ (-1)^{U_i} \left( \ell(\widetilde{W}_i^-, \widehat{Z}_i) - \ell(\widetilde{W}_i^+, \widehat{Z}_i) \right) \right]. \qquad (16)$$

This, as we have already seen in Eq. (5) in the main text, is used to derive hypotheses-conditioned CMI bounds in Section 4. It's easy to see that when $U_i = 0$, Eq. (16) becomes Eq. (12), and when $U_i = 1$, we obtain Eq. (15) via Eq. (16).

To obtain Eq. (14), we let $W_i = \widetilde{W}_{i,U_i}$, $\overline{W}_i = \widetilde{W}_{i,\overline{U}_i}$, and fix $\widehat{Z}_i = \widetilde{Z}_i^+$. Similarly,

$$\mathcal{E}_\mu(\mathcal{A}) = \frac{1}{n} \sum_{i=1}^n \mathbb{E}_{\widetilde{Z}_i^+} \left[ \mathbb{E}_{W_i, \overline{W}_i, U_i | \widetilde{Z}_i^+} \left[ (-1)^{U_i} \left( \ell(\overline{W}_i, \widetilde{Z}_i^+) - \ell(W_i, \widetilde{Z}_i^+) \right) \right] \right].$$

This is used to derive supersample-conditioned CMI bounds in Section 4. It's easy to see that both $U_i = 0$ and $U_i = 1$ will give us Eq. (12). $\qquad \square$

Like all the previous information-theoretic bounds, the following lemma is widely used in our paper.

**Lemma A.2** (Donsker-Varadhan (DV) variational representation of KL divergence [44, Theorem 3.5])**.** *Let $Q$, $P$ be probability measures on $\Theta$, for any bounded measurable function $f : \Theta \to \mathbb{R}$, we have* $D_{\mathrm{KL}}(Q||P) = \sup_f \mathbb{E}_{\theta \sim Q}[f(\theta)] - \ln \mathbb{E}_{\theta \sim P}[\exp f(\theta)]$.

We also invoke some other lemmas as given below.

**Lemma A.3** (Hoeffding's Lemma [26])**.** *Let $X \in [a, b]$ be a bounded random variable with mean $\mu$. Then, for all $t \in \mathbb{R}$, we have $\mathbb{E}\left[e^{tX}\right] \leq e^{t\mu + \frac{t^2(b-a)^2}{8}}$.*

**Lemma A.4** (Popoviciu's inequality [45])**.** *Let $M$ and $m$ be upper and lower bounds on the values of any random variable $X$, then $\mathrm{Var}(X) \leq \frac{(M-m)^2}{4}$.*

The following lemma is from [35, Lemma 2.8], we provide a self-contained proof.

**Lemma A.5.** *Let $h(x) = \frac{e^x - x - 1}{x^2}$ be the Bernstein function. If a random variable $X$ satisfies $\mathbb{E}[X] = 0$ and $X \leq b$, then $\mathbb{E}\left[e^X\right] \leq e^{h(b)\mathbb{E}[X^2]}$.*

*Proof.* It's easy to verify that $h(x)$ is an increasing function for $x > 0$. Thus, $h(x) \leq h(b)$ for $x \leq b$. Then,

$$e^x = x + 1 + x^2 h(x) \leq x + 1 + x^2 h(b).$$

For the bounded random variable $X$ with zero mean, we have

$$\mathbb{E}\left[e^X\right] \leq \mathbb{E}[X] + 1 + \mathbb{E}\left[X^2 h(b)\right] \leq e^{h(b)\mathbb{E}[X^2]}.$$

The last inequality is by $e^x \geq x + 1$. This completes the proof. $\qquad \square$

## B  Further Elaborations on SCH Stability

We note that the reason we introduce four types of SCH stability in Definition 2.1 is that solely using $\beta_2$ in our bounds might be too loose, as it considers the supremum over all sources of randomness. By incorporating SCH stabilities, we aim to demonstrate that theoretically, we can achieve significantly tighter stability parameters.

The basic set up is as follows. Assume a random sample $S$ gives rise to $W$. For each $Z_i \in S$, we construct $S^i$ by replacing $Z_i$ with another independently drawn instance; call training result $W^i$, the neighbor of $W$.

In a), $\gamma_1$-SCH-A stability measures the difference between the loss of $w$ and the expected loss of its neighbor $W^i$ at a worst $z$ and the worst possible $w$. While in (b), $\gamma_2$-SCH-B stability measures the

square of this difference, not in the worst case, but in an average case, where the average is over an independently $Z'$ for the loss evaluation, the training sample, and the algorithm randomness. Since "average is smaller than worst", $\gamma_2 \leq \gamma_1$.

In c), we consider the difference between the loss of $W$ and the loss of its neighbor when evaluated at the worst possible $Z_i$ that when included in $S$ gives rise to $W$. The expected value of this difference is $\gamma_3$-SCH-C stability.

In d), $\gamma_4$-SCH-D stability measures the expected squared difference between the loss of $W$ and the loss of its neighbor when evaluated at $Z_i$ (a member of $S$). For a similar "average smaller worst" reason, one expects that $\gamma_4 \leq \gamma_3$.

We expect that $\gamma_2$, $\gamma_3$, and $\gamma_4$ are all smaller than $\beta_1$. This is because in $\beta_1$, we consider the worst evaluated instance, whereas in the other cases, we take the expectation over all instances. Additionally, in Theorem 4.1, we expect that $\mathbb{E}_{\widetilde{W}_i} \Delta_1(\widetilde{W}_i)^2 \leq \beta_1^2$, this is because $\beta_1$-stability holds for all the possible $s$ and $s^i$, namely it holds for all the $(w, w^i)$ pair (that shares the same randomness) while in $\mathbb{E}_{\widetilde{W}_i} \Delta_1(\widetilde{W}_i)^2$, we take the expectation of these pairs.

We expect $\gamma_2 \leq \gamma_4$ due to the following reason: first by Jensen's inequality, we have $\mathbb{E}_{S,R,Z'} \left[ \ell(W, Z') - \mathbb{E}_{W^i|W} \ell(W^i, Z') \right]^2 \leq \mathbb{E}_{W,W',Z'} \left[ \ell(W, Z') - \ell(W^i, Z') \right]^2$, then since $Z'$ is an independent of both $W$ and $W'$, $Z'$ can be regarded as a testing point for both $W$ and $W'$, we could expect that the expectation of $\ell(W, Z') - \ell(W^i, Z')$ is small. While in $\mathbb{E}_{S,Z_i',R} \left[ \ell(W, Z_i) - \ell(W^i, Z_i) \right]^2$, $Z_i$ is a training point for obtaining $W$, so $\ell(W, Z_i)$ could be small in general, and $Z_i$ is a testing point for $W^i$. Therefore, it is reasonable to expect $\mathbb{E}_{W,W^i,Z'} \left[ \ell(W, Z') - \ell(W^i, Z') \right]^2 \leq \mathbb{E}_{S,Z_i',R} \left[ \ell(W, Z_i) - \ell(W^i, Z_i) \right]^2$, namely $\gamma_2 \leq \gamma_4$.

As a concrete example, let $\ell$ be zero-one loss and assume $\mathcal{A}$ is an interpolating algorithm and and randomly makes predictions for unseen data. By Jensen's inequality, $\gamma_2^2 \leq \mathbb{E}_{W,W^i,Z'} \left[ \ell(W, Z') - \ell(W^i, Z') \right]^2 = \mathbb{E}_{W,Z'} \left[ \ell(W, Z') \right] - 2\mathbb{E}_{W,W^i,Z'} \left[ (\ell(W, Z')\ell(W^i, Z')) \right] + \mathbb{E}_{W^i,Z'} \left[ \ell(W^i, Z') \right]^2$, where we use $\ell^2 = \ell$ for zero-one loss. Since $Z'$ is an unseen data for both $W$ and $W^i$, we have $\gamma_2^2 \leq \mathbb{E}_{W^i,Z'} \left[ \ell(W^i, Z') \right]^2 + \frac{1}{2} - \frac{1}{2} = \mathbb{E}_{W^i,Z'} \left[ \ell(W^i, Z') \right]^2$. While in this case $\gamma_4^2 = \mathbb{E}_{W^i,Z_i} \left[ \ell(W^i, Z_i) \right]^2$ so $\gamma_2 \leq \gamma_4$.

## C    Omitted Proofs and Additional Discussions in Section 3

### C.1    Proof of Theorem 3.1

*Proof.* Let $g(\tilde{w}^+, \tilde{z}_i^+) = \mathbb{E}_{\widetilde{W}_i^-|\tilde{w}^+} \left[ \ell(\widetilde{W}_i^-, \tilde{z}_i^+) \right] - \ell(\tilde{w}^+, \tilde{z}_i^+)$ be the average loss difference between $\tilde{w}^+$ and its neighboring hypothesis, and let $f = t \cdot g$ for $t > 0$ in Lemma A.2. Let $\widetilde{Z}_i^{+'}$ be an independent copy of $\widetilde{Z}_i^+$, then

$$\mathbb{E}_{\widetilde{W}^+, \widetilde{Z}_i^+} \left[ g(\widetilde{W}^+, \widetilde{Z}_i^+) \right] \leq \inf_{t>0} \frac{I(\widetilde{W}^+; \widetilde{Z}_i^+) + \log \mathbb{E}_{\widetilde{W}^+, \widetilde{Z}_i^{+'}} \left[ e^{tg(\widetilde{W}^+, \widetilde{Z}_i^{+'})} \right]}{t}. \qquad (17)$$

Since $\widetilde{Z}_i^{+'}$ is independent of both $\widetilde{W}_i^-$ and $\widetilde{W}^+$, and $\widetilde{W}_i^-$ and $\widetilde{W}^+$ are identically distributed, we have

$$\mathbb{E}_{\widetilde{W}^+, \widetilde{Z}_i^{+'}} \left[ g(\widetilde{W}^+, \widetilde{Z}_i^{+'}) \right] = \mathbb{E}_{\widetilde{W}_i^-, \widetilde{Z}_i^{+'}} \left[ \ell(\widetilde{W}_i^-, \widetilde{Z}_i^{+'}) \right] - \mathbb{E}_{\widetilde{W}^+, \widetilde{Z}_i^{+'}} \left[ \ell(\widetilde{W}^+, \widetilde{Z}_i^{+'}) \right] = 0.$$

By the definition of $\gamma_1$-SCH-A stability,

$$\sup_{\tilde{w}^+, z} \left| \mathbb{E}_{\widetilde{W}_i^-|\tilde{w}^+} \left[ \ell(\widetilde{W}_i^-, z) \right] - \ell(\tilde{w}^+, z) \right| \leq \gamma_1,$$

so $g(\widetilde{W}^+, \widetilde{Z}_i^{+'})$ is a zero-mean random variable bounded in $[-\gamma_1, \gamma_1]$. By Lemma A.3, we have

$$
\begin{aligned}
\mathbb{E}_{\widetilde{W}^+, \widetilde{Z}_i^+} \left[ g(\widetilde{W}^+, \widetilde{Z}_i^+) \right] &\leq \inf_{t>0} \frac{I(\widetilde{W}^+; \widetilde{Z}_i^+) + \log \mathbb{E}_{\widetilde{W}^+, \widetilde{Z}_i^{+'}} \left[ e^{tg(\widetilde{W}^+, \widetilde{Z}_i^{+'})} \right]}{t} \\
&\leq \inf_{t>0} \frac{I(\widetilde{W}^+; \widetilde{Z}_i^+) + \frac{t^2 \gamma_1^2}{2}}{t} \\
&= \sqrt{2\gamma_1^2 I(\widetilde{W}^+; \widetilde{Z}_i^+)},
\end{aligned}
$$

where the last equality is obtained by optimizing the bound over $t$, i.e. letting $t = \sqrt{\frac{I(\widetilde{W}^+; \widetilde{Z}_i^+)}{2\gamma_1^2}}$.

Recall Eq. (12) in Lemma A.1 and applying Jensen's inequality to the absolute function, the first bound is then obtained by

$$
|\mathcal{E}_\mu(\mathcal{A})| \leq \frac{1}{n} \sum_{i=1}^n \left| \mathbb{E}_{\widetilde{W}^+, \widetilde{Z}_i^+} \left[ g(\widetilde{W}^+, \widetilde{Z}_i^+) \right] \right| \leq \frac{\gamma_1}{n} \sum_{i=1}^n \sqrt{2 I(\widetilde{W}^+; \widetilde{Z}_i^+)},
$$

Furthermore, by the chain rule of mutual information,

$$
I(\widetilde{W}_i^-; \widetilde{Z}_i^+ | \widetilde{W}^+) + I(\widetilde{W}^+; \widetilde{Z}_i^+) = I(\widetilde{W}^+; \widetilde{Z}_i^+ | \widetilde{W}_i^-) + I(\widetilde{W}_i^-; \widetilde{Z}_i^+). \tag{18}
$$

Notice that $I(\widetilde{W}_i^-; \widetilde{Z}_i^+) = 0$ in the RHS, we have

$$
I(\widetilde{W}^+; \widetilde{Z}_i^+) \leq I(\widetilde{W}^+; \widetilde{Z}_i^+ | \widetilde{W}_i^-),
$$

which will give us the second bound. This concludes the proof. $\qquad\square$

**Remark C.1** (Comparison with Mutual Information Stability [46, 23]). *To compare with the mutual information stability $I\left( \widetilde{W}^+; \widetilde{Z}_i^+ | \widetilde{Z}_{[n]\backslash i}^+ \right)$, recall Eq.(18): $I(\widetilde{W}^+; \widetilde{Z}_i^+ | \widetilde{W}^-) = I(\widetilde{W}_i^-; \widetilde{Z}_i^+ | \widetilde{W}^+) + I(\widetilde{W}^+; \widetilde{Z}_i^+)$, and similarly we also have $I(\widetilde{W}^+; \widetilde{Z}_i^+ | \widetilde{Z}_{[n]\backslash i}^+) = I(\widetilde{Z}_{[n]\backslash i}^+; \widetilde{Z}_i^+ | \widetilde{W}^+) + I(\widetilde{W}^+; \widetilde{Z}_i^+)$.*

*Thus, we only need to compare $I(\widetilde{W}_i^-; \widetilde{Z}_i^+ | \widetilde{W}^+)$ and $I(\widetilde{Z}_{[n]\backslash i}^+; \widetilde{Z}_i^+ | \widetilde{W}^+)$. Notice that for a deterministic $\mathcal{A}$, we have $I(\widetilde{W}_i^-; \widetilde{Z}_i^+ | \widetilde{W}^+) \leq I(\widetilde{Z}_{[n]\backslash i}^+, \widetilde{Z}_i^-; \widetilde{Z}_i^+ | \widetilde{W}^+)$. Since $\widetilde{Z}_i^- \perp\!\!\!\perp \left( \widetilde{W}^+, \widetilde{Z}_{[n]}^+ \right)$, we further have $I(\widetilde{Z}_{[n]\backslash i}^+, \widetilde{Z}_i^-; \widetilde{Z}_i^+ | \widetilde{W}^+) = I(\widetilde{Z}_{[n]\backslash i}^+; \widetilde{Z}_i^+ | \widetilde{W}^+)$, which gives us the desired result:*

$$
I(\widetilde{W}_i^+; \widetilde{Z}_i^+ | \widetilde{W}^-) \leq I\left( \widetilde{W}^+; \widetilde{Z}_i^+ | \widetilde{Z}_{[n]\backslash i}^+ \right).
$$

## C.2 Proof of Theorem 3.2

*Proof.* The proof is nearly the same to the proof of Theorem 3.1, except that now the randomness of the algorithm is given for each DV auxiliary function, so the randomness of $\widetilde{W}_i$ is completely controlled by $\widetilde{Z}$.

Let $g(\tilde{w}^+, \tilde{z}_i^+, r) = \mathbb{E}_{\widetilde{W}_i^- | \tilde{w}^+, r} \left[ \ell(\widetilde{W}_i^-, \tilde{z}_i^+) \right] - \ell(\tilde{w}^+, \tilde{z}_i^+)$ and let $f = t \cdot g$ for $t > 0$ in Lemma A.2. Let $\widetilde{Z}_i^{+'}$ be an independent copy of $\widetilde{Z}_i^+$, then

$$
\mathbb{E}_{\widetilde{W}^+, \widetilde{Z}_i^+ | r} \left[ g(\widetilde{W}^+, \widetilde{Z}_i^+, r) \right] \leq \inf_{t>0} \frac{I(\widetilde{W}^+; \widetilde{Z}_i^+ | R = r) + \log \mathbb{E}_{\widetilde{W}^+, \widetilde{Z}_i^{+'} | r} \left[ e^{tg(\widetilde{W}^+, \widetilde{Z}_i^{+'}, r)} \right]}{t}.
$$

Notice that

$$
\mathbb{E}_{\widetilde{W}^+, \widetilde{Z}_i^{+'} | r} \left[ g(\widetilde{W}^+, \widetilde{Z}_i^{+'}, r) \right] = \mathbb{E}_{\widetilde{W}_i^-, \widetilde{Z}_i^{+'} | r} \left[ \ell(\widetilde{W}_i^-, \widetilde{Z}_i^{+'}) \right] - \mathbb{E}_{\widetilde{W}_i^+, \widetilde{Z}_i^{+'} | r} \left[ \ell(\widetilde{W}_i^+, \widetilde{Z}_i^{+'}) \right] = 0
$$

still holds since $\widetilde{Z}_i^+$ and $\widetilde{Z}_i^-$ are i.i.d. drawn.

Thus, $g(\widetilde{W}^+, \widetilde{Z}_i^{+'}, r)$ is a zero-mean random variable bounded in $[-\beta_2, \beta_2]$. By Lemma A.3, the remaining part is routine:

$$\mathbb{E}_{\widetilde{W}^+, \widetilde{Z}_i^+ | r}\left[ g(\widetilde{W}^+, \widetilde{Z}_i^+, r) \right] \leq \sqrt{2\beta_2^2 I(\widetilde{W}^+; \widetilde{Z}_i^+ | R = r)}.$$

Thus,

$$|\mathcal{E}_\mu(\mathcal{A})| \leq \frac{1}{n} \sum_{i=1}^n \left| \mathbb{E}_{\widetilde{W}^+, \widetilde{Z}_i^+, R}\left[ g(\widetilde{W}^+, \widetilde{Z}_i^+, R) \right] \right| \leq \frac{\beta_2}{n} \sum_{i=1}^n \mathbb{E}_R \sqrt{2I^R(\widetilde{W}^+; \widetilde{Z}_i^+)},$$

This completes the proof. $\qquad\square$

### C.3 Proof of Theorem 3.3

*Proof.* Let $h(x) = \frac{e^x - x - 1}{x^2}$ be the Bernstein function. Similar to the proof of Theorem 3.1, we let $g(\tilde{w}^+, \tilde{z}_i^+) = \mathbb{E}_{\widetilde{W}_i^- | \tilde{w}^+}\left[ \ell(\widetilde{W}_i^-, \tilde{z}_i^+) \right] - \ell(\tilde{w}^+, \tilde{z}_i^+)$. We have already known that $\mathbb{E}_{\widetilde{W}^+, \widetilde{Z}_i^{+'}}\left[ g(\widetilde{W}^+, \widetilde{Z}_i^{+'}) \right] = 0$ and $\left| g(\widetilde{W}^+, \widetilde{Z}_i^{+'}) \right| \leq \gamma_1$. By Lemma A.5,

$$\begin{aligned}
\log \mathbb{E}_{\widetilde{W}^+, \widetilde{Z}_i^{+'}}\left[ e^{tg(\widetilde{W}^+, \widetilde{Z}_i^{+'})} \right] &\leq h(\gamma_1 t)t^2 \mathbb{E}_{\widetilde{W}^+, \widetilde{Z}_i^{+'}}\left[ \left( \mathbb{E}_{\widetilde{W}_i^- | \widetilde{W}^+}\left[ \ell(\widetilde{W}_i^-, \widetilde{Z}_i^{+'}) \right] - \ell(\widetilde{W}^+, \widetilde{Z}_i^{+'}) \right)^2 \right] \\
&\leq h(\gamma_1 t)t^2 \gamma_2^2,
\end{aligned}$$

where the second inequality is by the definition of $\gamma_2$-SCH-B stability.

Plugging the above into Eq. (17),

$$\begin{aligned}
\mathbb{E}_{\widetilde{W}^+, \widetilde{Z}_i^+}\left[ g(\widetilde{W}^+, \widetilde{Z}_i^+) \right] &\leq \inf_{t>0} \frac{I(\widetilde{W}^+; \widetilde{Z}_i^+) + \log \mathbb{E}_{\widetilde{W}^+, \widetilde{Z}_i^{+'}}\left[ e^{tg(\widetilde{W}^+, \widetilde{Z}_i^{+'})} \right]}{t} \\
&\leq \inf_{t>0} \frac{I(\widetilde{W}^+; \widetilde{Z}_i^+)}{t} + h(\gamma_1 t)t\gamma_2^2.
\end{aligned}$$

Usually we have $\gamma_2^2 \leq \gamma_1^2 \leq \gamma_1$, we let $t = 1/\gamma_1$, then

$$h(\gamma_1 t)t\gamma_2^2 = \frac{h(1)\gamma_2^2}{\gamma_1} \approx 0.72 \frac{\gamma_2^2}{\gamma_1}.$$

Thus,

$$|\mathcal{E}_\mu(\mathcal{A})| \leq \frac{\gamma_1}{n} \sum_{i=1}^n I(\widetilde{W}^+; \widetilde{Z}_i^+) + \frac{0.72\gamma_2^2}{\gamma_1}.$$

This concludes the proof. $\qquad\square$

## D Omitted Proofs in Section 4

### D.1 Proof of Theorem 4.1

*Proof.* We now prove the first bound. Let $g(\tilde{w}_i, \hat{z}_i, u_i) = (-1)^{u_i}\left( \ell(\tilde{w}_i^-, \hat{z}_i) - \ell(\tilde{w}_i^+, \hat{z}_i) \right)$. By Lemma A.2, we have

$$\mathbb{E}_{\widehat{Z}_i, U_i | \tilde{w}_i}\left[ g(\tilde{w}_i, \widehat{Z}_i, U_i) \right] \leq \inf_{t>0} \frac{I(\widehat{Z}_i; U_i | \widetilde{W}_i = \tilde{w}_i) + \log \mathbb{E}_{\widehat{Z}_i, U_i' | \tilde{w}_i}\left[ e^{tg(\tilde{w}_i, \widehat{Z}_i, U_i')} \right]}{t}. \tag{19}$$

Since $U_i' \perp\!\!\!\perp \widehat{Z}_i$, we have $\mathbb{E}_{U_i'}[g(\tilde{w}_i, \hat{z}_i, U_i')] = \mathbb{E}_{U_i'}\left[ (-1)^{U_i'}\left( \ell(\tilde{w}_i^-, \hat{z}_i) - \ell(\tilde{w}_i^+, \hat{z}_i) \right) \right] = 0$ for any $\tilde{w}_i$ and $\hat{z}_i$. Ergo,

$$\mathbb{E}_{\widehat{Z}_i | \tilde{w}_i}\left[ \mathbb{E}_{U_i'}\left[ g(\tilde{w}_i, \widehat{Z}_i, U_i') \right] \right] = 0.$$

By the definition of $\Delta_1(\tilde{w}_i)$,

$$\left|g(\tilde{w}_i, \widehat{Z}_i, U_i')\right| = \left|\ell(\tilde{w}_i^-, \widehat{Z}_i) - \ell(\tilde{w}_i^+, \widehat{Z}_i)\right| \leq \sup_{z_i \in \mathcal{Z}_{\tilde{w}_i}} \left|\ell(\tilde{w}_i^-, z_i) - \ell(\tilde{w}_i^+, z_i)\right| \leq \Delta_1(\tilde{w}_i).$$

Thus, $g(\tilde{w}_i, \widehat{Z}_i, U_i')$ is a zero-mean random variable bounded in $[-\Delta_1(\tilde{w}_i), \Delta_1(\tilde{w}_i)]$ for a fixed $\tilde{w}_i$. By Lemma A.3, we have

$$\mathbb{E}_{\widehat{Z}_i, U_i' | \tilde{w}_i}\left[e^{tg(\tilde{w}, \widehat{Z}_i, U_i')}\right] \leq e^{\frac{t^2 \Delta_1(\tilde{w}_i)^2}{2}}.$$

Plugging the above into Eq. (19),

$$\mathbb{E}_{\widehat{Z}_i, U_i | \tilde{w}_i}\left[g(\tilde{w}_i, \widehat{Z}_i, U_i)\right] \leq \inf_{t>0} \frac{I(\widehat{Z}_i; U_i | \widetilde{W}_i = \tilde{w}_i) + \frac{t^2 \Delta_1(\tilde{w}_i)^2}{2}}{t} \tag{20}$$
$$= \Delta_1(\tilde{w}_i)\sqrt{2 I(\widehat{Z}_i; U_i | \widetilde{W}_i = \tilde{w}_i)}.$$

Recall Eq. (14) in Lemma A.1 and by Jensen's inequality for the absolute function, the first bound is obtained:

$$|\mathcal{E}_\mu(\mathcal{A})| \leq \frac{1}{n}\sum_{i=1}^n \mathbb{E}_{\widetilde{W}_i}\left[\Delta_1(\widetilde{W}_i)\sqrt{2 I^{\widetilde{W}_i}(\widehat{Z}_i; U_i)}\right]. \tag{21}$$

To prove the second bound, we return to Eq. (20), and take expectation over $\widetilde{W}_i$ first. By Jensen's inequality,

$$\mathbb{E}_{\widehat{Z}_i, U_i, \widetilde{W}_i}\left[g(\tilde{W}_i, \widehat{Z}_i, U_i)\right] \leq \inf_{t>0} \frac{I(\widehat{Z}_i; U_i | \widetilde{W}_i) + \frac{t^2 \mathbb{E}_{\widetilde{W}_i}\left[\Delta(\widetilde{W}_i)^2\right]}{2}}{t} \tag{22}$$
$$= \sqrt{2\mathbb{E}_{\widetilde{W}_i}\left[\Delta(\widetilde{W}_i)^2\right] I(\widehat{Z}_i; U_i | \widetilde{W}_i)}.$$

Therefore, we have the second bound as below

$$|\mathcal{E}_\mu(\mathcal{A})| \leq \frac{1}{n}\sum_{i=1}^n \sqrt{2\mathbb{E}_{\widetilde{W}_i}\left[\Delta(\widetilde{W}_i)^2\right] I(\widehat{Z}_i; U_i | \widetilde{W}_i)}. \tag{23}$$

For the second part of Theorem 4.1, notice that it's valid to let $\gamma_3 = \mathbb{E}_{\widetilde{W}_i}\left[\Delta(\widetilde{W}_i)\right]$, then recall Eq. (21),

$$|\mathcal{E}_\mu(\mathcal{A})| \leq \frac{1}{n}\sum_{i=1}^n \mathbb{E}_{\widetilde{W}_i}\left[\Delta(\widetilde{W}_i)\sqrt{2 I^{\widetilde{W}_i}(\widehat{Z}_i; U_i)}\right] \leq \frac{\sqrt{2}\gamma_3}{n}\sum_{i=1}^n \sqrt{\sup_{\tilde{w}_i \in (\mathcal{W}_s)^2_{s \in \mathcal{Z}^n}} I^{\tilde{w}_i}(\widehat{Z}_i; U_i)}.$$

This completes the proof. $\qquad\square$

## D.2 Proof of Theorem 4.2

*Proof.* The proof is similar to [18, Theorem 2.1]. By the chain rule,

$$I(\widehat{Z}_i; U_i, \widetilde{W}_i) = I(\widehat{Z}_i; U_i | \widetilde{W}_i) + I(\widehat{Z}_i; \widetilde{W}_i). \tag{24}$$

Since $H(\widehat{Z}_i | U_i, \widetilde{W}_i) = H(\widehat{Z}_i | W_i, U_i, \widetilde{W}_i) = H(\widehat{Z}_i | W_i)$, we have $I(\widehat{Z}_i; U_i, \widetilde{W}_i) = H(\widehat{Z}_i) - H(\widehat{Z}_i | U_i, \widetilde{W}_i) = H(\widehat{Z}_i) - H(\widehat{Z}_i | W_i) = I(\widehat{Z}_i; W_i)$. Thus, $I(\widehat{Z}_i; U_i, \widetilde{W}_i) = I(\widehat{Z}_i; W_i)$. Recall Eq. (24) and by the non-negativity of mutual information, we have $I(\widehat{Z}_i; U_i | \widetilde{W}_i) \leq I(W_i; \widehat{Z}_i)$. Note that $I(W_i; \widehat{Z}_i) = I(\widetilde{W}_i^+; \widetilde{Z}_i^+) = I(W; Z_i)$. This completes the proof. $\qquad\square$

### D.3 Proof of Theorem 4.3

*Proof.* We first return to Eq. (19) in the previous proof, and we have already known that $g(\tilde{w}_i, \widehat{Z}_i, U'_i)$ is a zero-mean random variable bounded in $[-\Delta_1(\tilde{w}_i), \Delta_1(\tilde{w}_i)]$ for a fixed $\tilde{w}_i$.

By Lemma A.5, we have

$$
\begin{aligned}
\log \mathbb{E}_{\widehat{Z}_i, U'_i | \tilde{w}_i} \left[ e^{tg(\tilde{w}_i, \widehat{Z}_i, U'_i)} \right] \leq & h\left(\Delta_1(\tilde{w}_i)t\right) t^2 \mathbb{E}_{\widehat{Z}_i, U'_i | \tilde{w}_i} \left[ g(\tilde{w}_i, \widehat{Z}_i, U'_i)^2 \right] \\
= & h\left(\Delta_1(\tilde{w}_i)t\right) t^2 \mathbb{E}_{\widehat{Z}_i | \tilde{w}_i} \left[ \left( \ell(\tilde{w}_i^-, \widehat{Z}_i) - \ell(\tilde{w}_i^+, \widehat{Z}_i) \right)^2 \right].
\end{aligned}
$$

Plugging the above into Eq. (19),

$$
\mathbb{E}_{\widehat{Z}_i, U_i | \tilde{w}_i} \left[ g(\tilde{w}_i, \widehat{Z}_i, U_i) \right] \leq \inf_{t>0} \frac{I(\widehat{Z}_i; U_i | \widetilde{W}_i = \tilde{w}_i)}{t} + h\left(\Delta_1(\tilde{w}_i)t\right) t \mathbb{E}_{\widehat{Z}_i | \tilde{w}_i} \left[ \left( \ell(\tilde{w}_i^-, \widehat{Z}_i) - \ell(\tilde{w}_i^+, \widehat{Z}_i) \right)^2 \right]. \tag{25}
$$

Let $t = \frac{1}{\Delta_1(\tilde{w}_i)}$, we have

$$
\mathbb{E}_{\widehat{Z}_i, U_i | \tilde{w}_i} \left[ g(\tilde{w}_i, \widehat{Z}_i, U_i) \right] \leq \Delta_1(\tilde{w}_i) I(\widehat{Z}_i; U_i | \widetilde{W}_i = \tilde{w}_i) + 0.72 \frac{\mathbb{E}_{\widehat{Z}_i | \tilde{w}_i} \left[ \left( \ell(\tilde{w}_i^-, \widehat{Z}_i) - \ell(\tilde{w}_i^+, \widehat{Z}_i) \right)^2 \right]}{\Delta_1(\tilde{w}_i)}.
$$

Let $\Lambda(\tilde{w}_i) = \mathbb{E}_{\widehat{Z}_i | \tilde{w}_i} \left[ \left( \ell(\tilde{w}_i^-, \widehat{Z}_i) - \ell(\tilde{w}_i^+, \widehat{Z}_i) \right)^2 \right] \Big/ \Delta_1(\tilde{w}_i)^2$, then

$$
|\mathcal{E}_\mu(\mathcal{A})| \leq \frac{1}{n} \sum_{i=1}^n \mathbb{E}_{\widetilde{W}_i} \left[ \Delta_1(\widetilde{W}_i) \left( I^{\widetilde{W}_i}(\widehat{Z}_i; U_i) + 0.72 \Lambda(\widetilde{W}_i) \right) \right].
$$

For the second part, if $\mathcal{A}$ is further $\beta_2$-uniform stable, recall Eq. (25) and by the non-decreasing property of $h$, we have

$$
\mathbb{E}_{\widehat{Z}_i, U_i | \tilde{w}_i} \left[ g(\tilde{w}_i, \widehat{Z}_i, U_i) \right] \leq \inf_{t>0} \frac{I(\widehat{Z}_i; U_i | \widetilde{W}_i = \tilde{w}_i)}{t} + h(\beta_2 t) t \mathbb{E}_{\widehat{Z}_i | \tilde{w}_i} \left[ \left( \ell(\tilde{w}_i^-, \widehat{Z}_i) - \ell(\tilde{w}_i^+, \widehat{Z}_i) \right)^2 \right].
$$

Let $t = \frac{1}{\beta_2}$ and taking expectation over $\widetilde{W}_i$, we have

$$
\begin{aligned}
\mathbb{E}_{\widehat{Z}_i, U_i, \widetilde{W}_i} \left[ g(\widetilde{W}_i, \widehat{Z}_i, U_i) \right] \leq & \beta_2 I(\widehat{Z}_i; U_i | \widetilde{W}_i) + 0.72 \frac{\mathbb{E}_{\widehat{Z}_i, \widetilde{W}_i} \left[ \left( \ell(\widetilde{W}_i^-, \widehat{Z}_i) - \ell(\widetilde{W}_i^+, \widehat{Z}_i) \right)^2 \right]}{\beta_2} \\
= & \beta_2 I(\widehat{Z}_i; U_i | \widetilde{W}_i) + 0.72 \frac{\gamma_4^2}{\beta_2},
\end{aligned}
$$

where the equality is by the definition of $\gamma_4$-SCH-D stability.

Thus,

$$
|\mathcal{E}_\mu(\mathcal{A})| \leq \frac{1}{n} \sum_{i=1}^n \beta_2 I(\widehat{Z}_i; U_i | \widetilde{W}_i) + 0.72 \frac{\gamma_4^2}{\beta_2}.
$$

This concludes the proof. $\qquad \square$

### D.4 Proof of Theorem 4.4

We present a stronger version of Theorem 4.4.

**Theorem D.1.** *Under the same conditions in Theorem 4.1, and we further assume that $\mathcal{A}$ is $\gamma_2$-SCH-B stable and symmetric with respect to $S$, i.e. it does not depend on the order of the elements in the training sample. Let $\bar{\Delta}_1(\widetilde{W}) = \frac{1}{n} \sum_{i=1}^n \Delta_1(\widetilde{W}_i)^2$, we have*

$$
\mathbb{E}_{W,S} \left[ (L_S(W) - L_\mu(W))^2 \right] \leq \frac{6}{n} \mathbb{E}_{\widetilde{W}} \left[ \bar{\Delta}_1(\widetilde{W}) \left( I^{\widetilde{W}}(E; U) + \frac{\log 3}{2} \right) \right] + \frac{1}{n} + 4\gamma_2^2.
$$

Then Theorem 4.4 is a corollary of Theorem D.1.

*Proof of Theorem 4.4.* For $\beta_2$-uniform stable algorithm, by $\bar{\Delta}_1(\widetilde{W}) \leq \beta_2^2$ and $\gamma_2^2 \leq \beta_2^2$, we have

$$\mathbb{E}_{W,S}\left[(L_S(W) - L_\mu(W))^2\right] \leq \frac{6\beta_2^2}{n}\left(I(E;U|\widetilde{W}) + \frac{\log 3}{2}\right) + \frac{1}{n} + 4\beta_2^2$$

$$= 4\beta_2^2\left(\frac{1.5 I(E;U|\widetilde{W}) + 0.82}{n} + 1\right) + \frac{1}{n}.$$

This completes the proof. $\square$

Before we prove Theorem D.1, we need to first obtain the following lemma.

**Lemma D.1.** *Under the same conditions in Theorem 4.1, let* $\bar{\Delta}_1(\widetilde{W}) = \frac{1}{n}\sum_{i=1}^n \Delta_1(\widetilde{W}_i)^2$, *we have*

$$\mathbb{E}_{W,S}\left[\left(\frac{1}{n}\sum_{i=1}^n \mathbb{E}_{W^i|W}\left[\ell(W^i, Z_i)\right] - L_S(W)\right)^2\right] \leq \frac{3}{n}\mathbb{E}_{\widetilde{W}}\left[\bar{\Delta}_1(\widetilde{W})\left(I^{\widetilde{W}}(E;U) + \frac{\log 3}{2}\right)\right].$$

*Proof of Lemma D.1.* Here we borrow some proof techniques used in [58, Thm. 2].

Let $g(\tilde{w}, \hat{z}_i, u_i) = (-1)^{u_i}\left(\ell(\tilde{w}_i^-, \hat{z}_i) - \ell(\tilde{w}_i^+, \hat{z}_i)\right)$ and let $G \sim \mathcal{N}(0,1)$ be an independent standard Gaussian random variable. Let $f = t \cdot (\frac{1}{n}\sum_{i=1}^n g)^2$ in Lemma A.2, then

$$\mathbb{E}_{E,U|\tilde{w}}\left[\left(\frac{1}{n}\sum_{i=1}^n g(\tilde{w}, \widehat{Z}_i, U_i)\right)^2\right] \leq \inf_{t>0}\frac{I(E;U|\widetilde{W}=\tilde{w}) + \log \mathbb{E}_{E,U'|\tilde{w}}\left[e^{t\left(\frac{1}{n}\sum_{i=1}^n g(\tilde{w}, \widehat{Z}_i, U_i')\right)^2}\right]}{t}$$

$$= \inf_{t>0}\frac{I(E;U|\widetilde{W}=\tilde{w}) + \log \mathbb{E}_{E,U'|\tilde{w}}\left[\mathbb{E}_G\left[e^{\frac{G\sqrt{2t}}{n}\sum_{i=1}^n g(\tilde{w}, \widehat{Z}_i, U_i')}\right]\right]}{t} \tag{26}$$

$$= \inf_{t>0}\frac{I(E;U|\widetilde{W}=\tilde{w}) + \log \mathbb{E}_{G,E|\tilde{w}}\left[\prod_{i=1}^n \mathbb{E}_{U_i'}\left[e^{\frac{G\sqrt{2t}}{n}g(\tilde{w}, \widehat{Z}_i, U_i')}\right]\right]}{t}$$

$$\leq \inf_{t>0}\frac{I(E;U|\widetilde{W}=\tilde{w}) + \log \mathbb{E}_G\left[e^{\frac{G^2 t \sum_{i=1}^n \Delta_1(\tilde{w}_i)^2}{n^2}}\right]}{t} \tag{27}$$

$$\leq \inf_{t\in\left(0, \frac{n^2}{2\sum_{i=1}^n \Delta_1(\tilde{w}_i)^2}\right)}\frac{I(E;U|\widetilde{W}=\tilde{w}) + \log\left(1\Big/\sqrt{1 - \frac{2t\sum_{i=1}^n \Delta_1(\tilde{w}_i)^2}{n^2}}\right)}{t}$$

$$\tag{28}$$

$$= \inf_{t\in\left(0, \frac{n^2}{2\sum_{i=1}^n \Delta_1(\tilde{w}_i)^2}\right)}\frac{I(E;U|\widetilde{W}=\tilde{w}) - \frac{1}{2}\log\left(1 - \frac{2t\sum_{i=1}^n \Delta_1(\tilde{w}_i)^2}{n^2}\right)}{t},$$

where Eq. (26) is by the moment generating function of Gaussian distribution: $\mathbb{E}_G\left[e^{\lambda G}\right] = e^{\frac{\lambda^2}{2}}$ for all $\lambda \in \mathbb{R}$, Eq. (27) is by Lemma A.3 and Eq. (28) is by the moment generating function of chi-squared distribution: $\mathbb{E}_G\left[e^{\lambda G^2}\right] \leq \frac{1}{\sqrt{1-2\lambda}}$ for $\lambda < \frac{1}{2}$.

Let $t = \frac{n^2}{3\sum_{i=1}^n \Delta_1(\tilde{w}_i)^2}$ be substituted to the last equation above, we have

$$\mathbb{E}_{E,U|\tilde{w}}\left[\left(\frac{1}{n}\sum_{i=1}^n g(\tilde{w}, \widehat{Z}_i, U_i)\right)^2\right] \leq \frac{3}{n^2}\sum_{i=1}^n \Delta_1(\tilde{w}_i)^2\left(I(E;U|\widetilde{W}=\tilde{w}) + \frac{\log 3}{2}\right). \tag{29}$$

Let $\bar{\Delta}_1(\tilde{w}) = \frac{1}{n}\sum_{i=1}^{n}\Delta_1(\tilde{w}_i)^2$, and taking expectation over $\widetilde{W}$ for both sides,

$$\mathbb{E}_{E,U,\widetilde{W}}\left[\left(\frac{1}{n}\sum_{i=1}^{n}g(\widetilde{W},\widehat{Z}_i,U_i)\right)^2\right] \leq \frac{3}{n}\mathbb{E}_{\widetilde{W}}\left[\bar{\Delta}_1(\widetilde{W})\left(I^{\widetilde{W}}(E;U) + \frac{\log 3}{2}\right)\right]. \qquad (30)$$

Applying Jensen's inequality to the square function, we have

$$\mathbb{E}_{W,S}\left[\left(\frac{1}{n}\sum_{i=1}^{n}\mathbb{E}_{W^i|W}\left[\ell(W^i,Z_i)\right] - L_S(W)\right)^2\right] \leq \mathbb{E}_{E,U,\widetilde{W}}\left[\left(\frac{1}{n}\sum_{i=1}^{n}g(\widetilde{W},\widehat{Z}_i,U_i)\right)^2\right].$$

Combining Eq. (30) with the inequality above will concludes the proof. $\qquad\square$

We are now in a position to prove Theorem D.1.

*Proof of Theorem D.1.*

$$\mathbb{E}_{W,S}\left[(L_S(W) - L_\mu(W))^2\right]$$

$$=\mathbb{E}_{W,S}\left[\left(\frac{1}{n}\sum_{i=1}^{n}\ell(W,Z_i) - \mathbb{E}_{Z'}\left[\ell(W,Z')\right]\right)^2\right]$$

$$=\mathbb{E}_{W,S}\left[\left(\frac{1}{n}\sum_{i=1}^{n}\ell(W,Z_i) - \frac{1}{n}\sum_{i=1}^{n}\mathbb{E}_{W^i|W}\left[\ell(W^i,Z_i)\right] + \frac{1}{n}\sum_{i=1}^{n}\mathbb{E}_{W^i|W}\left[\ell(W^i,Z_i)\right] - \mathbb{E}_{Z'}\left[\ell(W,Z')\right]\right)^2\right]$$

$$\leq 2\,\underbrace{\mathbb{E}_{W,S}\left[\left(\frac{1}{n}\sum_{i=1}^{n}\ell(W,Z_i) - \frac{1}{n}\sum_{i=1}^{n}\mathbb{E}_{W^i|W}\left[\ell(W^i,Z_i)\right]\right)^2\right]}_{B_1}$$

$$+ 2\,\underbrace{\mathbb{E}_{W,S}\left[\left(\frac{1}{n}\sum_{i=1}^{n}\mathbb{E}_{W^i|W}\left[\ell(W^i,Z_i)\right] - \mathbb{E}_{Z'}\left[\ell(W,Z')\right]\right)^2\right]}_{B_2},$$

where the last inequality is by $(x+y)^2 \leq 2x^2 + 2y^2$. Notice that $B_1$ can be bounded by using Lemma D.1. We now focus on $B_2$. Since $\mathbb{E}_{Z'}\left[\ell(W,Z')\right] = \frac{1}{n}\sum_{i=1}^{n}\mathbb{E}_{Z'}\left[\ell(W,Z')\right]$, we have

$$B_2 = \mathbb{E}_{W,S}\left[\left(\frac{1}{n}\sum_{i=1}^{n}\mathbb{E}_{W^i|W}\left[\ell(W^i,Z_i)\right] - \frac{1}{n}\sum_{i=1}^{n}\mathbb{E}_{Z'}\left[\ell(W,Z') - \mathbb{E}_{W^i|W}\left[\ell(W^i,Z')\right] + \mathbb{E}_{W^i|W}\left[\ell(W^i,Z')\right]\right]\right)^2\right]$$

$$=\mathbb{E}_{W,S}\left[\left(\frac{1}{n}\sum_{i=1}^{n}\mathbb{E}_{W^i|W}\left[\ell(W^i,Z_i) - \mathbb{E}_{Z'}\left[\ell(W^i,Z')\right]\right] - \frac{1}{n}\sum_{i=1}^{n}\mathbb{E}_{Z'}\left[\ell(W,Z') - \mathbb{E}_{W^i|W}\left[\ell(W^i,Z')\right]\right]\right)^2\right]$$

$$\leq 2\,\underbrace{\mathbb{E}_{W,S}\left[\left(\frac{1}{n}\sum_{i=1}^{n}\mathbb{E}_{W^i|W}\left[\ell(W^i,Z_i) - \mathbb{E}_{Z'}\left[\ell(W^i,Z')\right]\right]\right)^2\right]}_{B_3}$$

$$+ 2\,\underbrace{\mathbb{E}_{W}\left[\left(\frac{1}{n}\sum_{i=1}^{n}\mathbb{E}_{Z'}\left[\ell(W,Z') - \mathbb{E}_{W^i|W}\left[\ell(W^i,Z')\right]\right]\right)^2\right]}_{B_4}.$$

For $B_3$, we apply Jensen's inequality to move the expectation over $W^i$ outside of the square function,

$$B_3 \leq \mathbb{E}_{W^1, W^2, \ldots, W^n, S} \left[ \left( \frac{1}{n} \sum_{i=1}^{n} \ell(W^i, Z_i) - \mathbb{E}_{Z'} \left[ \ell(W^i, Z') \right] \right)^2 \right]$$

$$= \mathbb{E}_{S', S, R} \left[ \left( \frac{1}{n} \sum_{i=1}^{n} \ell(\mathcal{A}(S^i, R), Z_i) - \mathbb{E}_{Z'} \left[ \ell(\mathcal{A}(S^i, R), Z') \right] \right)^2 \right].$$

Notice that $S'$, $S$ and $R$ are all independent with each other (so $W^i$, $Z_i$ and $Z_i'$ are also independent with each other). If we further let $\mathcal{A}$ be symmetric, namely $W^i$ does not dependent on $i$, then the inequality above is equivalent to

$$B_3 \leq \mathbb{E}_{W, S'} \left[ \left( \frac{1}{n} \sum_{i=1}^{n} \ell(W, Z_i') - \mathbb{E}_{Z'} \left[ \ell(W, Z') \right] \right)^2 \right]$$

$$= \mathbb{E}_W \left[ \mathbb{E}_{S'} \left[ \left( \frac{1}{n} \sum_{i=1}^{n} \ell(W, Z_i') - \mathbb{E}_{Z'} \left[ \ell(W, Z') \right] \right)^2 \right] \right].$$

Hence, the inner expectation in the RHS above is just the variance of the sample mean of $n$ i.i.d bounded random variables. Recall that $\ell(\cdot, \cdot) \in [0, 1]$, thereby

$$B_3 \leq \mathbb{E}_W \left[ \frac{\mathrm{Var}_{Z'}(\ell(W, Z'))}{n} \right] \leq \frac{1}{4n},$$

where the second inequality is by Lemma A.4.

Then, for $B_4$, we also apply Jensen's inequality to the square function, and by the definition of $\gamma_2$-SCH-B stability, we have

$$B_4 \leq \mathbb{E}_{W, Z'} \left[ \left( \frac{1}{n} \sum_{i=1}^{n} \ell(W, Z') - \mathbb{E}_{W^i|W} \left[ \ell(W^i, Z') \right] \right)^2 \right]$$

$$\leq \frac{1}{n} \sum_{i=1}^{n} \mathbb{E}_{W, Z'} \left[ \left( \ell(W, Z') - \mathbb{E}_{W^i|W} \left[ \ell(W^i, Z') \right] \right)^2 \right] \leq \gamma_2^2.$$

Putting everthing together, we have

$$\begin{aligned}
\mathbb{E}_{W, S} \left[ (L_S(W) - L_\mu(W))^2 \right] &\leq 2B_1 + 2B_2 \\
&\leq 2B_1 + 4B_3 + 4B_4 \\
&\leq 2B_1 + \frac{1}{n} + 4\gamma_2 \\
&\leq \frac{6}{n} \mathbb{E}_{\widetilde{W}} \left[ \bar{\Delta}_1(\widetilde{W}) \left( I^{\widetilde{W}}(E; U) + \frac{\log 3}{2} \right) \right] + \frac{1}{n} + 4\gamma_2^2,
\end{aligned}$$

where the last inequality is by Lemma D.1. This completes the proof. $\qquad \square$

## D.5 Proof of Theorem 4.5

*Proof.* Let $g(\tilde{z}_i^+, w_i, \bar{w}_i, u_i) = (-1)^{u_i} \left( \ell(\bar{w}_i, \tilde{z}_i^+) - \ell(w_i, \tilde{z}_i^+) \right)$. Again, by Lemma A.2, we have

$$\mathbb{E}_{W_i, \overline{W}_i, U_i | \tilde{z}_i^+} \left[ g(\tilde{z}_i^+, W_i, \overline{W}_i, U_i) \right] \leq \inf_{t > 0} \frac{I(W_i, \overline{W}_i; U_i | \widetilde{Z}_i^+ = \tilde{z}_i^+) + \log \mathbb{E}_{W_i, \overline{W}_i, U_i' | \tilde{z}_i^+} \left[ e^{t g(\tilde{z}_i^+, W_i, \overline{W}_i, U_i')} \right]}{t}.$$

$$(31)$$

Similar to the previous proofs, it's easy to see that $g(\tilde{z}_i^+, W_i, \overline{W}_i, U_i')$ is a zero-mean random variable bounded in $[-\Delta_2(\tilde{z}_i^+), \Delta_2(\tilde{z}_i^+)]$. Thus,

$$\mathbb{E}_{W_i,\overline{W}_i,U_i|\tilde{z}_i^+}\left[g(\tilde{z}_i^+, W_i, \overline{W}_i, U_i)\right] \leq \inf_{t>0} \frac{I(W_i,\overline{W}_i;U_i|\widetilde{Z}_i^+ = \tilde{z}_i^+) + \frac{t^2 \Delta_2(\tilde{z}_i^+)^2}{2}}{t}. \tag{32}$$

To prove the first bound, we let $t = \sqrt{\frac{I(W_i,\overline{W}_i;U_i|\widetilde{Z}_i^+=\tilde{z}_i^+)}{2\Delta_2(\tilde{z}_i^+)^2}}$, then

$$\mathbb{E}_{W_i,\overline{W}_i,U_i|\tilde{z}_i^+}\left[g(\tilde{z}_i^+, W_i, \overline{W}_i, U_i)\right] \leq \Delta_2(\tilde{z}_i^+)\sqrt{2I(W_i,\overline{W}_i;U_i|\widetilde{Z}_i^+ = \tilde{z}_i^+)}.$$

Recall Eq. (14) in Lemma A.1, hence,

$$|\mathcal{E}_\mu(\mathcal{A})| \leq \frac{1}{n}\sum_{i=1}^n \mathbb{E}_{\widetilde{Z}_i^+}\left[\Delta_2(\widetilde{Z}_i^+)\sqrt{2I^{\widetilde{Z}_i^+}(W_i,\overline{W}_i;U_i)}\right]. \tag{33}$$

To prove the second bound, we take expectation over $\widetilde{Z}_i^+$ for Eq. (32),

$$\mathbb{E}_{W_i,\overline{W}_i,U_i,\widetilde{Z}_i^+}\left[g(\widetilde{Z}_i^+, W_i, \overline{W}_i, U_i)\right] \leq \inf_{t>0} \frac{I(W_i,\overline{W}_i;U_i|\widetilde{Z}_i^+) + \frac{t^2\mathbb{E}_{\widetilde{Z}_i^+}\left[\Delta_2(\widetilde{Z}_i^+)^2\right]}{2}}{t}.$$

Let $t = \sqrt{\frac{I(W_i,\overline{W}_i;U_i|\widetilde{Z}_i^+)}{2\mathbb{E}_{\widetilde{Z}_i^+}\left[\Delta_2(\widetilde{Z}_i^+)^2\right]}}$, then

$$\mathbb{E}_{W_i,\overline{W}_i,U_i,\widetilde{Z}_i^+}\left[g(\widetilde{Z}_i^+, W_i, \overline{W}_i, U_i)\right] \leq \sqrt{2\mathbb{E}_{\widetilde{Z}_i^+}\left[\Delta_2(\widetilde{Z}_i^+)^2\right]I(W_i,\overline{W}_i;U_i|\widetilde{Z}_i^+)}.$$

Ergo,

$$|\mathcal{E}_\mu(\mathcal{A})| \leq \frac{1}{n}\sum_{i=1}^n \sqrt{2\mathbb{E}_{\widetilde{Z}_i^+}\left[\Delta_2(\widetilde{Z}_i^+)^2\right]I(W_i,\overline{W}_i;U_i|\widetilde{Z}_i^+)}. \tag{34}$$

This completes the proof. $\qquad\square$

# E  Omitted Proof in Section 6

## E.1  Proof of Proposition 1

*Proof.* By Jensen's inequality and triangle inequality, for any $i \in [n]$, we have

$$\mathbb{E}_{S,R,Z'}\left[\left(\ell(W, Z') - \mathbb{E}_{W^i|W}\left[\ell(W^i, Z')\right]\right)^2\right]$$

$$\leq \mathbb{E}_{W,W^i,Z'}\left[\left(\ell(W, Z') - \ell(W^i, Z')\right)^2\right]$$

$$= \mathbb{E}_{W,W^i,Z'}\left[\left(\ell(W, Z') - \ell(w^*, Z') + \ell(w^*, Z') - \ell(W^i, Z')\right)^2\right]$$

$$\leq 2\mathbb{E}_{W,Z'}\left[\left(\ell(W, Z') - \ell(w^*, Z')\right)^2\right] + 2\mathbb{E}_{W^i,Z'}\left[\left(\ell(W^i, Z') - \ell(w^*, Z')\right)^2\right]$$

$$\leq 4B\mathbb{E}_W\left[(L_\mu(W) - L_\mu(w^*))^{\frac{1}{\kappa}}\right],$$

where the last inequality is by the definition of the Bernstein condition. This completes the proof. $\quad\square$

## E.2  Proof of Theorem 6.1

*Proof.* Let $g(\Delta\ell_i, u_i) = (-1)^{u_i}\Delta\ell_i$. Notice that $|g(\Delta L_i, U_i')| \leq \beta_2$ and $g(\Delta L_i, U_i')$ is agian zero-mean, then

$$\mathbb{E}_{\Delta L_i,U_i}\left[g(\Delta L_i, U_i)\right] \leq \frac{I(\Delta L_i;U_i) + \log\mathbb{E}_{\Delta L_i,U_i'}\left[e^{tg(\Delta L_i,U_i')}\right]}{t}$$

$$\leq \beta_2\sqrt{2I(\Delta L_i;U_i)}.$$

Thus,

$$|\mathcal{E}_\mu(\mathcal{A})| \leq \frac{\beta_2}{n} \sum_{i=1}^n \sqrt{2I(\Delta L_i; U_i)}.$$

To prove the disintegrated CMI bound, we let $g$ be defined in the same way, and the remaining development is the same with the proof in Theorem 4.5.

For the second inequality, notice that $I(\Delta L_i; U_i) \leq I(\Delta L_i; U_i | \widetilde{Z}_i^+)$ by using the chain rule of mutual information and the independence between $\widetilde{Z}_i^+$ and $U_i$. In addition, moving the expectation over $\widetilde{Z}_i^+$ inside the square-root function by Jensen's inequality, we have $\mathbb{E}_{\widetilde{Z}_i^+} \sqrt{I^{\widetilde{Z}_i^+}(\Delta L_i; U_i)} \leq \sqrt{I(\Delta L_i; U_i | \widetilde{Z}_i^+)}.$ □

### E.3 Proof of Theorem 6.2

*Proof.* Before we prove Theorem 6.2, we first show the following lemma.

**Lemma E.1.** *For any $i \in [n]$, we have $\sum_{i=1}^n I(F_i, \bar{F}_i; U_i | \widetilde{Z}_i^+) \leq I(F_{[n]}, \bar{F}_{[n]}; U | \widetilde{Z}_{[n]}^+).$*

*Proof of Lemma E.1.* First, by $I(F_i, \bar{F}_i; U_i | \widetilde{Z}_i^+) = H(U_i) - H(U_i | F_i, \bar{F}_i, \widetilde{Z}_i^+)$ and $I(F_i, \bar{F}_i; U_i | \widetilde{Z}_{[n]}^+) = H(U_i) - H(U_i | F_i, \bar{F}_i, \widetilde{Z}_{[n]}^+)$, and notice that $H(U_i | F_i, \bar{F}_i, \widetilde{Z}_{[n]}^+) \leq H(U_i | F_i, \bar{F}_i, \widetilde{Z}_i^+)$, we have

$$I(F_i, \bar{F}_i; U_i | \widetilde{Z}_i^+) \leq I(F_i, \bar{F}_i; U_i | \widetilde{Z}_{[n]}^+). \tag{35}$$

Then, using the chain rule,

$$I(F_i, \bar{F}_i; U_i | \widetilde{Z}_{[n]}^+) + I(F_{[n]\setminus i}, \bar{F}_{[n]\setminus i}; U_i | \widetilde{Z}_{[n]}^+, F_i, \bar{F}_i) = I(F_{[n]}, \bar{F}_{[n]}; U_i | \widetilde{Z}_{[n]}^+).$$

By the non-negativity of mutual information, we have

$$I(F_i, \bar{F}_i; U_i | \widetilde{Z}_{[n]}^+) \leq I(F_{[n]}, \bar{F}_{[n]}; U_i | \widetilde{Z}_{[n]}^+). \tag{36}$$

Furthermore, by the independence of each $U_i$ (i.e. $I(U_i; U_{[n]\setminus i} | \widetilde{Z}_{[n]}^+) = 0$), we have

$$\sum_{i=1}^n I(F_{[n]}, \bar{F}_{[n]}; U_i | \widetilde{Z}_{[n]}^+) \leq I(F_{[n]}, \bar{F}_{[n]}; U | \widetilde{Z}_{[n]}^+). \tag{37}$$

Combining Eq. (35-37) will conclude the proof. □

We now prove Theorem 6.2.

For a given $\widetilde{Z}_{[n]}$, the number of distinct values of their predictions, denoted by $k$, are upper bounded by the growth function of $\mathcal{F}$ evaluated at $n$,

$$k \leq \sum_{i=1}^d \binom{n}{i} \leq \left(\frac{en}{d}\right)^d,$$

where the second inequality is by Sauer-Shelah lemma [53, 57] for $n > d + 1$.

Thus,

$$I(F_{[n]}, \bar{F}_{[n]}; U | \widetilde{Z}_{[n]}^+) \leq H(F_{[n]}, \bar{F}_{[n]} | \widetilde{Z}_{[n]}^+) \leq H(F_{[n]} | \widetilde{Z}_{[n]}^+) + H(\bar{F}_{[n]} | \widetilde{Z}_{[n]}^+) \leq 2d \log\left(\frac{en}{d}\right). \tag{38}$$

By Jensen's inequality and Lemma E.1, we have

$$\frac{1}{n}\sum_{i=1}^{n}\sqrt{I(F_i,\bar{F}_i;U_i|\widetilde{Z}_i^+)} \leq \sqrt{\frac{1}{n}\sum_{i=1}^{n}I(F_i,\bar{F}_i;U_i|\widetilde{Z}_i^+)} \leq \sqrt{\frac{I(F_{[n]},\bar{F}_{[n]};U|\widetilde{Z}_{[n]}^+)}{n}}.$$

Plugging Eq. (38) into the inequality above,

$$\frac{1}{n}\sum_{i=1}^{n}\sqrt{I(F_i,\bar{F}_i;U_i|\widetilde{Z}_i^+)} \leq \mathcal{O}\left(\sqrt{\frac{d}{n}\log\left(\frac{n}{d}\right)}\right),$$

which completes the proof. $\square$

## F   CLB Examples

In Example 1, [21, Thm. 17] demonstrates the non-vanishing behavior of individual IOMI and e-CMI. This is primarily attributed to the dimension-dependent nature of IOMI and CMI. Specifically, there are certain dimensional settings where IOMI can grow faster than $\mathcal{O}(n)$, as shown in [21, Thm.4], and CMI approaches a certain fraction of its upper bound, as illustrated in Example 1, resulting in non-vanishing behavior. Specifically, in Example 1, [21] employs the birthday paradox [37, Sec. 5.1] problem to demonstrate that for a large value of $d$, the probability that no pair of instances in $\widetilde{Z}$ sharing the same non-zero coordinate (referred to as event $E_0$) is smaller than a constant probability (that could be independent of $n$). Particularly, it is shown that if $d \geq \frac{2n-1}{1-c^{1/(2n-1)}}$, then $P(E_0) \geq c \geq \left(1-\frac{2n-1}{d}\right)^{2n-1}$. As an example, [21] chooses $d = 2n^2$, resulting in $c \geq 0.1$.

**Failure of $I(W;Z_i)$**   Consider the case where $d = 2n^2$. For the individual CMI [49, 70], $I(W;U_i|\widetilde{Z}_i)$, we have the following:

$$I(W;U_i|\widetilde{Z}_i) = \log 2 - H(U_i|W,\widetilde{Z}_i) \geq 0.1 \cdot \log 2.$$

This inequality holds because when event $E_0$ does not occur, one can determine the value of $U_i$ completely, as the returned hypothesis is a weighted sum of the sample. In other words, examining the non-zero coordinates of $W$ is sufficient to determine $U_i$. For an in-depth derivation of this inequality, readers are referred to the updated version of [21], where their corrected proof involves Fano's inequality. Furthermore, using the relation $I(W;Z_i) \geq I(W;U_i|\widetilde{Z}_i)$ [70], we conclude that $I(W;Z_i) \in \Omega(1)$.

**Failure of $I(\widehat{Z}_i;U_i|\widetilde{W}_i)$**   Notably, our hypotheses-conditioned CMI also does not vanish for the same reason. More precisely, when $\widetilde{W}_i$ and $\widehat{Z}_i$ are given, there exists a constant probability (independent of $n$) that allows us to fully determine the returned hypothesis based on $\widehat{Z}_i$, thereby determining the value of $U_i$.

**Failure of $I(\Delta L_i;U_i)$**   Furthermore, even the loss difference based CMI (e.g., as shown in Theorem 6.1), which provides the tightest CMI measure, still does not vanish. This is attributed to the fact that if the hypothesis $W$ is independent of certain $Z$, there exists a constant probability where the loss becomes zero (recall that the loss is the negative inner product of $W$ and $Z$). Consequently, one can determine the value of $U_i$ by observing the sign of the random variable $\Delta L_i$. This also indicates the limitations of e-CMI and $f$-CMI in capturing the generalization behavior for Example 1.

In Example 2, following the approach in [21], the training sample $S = \{Z_i\}_{i=1}^{n} \sim \mu^n$ can be represented as $S = \frac{z_0}{R_0}(\varepsilon_1,\varepsilon_2,\ldots,\varepsilon_n)$, where $\{\varepsilon_i\}_{i=1}^{n}$ is a sequence of independent Rademacher random variables, i.e., $\varepsilon_i \sim \text{Unif}(\{-1,1\})$. The empirical risk is given by $L_S(W) = -\frac{L}{nR_0}\langle W,\sum_{i=1}^{n}\varepsilon_i\rangle$. In this case, the ERM solution is $W_{ERM} = z_0$ if $\text{sign}(\sum_{i=1}^{n}\varepsilon_i) = 1$, and $W_{ERM} = -z_0$ if $\text{sign}(\sum_{i=1}^{n}\varepsilon_i) = -1$. It is clear that

$$\sup_{w,w^i,z}\left|\ell(w,z)-\ell(w^i,z)\right| \leq \sup_{w,w^i,z} L\|w-w^i\| \leq 2LR_0.$$

Hence, we observe that $\beta_2$ is now a constant, whereas IOMI has an upper bound: $\sum_{i=1}^{n} I(W; Z_i) \leq I(W; S) \leq I(W; \mathrm{sign}(\sum_{i=1}^{n} \varepsilon_i)) \leq H(\mathrm{sign}(\sum_{i=1}^{n} \varepsilon_i)) \leq 1$, where the second inequality follows from the Markov chain $S - \mathrm{sign}(\sum_{i=1}^{n} \varepsilon_i) - W$. This provides us with a generalization bound of $\frac{2LR_0}{\sqrt{n}}$. Meanwhile, the actual generalization error satisfies $\mathcal{E}_\mu(\mathcal{A}) \geq \frac{LR_0}{\sqrt{2n}}$ (see [21, AppendixB] for a derivation). Thus, the IOMI bound is tight up to a constant, and the stability bound $\beta_2$ itself is vacuous. It is worth noting that $I(\widehat{Z}_i; U_i | \widetilde{W}_i) \leq I(W; Z_i) \leq 1$ by Theorem4.2, indicating that the CMI bound is also tight.

We would like to note that the failures of chained mutual information bounds [2] are not demonstrated in the counterexamples presented in [21]. Notably, when the hypothesis is quantized, it becomes more challenging to guess $U_i$ or $Z_i$. Therefore, exploring the potential of chained information-theoretic bounds, which do not necessarily rely on stability notions, could be another avenue to explain the generalization behavior observed in these counterexamples.

# G  Additional Applications

## G.1  Compression Schemes

We now consider the algorithm that has a compression scheme [30]. Formally, a sample compression scheme of size $k \in \mathbb{N}$ is a pair of maps $(\mathcal{A}_1, \mathcal{A}_2)$. Specifically, for all samples $s$ with $n > k$, $\mathcal{A}_1 : \mathcal{Z}^n \to \mathcal{Z}^k$ compresses the sample into a length-$k$ subsequence $\mathcal{A}_1(s) \subseteq s$. Then $\mathcal{A}_2 : \mathcal{Z}^k \to \mathcal{W}$ could be some arbitrary mapping. Hence, $\mathcal{A}(\cdot) = \mathcal{A}_2(\mathcal{A}_1(\cdot))$. Let $K$ be the index set for $S$ selected by $\mathcal{A}_1$, and let $\overline{K}$ be the selected index set for $S^i$. In this case, our supersample-conditioned CMI has an upper bound: $I(W_i, \overline{W}_i; U_i | \widetilde{Z}_i^+) \leq I(K, \overline{K}; U_i | \widetilde{Z}_i^+) \leq H(K, \overline{K} | \widetilde{Z}_i^+) \leq 2 \log \binom{n}{k} \leq 2k \log n$. Then, if $\mathcal{A}$ is further $\beta_2$-uniform stable, then we have the generalization bound $\mathcal{E}_\mu(\mathcal{A}) \leq \mathcal{O}(\beta_2 \sqrt{k \log n})$. If $\beta_2 < \mathcal{O}(1/\sqrt{n})$, this bound improves the bound in [58]. It is unclear if we can obtain any improved bound for *stable* compression schemes [7], in which case [19] provides an optimal bound that removing the $\log n$ factor for the realizable setting. A main difficulty is that an interpolating algorithm is usually unstable due to the fitting-stability tradeoff [54, Sec. 13.4].

## G.2  Distillation Algorithm

The high-probability generalization property of distillation algorithm is studied in [16]. In the first training stage of distillation, we obtain a $w_s^*$ from a highly complex hypothesis space $\mathcal{W}_1$ based on a training sample $s$. Same to [16], we assume that the first learning stage is $\alpha$-sensitive, namely $||w_s^* - w_{s^i}^*|| \leq \alpha = \mathcal{O}(1/n)$. In the second stage, the algorithm $\mathcal{A}$ will select a hypothesis that is $\lambda$-close to $w_s^*$ from a less complex hypothesis space $\mathcal{W}_2 = \{w \in \mathcal{W} : ||w - w_s^*||_\infty \leq \lambda\}$. Let the loss function $\ell$ be $L$-Lipschitz with respect to the first argument. Consequently, $\gamma_3 \leq L||w_s^* - w_{s^i}^*|| \leq L\alpha$. Then, by Theorem 4.1, we have $\mathcal{E}_\mu(\mathcal{A}) \leq L\alpha \frac{1}{n} \sum_{i=1}^{n} \sqrt{2H(U_i)} = \sqrt{2 \log 2} L\alpha$. Notice that the loss here may not necessarily be bounded or sub-Gaussian, rendering previous bounds inapplicable.

## G.3  Regularized Empirical Risk Minimization

Regularized Empirical Risk Minimization (ERM) learning rules involve minimizing the empirical risk and a regularization function jointly: $\arg \min_{w \in \mathcal{W}} L_S(w) + f_{\mathrm{reg}}(w)$, where $f_{\mathrm{reg}} : \mathcal{W} \to \mathbb{R}$. Here we specifically consider Tikhonov regularization [54], namely $f_{\mathrm{reg}}(w) = \lambda ||w||^2$, where $\lambda > 0$ is a tradeoff coefficient. The regularized ERM algorithm $\mathcal{A}$ aims to find

$$w = \arg \min_{w \in \mathcal{W}} L_S(w) + \lambda ||w||^2.$$

This regularization term ensures strong convexity of the training objective. Based on Theorem 4.3, we can derive the following results.

**Corollary G.1.** *Assume that the loss function $\ell$ is convex and L-Lipschitz. Then, for the regularized ERM algorithm with Tikhonov regularization, we have*

$$|\mathcal{E}_\mu(\mathcal{A})| \leq \frac{2L^2}{\lambda n} \left( \frac{1}{n} \sum_{i=1}^{n} I(\widehat{Z}; U_i | \widetilde{W}_i) + 0.72 \right).$$

*Proof of Corollary G.1.* By invoking [54, Corollary 13.6], we know that $\gamma_4 \leq \beta_2 = \frac{2L^2}{\lambda n}$. Plugging the value of $\beta_2$ will give us the desired result. $\qquad\square$

**Corollary G.2.** *Assume that the loss function is $\rho$-smooth and nonnegative. Let $\lambda \geq \frac{2\rho}{n}$. Then, for the regularized ERM algorithm with Tikhonov regularization, we have*

$$|\mathcal{E}_\mu(\mathcal{A})| \leq \frac{48\rho \hat{L}_n}{\lambda n} \left( \frac{1}{n} \sum_{i=1}^n I(\widehat{Z}; U_i | \widetilde{W}_i) + 0.72 \right).$$

*Proof of Corollary G.2.* By invoking [54, Corollary 13.7], we know that $\gamma_4 \leq \beta_2 = \frac{48\rho \hat{L}_n}{\lambda n}$. Plugging the value of $\beta_2$ will give us the desired result. $\qquad\square$

Although these bounds do not enhance the convergence rate of $\mathcal{O}(1/n)$ in these settings, they consistently offer tighter results compared to uniform stability-based bounds if $\frac{1}{n} \sum_{i=1}^n I(\widehat{Z}; U_i | \widetilde{W}_i) \leq 0.28$. In addition, the expected empirical risk $\hat{L}_n$ appears in the bound of Corollary G.2. While $\lambda$ has a lower bound, $\hat{L}_n$ could not be arbitrarily small for the regularized ERM.

Notice that previous information-theoretic bounds could not obtain the convergence rate of $\mathcal{O}(1/n)$ as in our results unless ICMI or CMI itself decays with $\mathcal{O}(1/n)$.

# H  Additional Discussions and Open Problems

**Stochastic Gradient Descent (SGD)**   Since the influential work of [22], stability approaches have been widely employed to provide generalization guarantees for (sub)gradient-based optimization algorithms, such as SGD, under certain conditions like the convex-smooth-Lipschitz loss. More recently, [5] extended the results of [22] to the non-smooth loss function in the SCO setting.

In contrast, information-theoretic (weight/hypothesis-based) bounds are typically used to analyze the noisy version of SGD, known as SGLD [43, 40, 18, 49, 60]. Directly analyzing SGD poses challenges because the returned hypothesis $W$ contains a significant amount of information about $S$ or $Z_i$, resulting in potentially large (even infinite) mutual information. The prevalent approach to applying information-theoretic bounds to SGD is by introducing noise [41, 61], but this has been shown to yield non-vanishing bounds in [21, Thm. 4].

The combination of information-theoretic bounds with stability for analyzing the generalization of SGD presents a promising future direction. However, some potential difficulties may arise. For instance, if we continue to use the Gaussian noise perturbation method for the weight-based information-theoretic bounds, we would need to characterize the stability property for the perturbed SGD, which might require techniques employed in [38]. Additionally, when combining stability notions with loss difference based CMI (or e-CMI/$f$-CMI) bounds, as they cannot be unrolled using the chain rule and data processing inequality as in the case of weight-based IOMI/CMI bounds, it may not be possible to bound such CMI terms using trajectory-based quantities. This raises doubts about the potential for obtaining more informative generalization bounds compared to the stability-based bounds themselves.

**Generalization Bounds beyond Mutual Information**   In the information-theoretic literature, it is common to replace mutual information with alternative distributional measures, such as Wasserstein distance based bounds and total variation based bounds [50]. A promising future direction is to incorporate the stability property of algorithms into these bounds, as demonstrated in this work. It is worth noting that obtaining KL divergence-based bounds should be straightforward since they rely on the same foundational Lemma A.2 as discussed in this paper.

