# OpenReview forum: "Sample-Conditioned Hypothesis Stability Sharpens Information-Theoretic Generalization Bounds"
_NeurIPS.cc/2023/Conference — NeurIPS 2023 poster_

### Official Review · Reviewer_uPAq · 2023-06-30

**Soundness:** 3 good
**Presentation:** 2 fair
**Contribution:** 3 good
**Rating:** 6
**Confidence:** 4

**Summary:**

Information-theoretic generalization bounds offer a new approach in generalization theory by providing complexity measures that depend on the data distribution and learning algorithm itself. In the past years, it has been observed that information-theoretic generalization bounds are not compatible with the classical approaches to prove generalization such as vc theory or uniform stability.

In this paper the authors propose several new information theoretic quantities to study generalization. The proposed bounds are interesting as they are compatible with several notions of uniform stability.


**Strengths:**

I think the paper addresses an important limitation of information-theoretic generalization that is incompatibility with the uniform stability framework. They also provide various examples to show the expressiveness of their bounds.



**Weaknesses:**

1– The presentation of the paper needs significant improvement. Specifically the part on defining the additional structure is extremely vague. For instance, there are many variables that are not clearly defined. For instance, what is the tilde{W}_{i,0} and tilde{W}_{i,1}.

2– Compared to CMI bound, the intuition behind the results is not clear. For instance CMI is closely related to membership inference.

3– Some of the results in the paper seem redundant. For instance, theorem 3.1 has been proved by work by Bu et al under the name of individual sample bounds.


4– The major drawback of the bounds in the paper is the following: Uniform stability bounds for randomized algorithms are very important. For instance, in the seminal work by Hardt et al, the fact that the algorithm is randomized is important. The other example is the algorithm by Feldman and Dagan for leaning VC classes with stable algorithms. However, to me all the bounds in the paper rely on a very strong notion of stability which only holds for “deterministic algorithms. It is not clear what is the roadblock for extending the results for randomized algorithms.


Feldman, Dagan. PAC learning with stable and private predictions


5– The main limitation of uniform stability bounds is that it considers the “worst-case” data distribution. However, a learning algorithm may be more stable for “easier” distributions. However, all the bounds in the paper already depend on the worst-case stability parameter. I think this leads to suboptimal bounds.


**Questions:**

1- weakly uniform stability versus strong uniform stability: Consider the seminal work of Hardt et al paper in which they derive stability bound for SGD under the notion of weakly uniform stability. Why in the paper do authors only consider the strong notion of stability?

2- text after corollary 3.1 seems unclear to me. If the algorithm is deterministic then the mutual information term will blow up. It is not clear to me how this may result in improvement.

3- Theorem 3.3: Let's assume that the algorithm is deterministic and also permutation invariant in the sense that the order of the training points does not affect the output. Then, this results states that  gen.gap <= gamma_1  (  1  + mutual information term). It shows that even if the mutual information term goes to zero, the convergence rate is determined by gamma_1.


4- For Example 1, what is the exact generalization error? It should be of order O(1/n), however, the obtained bound using the proposed approach has an order of 1/sqrt(n).

Typos:

Lemma A.5. Bernstein.


**Limitations:**

The main limitations are explained in Items 4 and 5 of weakness.

---

> ### Author Rebuttal · Authors · 2023-08-08
>
> Thank you for your constructive comments. Our responses follow.
>
> >- The presentation of the paper needs significant improvement. Specifically the part on defining the additional structure is extremely vague. For instance, there are many variables that are not clearly defined. For instance, what is the tilde{W}{i,0} and tilde{W}{i,1}.
>
> **Response.** We apologize for lacking clarity at places and will make our best effort in the revision. Specifically, $\widetilde{W}\_{i,0}$ and $\widetilde{W}\_{i,1}$ represents the $i$-th row, the first column and the second column of the matrix $\widetilde{W}$, respectively.
>
> >- Compared to CMI bound, the intuition behind the results is not clear. For instance CMI is closely related to membership inference.
>
> **Response.** We have tried to explain the intuition of $I(\widehat{Z}_i;U_i|\widetilde{W}_i)$ in Line 212-217, and here we restate: the new CMI terms in our paper are also closely related to the membership inference. Specifically, $I(\widehat{Z}_i;U_i|\widetilde{W}_i)$ measures the ability to decide if an instance $\widehat{Z}_i$ contributes to the training of $\widetilde{W}^+$ or  to the training of $\widetilde{W}^+_i$ when we know it contributes to only one of them.
>
> >- Some of the results in the paper seem redundant. For instance, theorem 3.1 has been proved by work by Bu et al ....
>
> **Response.** Agreeably our Theorem 3.1 has a very similar form to the bound of Bu et al. However, the two bounds differ significantly due to the replacement of sub-gaussian variance proxy with stability parameter. This difference is critical since   the bound of Bu et al. has been shown inadequate in explaining the learnability of certain SCO problems [1], as illustrated by Example 1 (please refer to Section E for an explanation) in our paper. However, our Theorem 3.1 overcomes this limitation. For further comparisons between the two bounds, please see Line 191-197 in our paper.
>
> >- The major drawback of the bounds in the paper is the following: Uniform stability bounds for randomized algorithms are very important. .... However, to me all the bounds in the paper rely on a very strong notion of stability which only holds for “deterministic algorithms....
> >- weakly uniform stability versus strong uniform stability: .... Why in the paper do authors only consider the strong notion of stability?
>
> **Response.** On the one hand, our four SCH stability notions in different bounds relax the strong notion of uniform stability, taking into account the randomness of the algorithm, particularly for $\gamma_2$, $\gamma_3$, and $\gamma_4$.
>
> On the other hand, even for the strong uniform stability notion, such as the $\beta_2$-based bounds, they remain applicable to randomized algorithms. This is because the MI and CMI terms in these bounds are algorithm-dependent, capturing the randomness in the algorithm. Notably, $\beta_2$-uniform stability is still a weaker assumption than the bounded loss assumption.
>
> One important reason that we apply strong uniform stability notion here is because information-theoretic bounds are sometimes vacuous in the deterministic setting, so incorporating $\beta_2$ can significantly sharpens the information-theoretic bounds in this setting, as shown in the examples in this paper. Moreover, it's worth noting that replacing $\beta_2$ with the weak notion $\beta_1$ in our bounds may not be feasible.
>
> >- The main limitation of uniform stability bounds is that it considers the “worst-case” data distribution ....
>
> **Response.** We note that the $\gamma_2$-SCH-B stability and the $\gamma_4$-SCH-D stability are distribution-dependent notions. They are not defined from the worst-case data distribution. Additionally, if the loss function is upper-bounded by $C$, as commonly employed in proving the optimal rate of the stability-based bound, all the worst-case stability parameters can be replaced by $C$, allowing us to focus solely on the distribution-dependent notions. Remarkably, we achieve the known fastest rate of the second moment generalization error bound, as demonstrated in Theorem 4.4.
>
> >- text after corollary 3.1 seems unclear to me....
>
> **Response.** For deterministic algorithms, the individual mutual information term $I(W;Z_i)$ may not necessarily blow up. Simple examples illustrating this can be found in Example A and B in Bu et al.'s work. In fact, the primary motivation behind Bu et al.'s proposal for the individual bound is to mitigate the problem of blowing up in the sample-wise mutual information ($I(W;S)$) under the deterministic setting.
>
> >- Theorem 3.3: Let's assume that the algorithm is deterministic and also permutation invariant ....
>
> **Response.** We completely agree. While Theorem 3.3 exhibits a faster decay rate compared to Theorem 3.1 due to the removal of the square-root function, it still does not surpass $\gamma_1$ in terms of rate. We must note that our work also demonstrates the enhancement of the stability-based framework through information-theoretic analysis, as highlighted in Example 2.
>
> >- For Example 1, what is the exact generalization error? It should be of order O(1/n)...
>
> **Response.** The exact generalization error in Example 1 decays with the rate of $\mathcal{O}(1/\sqrt{n})$, as given in [1, Theorem 17].
>
> In fact, for GD with nonsmooth convex loss, such as in Example 1, [2, Theorem 3.2] gives a tight generalization lower bound: $\mathcal{O}(\eta \sqrt{T}+\frac{\eta T}{n})$. By substituting $\eta=\frac{1}{n\sqrt{n}}$ and $T=n^2$ from Example 1, we can deduce the lower bound of $\mathcal{O}(1/\sqrt{n})$ as well. Combining the upper bound obtained in our paper, we conclude that the exact generalization error has the order of $\mathcal{O}(1/\sqrt{n})$.
>
> [1] Mahdi Haghifam, et. al. Limitations of information-theoretic generalization bounds for gradient descent methods in stochastic convex optimization. ALT 2023.
>
> [2] Raef Bassily, et al. Stability of stochastic gradient descent on nonsmooth convex losses. NeurIPS 2020.

---

> > ### Comment · Reviewer_uPAq · 2023-08-16
> > **response after rebuttals**
> >
> > I would like to thank the reviewer for their comment.
> >
> > I still have questions regarding Example 1. Linear function is indeed a smooth function. So, what you have as the response is not correct. I think it is easy to find the exact dependence of the generalization bound to $n$.

---

> > > ### Author Response · Authors · 2023-08-16
> > >
> > > Thanks for your reply!
> > >
> > > >- I still have questions regarding Example 1. Linear function is indeed a smooth function. So, what you have as the response is not correct.
> > >
> > > Thanks for pointing out this, you are right, the loss in Example 1 is indeed smooth. Yet, our previous response still holds because [2, Theorem 3.2] is proved without using the additional smoothness condition of the loss, so $\mathcal{E}\_\mu(\mathcal{A})\in \Omega(1/\sqrt{n})$ holds true.
> > >
> > > While it is completely possible that [2, Theorem 3.2] is not tight for the smooth loss case, our generalization upper bounds is $\mathcal{O}(1/\sqrt{n})$. Combining them together guarantees $\mathcal{E}\_\mu(\mathcal{A})\in\mathcal{O}(1/\sqrt{n})$.

---

> > > > ### Comment · Reviewer_uPAq · 2023-08-17
> > > > **reply to authors**
> > > >
> > > > Thank you. All my questions have been addressed.
> > > >
> > > > I think it is important that the authors incorporate all the comments and suggestions by the reviewers and AC. I think these suggestion significantly improve the paper. I raise my score to 6.

---

> > > > > ### Author Response · Authors · 2023-08-17
> > > > > **Thanks**
> > > > >
> > > > > Thank you for your reply. We really appreciate the valuable discussions and constructive suggestions provided by all the reviewers and AC. These inputs have greatly contributed to improving our paper, and we will definitely include them in the revision.

---

### Official Review · Reviewer_Ptk3 · 2023-07-05

**Soundness:** 3 good
**Presentation:** 3 good
**Contribution:** 2 fair
**Rating:** 5
**Confidence:** 4

**Summary:**

The paper proposes several new stability assumptions named the sample-conditioned hypothesis (SCH) stability. Based on these notions, the authors present new IOMI and CMI bounds to address the limitations of existing information-theoretic bounds in the context of stochastic convex optimization (SCO) problems.

**Strengths:**

- The paper adopts different assumptions for bounding the CGF, which can improve existing information-theoretic bounds, especially in the context of SCO.
- The connection between Bernstein and the proposed SCH-B stability is interesting.
- The paper is generally well-written.

**Weaknesses:**

- It seems to me the most significant contribution is to introduce uniform stability for bounding the CGF, which can solve the counter-example proposed in [1]. However, it's hard to estimate the order of the stability parameter without any knowledge of the distribution. Especially in practice, such parameters do not affect the algorithm design.

- The effectiveness of the proposed SCH stabilities is unclear. It's mentioned that all the stability parameters $\gamma_{1:4}$ are smaller than $\beta_2$. From Theorem 4.3, we have $\|\mathcal{E}\_{\mu}(A)\| \leq \frac{\beta_2}{n} \sum_{i=1}^n I(\hat{Z_i};U_i|\tilde{W_i}) + 0.72 \beta_2$. Similar to the discussion in Example 2, the first term can vanish. However, the second term still exists since $\beta_2$ is a constant. To prove the effectiveness of the SCH stability bounds, it needs to identify the order of $\gamma_4^2$.

- Example 1 requires the data and parameter dimension $d=2n^2$ to ***grow with $n$*** for proving the $\Omega(1)$ lower bound of the information quantities. Does this really make sense?  Even though the uniform stability is much tighter in this case, I do not think this counter-example is representative enough. Could you conceive another counter-example that is more convincing?

- The SCH-B and SCH-D stability are distribution dependent, the corresponding stability parameter cannot be estimated, and these notions can be confused with other distribution assumptions.
- The SCH-A and SCH-C stability are distribution-free but related to the distribution (or set) of the output model parameters. It’s unclear to me how to identify the stability parameter effectively for different algorithms.

>[1] Mahdi Haghifam, Borja Rodríguez-Gálvez, Ragnar Thobaben, Mikael Skoglund, Daniel M Roy, and Gintare Karolina Dziugaite. Limitations of information-theoretic generalization bounds for gradient descent methods in stochastic convex optimization. In International Conference on Algorithmic Learning Theory, pages 663–706. PMLR, 2023.

**Questions:**

Please see the above section.

**Limitations:**

Yes.

---

> ### Author Rebuttal · Authors · 2023-08-08
>
> We thank you sincerely for your comments to our paper. Our responses follow.
>
> >- It seems to me the most significant contribution ...
>
> **Response.** Accurately estimating the order of the stability parameter is challenging in all stability-based bounds. However it is still possible to bound the parameter, without access to the underlying distribution, and obtain insights to guide the design of learning algorithms. As shown in Section F.2, without the knowledge of the data distribution, it is possible to explicitly bound the stability parameter, and the bound inspires Tikhonov regularization, which finds smoother solutions for better generalization.
>
> In addition to bounding the stability parameter, one may also have the option of estimating the parameter via influence function [1]. By employing such estimation, the appearance of the Hessian matrix of the solution can also offer valuable insights for algorithm design.
>
> We would like to re-emphasize that our primary focus is to demonstrate the existence of information-theoretic bounds capable of explaining the learnability of problems with certain stability properties, such as the SCO problems discussed in Section 5, and resolving the limitations shown in [2]. While it is not our primary goal, we agree that providing new algorithmic insights is indeed an important dimension for studying generalization. Notably novel generalization bounds may lead to the discovery of new algorithmic properties that improve generalization for the problem or algorithm is specialized to certain forms; for example, the bounds of [3] specialized to SGD leads to new insights in understanding and improving SGD in [4]. It is curious to explore this aspect for the bounds developed in this paper.
>
> >- The effectiveness of ....
>
> **Response.** If both $\beta_2$ and $\gamma_4$ are constants, then the algorithm is neither $\beta_2$-uniform stable nor $\gamma_4$-SCH-D stable. In this case, Theorem 4.3 is not of any interest for algorithms that are not stable in these senses.
>
> If $\beta_2$ is a constant, i.e., the algorithm is not $\beta_2$-stable. In this case, there is still an opportunity that the algorithm is $\gamma_4$-SCH-D stable and the second term vanishes at least at the speed of $\gamma_4$. Then we agree that identifying the order of $\gamma_4$ is needed for the bound to be useful. Similar to other distribution-dependent stability notions, determining the order of $\gamma^2_4$ is challenging. Consequently, uniform stability remains the most extensively studied and well-understood concept in the learning theory community. However, we would like to note that our SCH parameters offer an extension of the bounds' applicability, such as enabling connections with the Bernstein condition.
>
> >- Example 1 requires the data ...
>
> **Response.**  Example 1 was constructed in [2, Theorem 17]. The objective of this exercise is to create a "bad" learning problem for which information-theoretic bounds demonstrate certain limitations. Such an exercise does not require the created "bad problem" to be representative of the reality. In this paper, we show that even for such "bad problems", the presented bounds in this paper overcome these limitations.
>
> Regarding the choice of $d=2n^2$, we remark that the dimension $d$ is usually related to the model capacity, and many generalization bounds have the rate of $\mathcal{O}((\frac{d}{n})^{\alpha})$ (for some $\alpha>0$). Thus, we do care about how $d$ changes affect the generalization.  An important feature of gradient descent for SCO problems is that the sample complexity is dimension-independent. Thus, for any generalization bounds that explicitly depend on the model dimension, there may always exist a setting where they don't diminish, e.g., $d=2n^2$ or $d=3T2^n/4$ for MI/CMI bounds in [2].  In fact, this setting does make sense because we hope the generalization bounds to explain the learnability of a class of CLB problems here instead of only a specific setting with a fixed $d$. If $d$ is treated as a constant, then there would be no such limitations since the dimension-dependent property of MI/CMI bounds would not be considered. Thus, our key message here is  that when $d=2n^2$ in this CLB problem, the exact generalization error of gradient descent does vanish with $n$ increases, but  the previous information-theoretic bounds are unable to establish the learnability of this learning problem. In this sense, this example stands as a representative case.
>
> Additionally, here we focus on the regime of overparameterization where $d>n$, which holds true for the most deep neural networks. Also allowing the model dimension grows with $n$ aligns with practical scenarios. For example, achieving good performance on CIFAR-10 and ImageNet often necessitates employing a larger model for the latter.
>
> >- The SCH-B and ....
> >- The SCH-A and ....
>
> **Response.** We acknowledge that these SCH stability parameters are hard to estimate in general, which we also discussed in Section G. But this limitation, which also exists in many other bounds, does not negate the potential usefulness of these bounds, as we already discussed above. In addition, as we emphasize before, uniform stability is still the only one that has been widely studied and well understood in the learning theory community. Further development of the practical usage of stability itself may eventually also help to overcome the estimation difficulty of our bounds.
>
> [1] Pang Wei Koh, et. al. Understanding black-box predictions via influence functions. ICML 2017.
>
> [2] Mahdi Haghifam, et. al. Limitations of information-theoretic generalization bounds for gradient descent methods in stochastic convex optimization. ALT 2023.
>
> [3] Aolin Xu, et. al. Information-theoretic analysis of generalization capability of learning algorithms. NeurIPS 2017.
>
> [4] Ziqiao Wang, et. al. On the generalization of models trained with SGD: Information-theoretic bounds and implications. ICLR 2022.

---

> > ### Comment · Reviewer_Ptk3 · 2023-08-14
> > **Response**
> >
> > Thanks for your clarifications. I will take these into consideration and engage in the discussion process.

---

> > ### Comment · Reviewer_Ptk3 · 2023-08-19
> > **Follow-up question**
> >
> > Thanks for introducing the new examples. I have the following question that need more clarification.
> >
> > ### Example 1
> >
> > It seems to me the proposed method does not solve the counter-example. The original discussion in Line 280 did not consider the dimension.
> >
> > In fact, $\|\| w_t - w_t^i\|\|\leq \|\|\eta t (\hat{\mu} - \hat{\mu}^i\)\|\| = \frac{\eta t}{n} \|\|z_i - z_i^\prime\|\| \leq \mathcal{O}(\frac{\eta t \sqrt{d}}{n})$. When $d=2n^2$, $\beta_2 \leq \mathcal{O}(\frac{\eta T \sqrt{d}}{n}) \in \mathcal{O}(\sqrt{n})$.

---

> > > ### Author Response · Authors · 2023-08-19
> > >
> > > Thanks for the question. Please notice that $Z$ is a one-hot vector in Example 1. Thus, $\frac{\eta t}{n}||z_i-z_i'||\leq\frac{\eta t}{n}(||z_i||+||z_i'||)=\frac{2\eta t}{n}=\mathcal{O}(\frac{1}{\sqrt{n}})$. In addition, $w_t$ is restricted in a unit ball, the distance $||w_t-w_t^i||$ will not explode. I hope this addresses the reviewer's concern.

---

> > > > ### Comment · Reviewer_Ptk3 · 2023-08-19
> > > > **Thanks for clarification**
> > > >
> > > > Thanks for clarification. I forgot this when I resumed the details of this article.
> > > >
> > > > In my opinion, this counter-example for the SCO problem proposed by Haghifam et al. is mainly affected by inflating the MI bound with $LR$ that does not consider the distribution and specific algorithm.
> > > >
> > > > * Let us consider the MI bounds with $\sigma$-sub-Gaussian assumption, where $\ell(W, Z) - \mathbb{E} \ell(W,Z)$ is $\sigma$-sub-Gaussian under $P_{W,Z}$. Now let us identify the order of $\sigma$. For $W_t=\frac{\eta t \hat{\mu}}{\|\|\eta t \hat{\mu}\|\|}, \hat{\mu}=\frac{1}{n}\sum_{i=1}^n Z_i$, $\ell(W_t, Z)$ has $\frac{1}{n}$-bounded difference.   Then we can use the Doob martingale to prove that  $\ell(W_t, Z) - \mathbb{E} \ell(W_t,Z)$ is $\frac{1}{2\sqrt{n}}$-sub-Gaussian. So we have $\sigma \in \mathcal{O}(\frac{1}{\sqrt{n}})$, the original MI bounds can converge for this counter-example even though $I(W;S)\in\Omega(1)$.
> > > >
> > > >     -  So what are the benefits of the proposed theory compared to the original MI bounds?
> > > >
> > > > * Consider the truncated version where $w_t=\frac{\eta t \hat{\mu}}{\|\|\eta t \hat{\mu}\|\|}$, shouldn't we have $|\mathcal{E}\_{\mu}(\mathcal{A})| \leq \beta_2 \leq \mathcal{O}(\frac{1}{n})$? However, w.r.t your response to reviewer uPAq, $\mathcal{E}_{\mu}(\mathcal{A}) \in \Omega(\frac{1}{\sqrt{n}})$ holds true without using the smoothness condition.

---

> > > > > ### Author Response · Authors · 2023-08-19
> > > > >
> > > > > Thank you for the comments. Our response follows.
> > > > >
> > > > > >- Let us consider the MI bounds with... the original MI bounds can converge for this counter-example even though $I(W;S)\in\Omega(1)$
> > > > >
> > > > > **Response.** We would like to clarify that  in Example 1, Haghifam et al. first proves the individual CMI qunatity has $I(W;U_i|\widetilde{Z})\geq\Omega(1)$, and they demonstrate the failure of the individual MI quantity with a simple argument:  $I(W;U_i|\widetilde{Z})\leq I(W;Z_i)$. However, it's important to note that while $I(W;U_i|\widetilde{Z})$ is bounded by $\log{2}$, $I(W;Z_i)$ can be unbounded. Specifically, in Example 1, we have $I(W;Z_i)=H(Z_i)-H(Z_i|W)\geq\log{d}-0.9\cdot\log{d}=0.1\cdot\log{d}$. Clearly, $I(W;Z_i)$ can grow with $d$ in an unbounded manner, so there always exists a setting to let it explode, as discussed in [5], regardless of the order of the sub-Gaussian variance proxy.
> > > > >
> > > > > The remaining question is whether we can attain an improved parameter order for classical CMI. In Theorem 5.1 of [6] (the arXiv version, not their COLT version), they provide the following general result, avoiding reliance on the boundedness of the loss.
> > > > > $$
> > > > > |\mathcal{E}\_{\mu}(\mathcal{A})|\leq\sqrt{\frac{2}{n}\text{CMI}\cdot\mathbb{E}[\Delta(Z\_1,Z\_2)^2]},
> > > > > $$
> > > > > where $\Delta(Z\_1,Z\_2)\geq\sup_{w}|\ell(w,Z_1)-\ell(w,Z_2)|$. We are unable to observe how $\Delta(Z\_1,Z\_2)^2$ can vanish in Example 1. Thus, to the best of our knowlegde, the conventional forms of MI/CMI bounds cannot address Example 1. In fact, our Theorem 4.1 is highly inspired by [6, Theorem 5.1], and Corollary 5.1 is our solution for Example 1. It's worth noting that our Corollary 5.1 is just a potential solution, and might not be the only one.
> > > > >
> > > > > >- In my opinion, this counter-example for the SCO problem ... does not consider the distribution and specific algorithm.
> > > > >
> > > > > **Response.** We agree with this opinion, and this is also the motivation of our work. Selecting the suitable DV auxiliary function and bounding the CGF through the right assumption, as the reviewer did in the comments, is indeed a pivotal message conveyed by our paper (see the Introduction part). Thus, we believe that introducing other new techniques to bound CGF for classical CMI won't diminish the value of the bounds established in our paper.
> > > > >
> > > > > >- Consider the truncated version where ... holds true without using the smoothness condition.
> > > > >
> > > > > **Response.** We are uncertain about how to achieve $\beta_2\leq\mathcal{O}(\frac{1}{n})$ in this context. Notice that the truncated version is a Euclidean projection of $\eta t\hat{\mu}$ onto a convex set $\mathcal{W}$, which, as demonstrated in [7, Lemma 4.6] (in the arxiv version), is 1-expansive. That is, $||\text{Proj}\_\mathcal{X} (x_1)- \text{Proj}\_\mathcal{X} (x_2)|| \leq ||x_1 - x_2||$ for the convex set $\mathcal{X}$. This is also mentioned in the text before Eq.(2) in [8]. Consequently, demonstrating the rate of the untructed version is sufficient for the convex case. Moreover, as Example 1 employs a smooth function, the upper bound of $\beta_2$ in [7] can be directly applied, yielding the exact same rate of $\mathcal{O}(\frac{\eta t}{n})$.
> > > > >
> > > > > [5] Livni, Roi. "Information Theoretic Lower Bounds for Information Theoretic Upper Bounds." arXiv preprint arXiv:2302.04925 (2023).
> > > > >
> > > > > [6] Steinke, Thomas, and Lydia Zakynthinou. "Reasoning About Generalization via Conditional Mutual Information." arXiv preprint arXiv:2001.09122 (2020).
> > > > >
> > > > > [7] Hardt, Moritz, Benjamin Recht, and Yoram Singer. "Train faster, generalize better: Stability of stochastic gradient descent." arXiv preprint arXiv:1509.01240 (2015).
> > > > >
> > > > > [8] Raef Bassily, et al. Stability of stochastic gradient descent on nonsmooth convex losses. NeurIPS 2020.

---

> > > > > > ### Author Response · Authors · 2023-08-19
> > > > > > **Further Comment**
> > > > > >
> > > > > > >- Now let us identify the order of $\sigma$. For...,  $\ell(W_t,Z)$ has $\frac{1}{n}$-bounded difference. Then we can use the Doob martingale to prove that $\ell(W_t,Z)-\mathbb{E}\ell(W_t,Z)$ is $\frac{1}{2\sqrt{n}}$-sub-Gaussian.
> > > > > >
> > > > > > We have just realized that the reviewer determined the order of $\sigma$ based on the $\frac{1}{n}$-bounded difference, which, as explained in the previous response, is not correct. In Example 1, $\ell(W_t,Z)$ has $\frac{1}{\sqrt{n}}$-bounded difference, in which case we think $\sigma$ is still a constant. Please let us know if we are missing anything here.

---

> > > > > > > ### Comment · Reviewer_Ptk3 · 2023-08-19
> > > > > > >
> > > > > > > Please note that I used the truncated version of $w_t$ presented in your paper Line 277.

---

> > > > > > > > ### Author Response · Authors · 2023-08-19
> > > > > > > >
> > > > > > > > Thanks for the reply. We understand, but even considering the truncated version, we still have the order of $\frac{1}{\sqrt{n}}$. Please refer to our previous response.

---

> > > > > > > > > ### Comment · Reviewer_Ptk3 · 2023-08-20
> > > > > > > > >
> > > > > > > > > Dear authors,
> > > > > > > > >
> > > > > > > > > I have checked your previous responses, but I cannot find how you demonstrate that $\ell(W_t, Z)$ has $\frac{1}{\sqrt{n}}$-bounded differences. Could you be more specific?

---

> > > > > > > > > > ### Author Response · Authors · 2023-08-21
> > > > > > > > > >
> > > > > > > > > > Thank you for your response. We have a feeling that the reviewer might igore the condition for using the truncted $w_t$, namely $\eta t||\hat{\mu}||\geq1$ or $||\hat{\mu}||\geq \frac{1}{\sqrt{n}}$ equivalently. We now elaborate more on this.
> > > > > > > > > >
> > > > > > > > > > If we understand correctly, the rate of bounded difference for the truncated version of $w_t$ is the same as the rate for $\sup_{s,s^i}||\frac{w_t}{||w_t||}-\frac{w^i_t}{||w^i_t||}||$. Let's give a quick anwser first: to obtain an upper bound, we have $||\frac{w_t}{||w_t||}-\frac{w^i_t}{||w^i_t||}||\leq ||w_t-w^i_t||\leq\mathcal{O}(1/\sqrt{n})$, where the first inequality is by Lemma 4.6 in [7]. For a lower bound, for GD with $d>\min\\{T,1/\eta^2\\}$, we have $||\frac{w_t}{||w_t||}-\frac{w^i_t}{||w^i_t||}||\geq\Omega(\eta\sqrt{T}+\eta T/n)=\Omega(1/\sqrt{n})$, where the inequality is by Theorem 4.1 in [8]. By combining the upper and lower bounds, we have $\sup_{s,s^i}||\frac{w_t}{||w_t||}-\frac{w^i_t}{||w^i_t||}||\in\mathcal{O}(1/\sqrt{n})$.
> > > > > > > > > >
> > > > > > > > > >
> > > > > > > > > > Then, given that $\sup_{s,s^i}||\frac{w_t}{||w_t||}-\frac{w^i_t}{||w^i_t||}||$ represents the worst-case rate, we provide an example where $||\frac{w_t}{||w_t||}-\frac{w^i_t}{||w^i_t||}||=\mathcal{O}(1/\sqrt{n})$ for specific $s$ and $s^i$. This naturally demonstrates the incorrectness of $\sup_{s,s^i}||\frac{w_t}{||w_t||}-\frac{w^i_t}{||w^i_t||}||\leq\mathcal{O}(1/n)$. To illustrate this, recall the condition of applying the truncted version of $w_t$, namely $||w_t||\geq 1$ or $||\hat{\mu}||\geq \frac{1}{\sqrt{n}}$. Now assume $||\hat{\mu}||=\frac{1}{\sqrt{n}}$, which indicates that each $z_i$ is different from each other. If we replace one instance, $z_i$, in $s$ with a new instance $z'_i$, where $z'_i$ is also distinct from every other instance in $s$, then $||\hat{\mu}^i||=\frac{1}{\sqrt{n}}$. This leads to $||\frac{w_t}{||w_t||}-\frac{w^i_t}{||w^i_t||}||=||\frac{\hat{\mu}}{||\hat{\mu}||}-\frac{\hat{\mu}^i}{||\hat{\mu}^i||}||= \sqrt{n} ||\hat{\mu}-\hat{\mu}^i||=\mathcal{O}(1/\sqrt{n})$.
> > > > > > > > > >
> > > > > > > > > > Therefore, this suggest that $\sup_{s,s^i}||\frac{w_t}{||w_t||}-\frac{w^i_t}{||w^i_t||}||\leq\mathcal{O}(1/n)$ is incorrect.
> > > > > > > > > >
> > > > > > > > > > If the reviewer could provide details on how to obtain $\sup_{s,s^i}||\frac{w_t}{||w_t||}-\frac{w^i_t}{||w^i_t||}||\leq\mathcal{O}(1/n)$, we are open to further discussion. Additionally, we would like to emphasize that $\frac{\sigma}{n}\sum_{i=1}^n\sqrt{I(W;Z_i)}$  will grow with dimesion $d$ in an unbounded manner, regardless of the order of $\sigma$.
> > > > > > > > > >
> > > > > > > > > > We are pleased that the reviewer acknowledges the importance of an algorithm-specific parameter in the information-theoretic bounds. Your previous comments align closely with the early stages of this work.

---

> > > > > > > > > > > ### Comment · Reviewer_Ptk3 · 2023-08-21
> > > > > > > > > > > **Response**
> > > > > > > > > > >
> > > > > > > > > > > Thanks for the clarification. I will verify this. The unbounded MI is not a huge problem actually. Theorem 5.1 of [6] did not make a sub-Gaussian assumption, and the coefficient has been amplified by considering $sup_{w}$.
> > > > > > > > > > >
> > > > > > > > > > > Assume that $g(W, V, \tilde{Z})=\ell(W, \tilde{Z}\_{V}) - \ell(W, \tilde{Z}\_{\bar{V}})$ is $\sigma$-sub-Gaussian under $P_{W}\cdot P_{V}$ for any $\tilde{Z}\sim \mu^{2n}$, where $V \in \\{0,1\\}$. Let $U = (U_1, ..., U_n)\in \\{0,1\\}^n$, then we can prove the following CMI bound:
> > > > > > > > > > > $$
> > > > > > > > > > > \mathbb{E}\_{W,S}[L\_{\mu}(W) -L\_S(W)]\leq \sqrt{\frac{2\sigma^2 I(W;U|\tilde{Z})}{n}}\leq \sqrt{\frac{2\sigma^2 H(U)}{n}} = \sqrt{2\sigma^2\log2} \,.
> > > > > > > > > > > $$

---

> > > > > > > > > > > > ### Author Response · Authors · 2023-08-21
> > > > > > > > > > > >
> > > > > > > > > > > > Thank you for taking the time to verify it. We sincerely hope the reviewer will allow us enough time to respond to any further questions, given that the reviewer-author discussion is concluding in less than 15 hours. Meanwhile, we remain somewhat confuesed by the reviewer's focus solely on the truncated $w_t$, especially when the untruncated version clearly provides a consistent subGaussian variance proxy in this context. In demonstrating a hardness result, the untruncated version already showcases the limitations of the original MI/CMI bounds as a worst-case scenario.

---

> > > > > > > > > > > > > ### Comment · Reviewer_Ptk3 · 2023-08-21
> > > > > > > > > > > > >
> > > > > > > > > > > > > Thanks for the authors' detailed responses. It's true that we have $1/\sqrt{n}$-bounded differences for GD in example 1, where we cannot simply identify a decay $\sigma$. Please clearly describe the truncate step as eq(60) in [2]. My major concern has been fixed, so I raised the score accordingly.

---

> > > > > > > > > > > > > > ### Author Response · Authors · 2023-08-21
> > > > > > > > > > > > > > **Thank you**
> > > > > > > > > > > > > >
> > > > > > > > > > > > > > Thank you for both your constructive review, which has strengthened our manuscript, and the increased score. We will carefully revise our submission based on your suggestions.

---

### Official Review · Reviewer_s7qd · 2023-07-05

**Soundness:** 3 good
**Presentation:** 2 fair
**Contribution:** 2 fair
**Rating:** 5
**Confidence:** 3

**Summary:**

This paper studies how to develop information-theoretic generalization error bounds under the assumption that the algorithm is uniformly stable under a certain loss. Typically, these kinds of bounds are based on properties of the loss with respect to the data (e.g. subgaussian or bounded), while in this case they look at a property of both the loss and the algorithm, namely that it is uniformly stable. Considering these two together allows the bounds to decrease at faster rates as the stability parameter can decrease with the number of samples while the boundedness or subgaussianty parameter does not.

In particular, they extend the individual mutual information bounds from [9] to this assumption. They also consider a matrix of $n \times 2$ hypotheses, where the first column is the output hypotheses, and each row in the second column is the hypothesis outputted by the algorithm when the $i$-th sample is swapped by another sample that is i.i.d. In a similar fashion to [58], they manage to develop conditional mutual information bounds, where they condition on this set of hypotheses. Intuitively, these bounds state the ability to determine if a sample was used to generate the real hypothesis or the hypothesis where the $i$-th sample was swapped. Then, they develop individual conditional mutual information bounds similar to [49] under this setup.

As an application, they showcase that while most of the current information-theoretic bounds fail in the example from [21], the presented bounds can succeed as they are stable with a stability parameter that decreases with the number of samples. Also, they make the case that for very stable algorithms (with constant in $\Omega(1/\sqrt{n})$) their bound improves upon [58]'s. Finally, they connect their results with the Berstein condition and further adapt the bounds from [63] to their setup.

**Strengths:**

The main strength of the paper is the link between stability and the current individual mutual information bounds, which in turn develop into mutual information bounds for stable algorithms with an explicit appearance of the stability parameter. This is good because we know the stability parameter in certain algorithms and this mutual information can be bounded in a good amount of settings.

Similarly, the connection with the Berstein condition is also important as we also know that a good amount of algorithms under certain losses respect this condition, and therefore these bounds can be applied.

Finally, as in [63], the paper showcases how a simple trick like the Donsker-Varadhan together with a careful choice of the function in the supremum can lead to good bounds.

**Weaknesses:**

While the results are interesting, I fail to understand exactly how results other than those in Section 3 are useful. Yes, they can achieve the rates desired in Example 1, but this comes basically because the individual CMI is <= 1 and one can directly have the bound expected generalization error <= $\beta$, which is already known from the uniform stability literature.

* A suggestion for improvement, is to state that directly as a stronger result. Since the individual CMI <= 1, then one may recover the bounds from the uniform stability literature. Then, every result that we know from there is also subsumed into this setup. This way, one can see more clearly that Example 1 works immediately from uniform stability and many other results as well. I believe that this way you may be able to comment on larger sets of problems.

I think that they definitely have potential, but this potential is not explained in the paper. The main reason I mention this is that the previous bounds were dealing with a KL divergence between terms that were "easily tractable" like the posterior $P\_{W|S}$ (or $P\_{W|Z\_i}$) and a prior $Q\_{W}$. However, the terms appearing in these bounds seem very difficult to treat and characterize.

* Another suggestion would be to find an example where this mutual information can be calculated so the reader can have an idea of why this is interesting other than the fact that it can be reduced to the known bound from uniform stability as I mentioned above. An interesting example would include a setup where neither uniform stability nor "classical" information-theoretic bounds can achieve the desired expected generalization error rate, but combined with the presented bounds can.

Some results and statements are included but are loosely explained.

* Theorems 3.3 and 4.3 are examples of this. When can this be useful? Could you provide us with a particular example where we can see how are we benefitting from including $\gamma\_2^2 / \gamma_1$ or $\Lambda(\Tilde{W}\_i)$?
* In Remark 2.2, what do you mean with "it is expected that $\gamma\_4$ is larger than $\gamma\_2$ due to the independence of $Z'$ in Eq. (7)"? A similar claim is done in lines 217-218. When doing this kind of remarks and statements, it would be important to justify more the reasoning behind them.
* The whole set of definitions in Definiton 2.1 is not very clear to me. It would be nice to have a larger, more comprehensive commentary on what are these definitions and why are they included / why are they useful.

Small corrections in the literature, not very important:

* In line 31, it should be "decay at a faster rate, e.g. $\mathcal{O}(1/n)$".
* In line 33, "demonstrating tightness in non-convex learning cases such as deep learning" may be misleading. Some people may interpret that this means that they achieve the correct rate and therefore are tight. An alternative writing could be "demonstrating a good characterization in some instances of non-convex settings (e.g. deep learning)".
* In line 198, $R$-conditioned information-theoretic bounds also appeared in [49].


**Questions:**

* Could you clarify further the definitions in Definition 2.1? Also, could you expand in the reasoning of why some parameters are larger/smaller than the others?

* Could you give some examples where Theorems 3.3 and 4.3 improve upon the preceding theorems?

* Could you try to find examples where neither uniform stability nor "classical" information-theoretic bounds can achieve the desired expected generalization error rate, but combined with the presented bounds can?

* Could you find examples where the bounds with the conditioning can be employed to gain an understanding of some problem beyond the fact that it is stable?

I am happy to increase my score if these questions are satisfactorily answered. I like the paper, but it seems a little immature at the moment. Once these things are addressed, it could become a better paper that ticks many boxes and deals with important questions.

**Limitations:**

Some limitations are included throughout the text and in the Appendices. Some others, like the ones included in the weaknesses, are not. Regarding potential negative societal impact, as the work is of a theoretical, fundamental nature, I do not forsee any issue with not addressing them.

---

> ### Author Rebuttal · Authors · 2023-08-08
>
> Thank you very much for your constructive comments, and we appreciate your positive feedback on our paper. Our responses follow.
>
> >- Could you clarify further the definitions in Definition 2.1? Also, could you expand in the reasoning of why some parameters are larger/smaller than the others?
>
> **Response.** Due to the character constraints in the separate rebuttal, we have made the decision to relocate the response to this question to the global response. Kindly refer to the global response provided above; we sincerely apologize for any inconvenience caused.
>
> >- Could you give some examples where Theorems 3.3 and 4.3 improve upon the preceding theorems?
>
> **Response.** Consider the loss function is bounded by $C$, if an algorithm $\mathcal{A}$ is $\gamma_2$-SCH-B stable but not $\gamma_1$-SCH-A stable (note that if $\mathcal{A}$ is $\gamma_1$-SCH-A stable, it is also $\gamma_1$-SCH-B stable since $\gamma_2\leq\gamma_1$),  we replace $\gamma_1$ by $C$ in Theorem 3.1 and Theorem 3.3, since  $I(\widetilde{W}^+;\widetilde{Z}^+\_{i})$ decays faster than $\sqrt{I(\widetilde{W}^+;\widetilde{Z}^+\_{i})}$, as long as $\gamma_2^2$ also decays faster than $\sqrt{I(\widetilde{W}^+;\widetilde{Z}^+\_{i})}$, Theorem 3.3 is tighter than Theorem 3.1. Similar arguments also apply to comparing Theorem 4.1 and Theorem 4.3 when the $\mathcal{A}$ is $\gamma_4$-SCH-D stable but not $\gamma_3$-SCH-C stable.
>
> In addition, the $\gamma_2$-SCH-B stablility condition in Theorem 3.3 gives us a chance to connect to the Bernstein condition as shown in Corollary 6.1. Thus, if the Bernstein condition is satisfied but $\mathcal{A}$ is not $\gamma_1$-SCH-A stable, Theorem 3.3 will be tighter.
> As a simple concrete example: Let $\mathcal{W}$ be finite, i.e. $|\mathcal{W}|=K$. To simplify the setting, assume $L_\mu(w^*)=0$ and let $\mathcal{A}$ be an interpolating algorithm. In this case, for Theorem 3.1, we have $\mathcal{E}\_\mu(\mathcal{A})\leq\frac{\sqrt{2}C}{n}\sum_{i=1}^n\sqrt{I(W;Z_i)}\leq\sqrt{2}C\sqrt{\frac{I(W;S)}{n}}\leq\mathcal{O}(\log{K}/\sqrt{n})$. For Theorem 3.3 or Corollary 6.1, we have $\mathcal{E}\_\mu(\mathcal{A})\leq\mathcal{O}(\sum_{i=1}^n\frac{I(W;Z_i)}{n})\leq\mathcal{O}(\frac{I(W;S)}{n})\leq\mathcal{O}(\frac{\log{K}}{n})$. Then clearly, Theorem 3.3 is tighter than Theorem 3.1. We will try to construct more examples in the revision.
>
> >- Could you try to find examples where neither uniform stability nor "classical" information-theoretic bounds can achieve the desired expected generalization error rate, but combined with the presented bounds can?
>
> **Response.** This is a good question and we did try to find such examples, but this appears difficult.
>
> The reason is that if the algorithm satisfies certain stability assumptions, stability-based bounds often achieve the optimal rate, particularly after the work of [1,2]. Specifically, in the context of SCO with SGD/GD, where the loss function is either smooth or nonsmooth, stability-based bounds can attain tight upper and lower bounds [3,4]. However, in the context of nonconvex learning, the challenge lies in obtaining the exact generalization error rate and the decaying rate of both the stability parameter and the MI/CMI term.
>
> We note that we have demonstrated that information-theoretic quantities can enhance the stability analysis, as exemplified in Example 2 and Section F.3. Thus, the advantages of our bounds are beyond mere reduction to uniform stability bounds.
> Moreover, when our interest extends beyond the decaying rate w.r.t. $n$, there is a potential case that our bounds improve both stability-based bounds and the MI/CMI bounds before. For instance, in Figure 3(d) in [5], previous MI/CMI bounds are loose at the early phase of training. As the stability parameter usually grows with the iteration number, it remains small at the beginning, and its large value at the end can be mitigated by the MI/CMI term. Consequently, the product of the stability parameter and the MI/CMI term could potentially offer a more accurate reflection of the dynamics of the true generalization error. The  remaining challenge is to rigorously characterze SCH parameters for SGD (or SGLD), as previous analyses were mainly limited to characterizing $\beta_1$.
>
> >- Could you find examples where the bounds with the conditioning can be employed to gain an understanding of some problem beyond the fact that it is stable?
>
> **Response.** We note that CMI bounds have already been widely discussed in the existing literature where the algorithm is not necessarily stable. In Section F.3, we establish a connection between our newly introduced CMI terms and the classical VC theory, building upon previous research. Consequently, for learning problems with finite VC dimension, the CMI terms themselves can serve as a means to demonstrate the learnability of such problems.
>
> >- A suggestion for improvement, ...
> >- Another suggestion would be ...
> >- Small corrections in the literature, ...
>
> Thank you again for providing these valuable suggestions for improving our paper, we will revise our paper according to your comments.
>
> Please do let us know if you have further questions.
>
> [1] Yegor Klochkov, et. al. Stability and Deviation Optimal Risk Bounds with Convergence Rate O(1/n). NeurIPS 2021.
>
> [2] Olivier Bousquet, et. al. Sharper bounds for uniformly stable algorithms. COLT 2020.
>
> [3] Moritz Hardt, et. al. Train faster, generalize better: Stability of stochastic gradient descent. ICML 2016.
>
> [4] Raef Bassily, et al. Stability of stochastic gradient descent on nonsmooth convex losses. NeurIPS 2020.
>
> [5] Ziqiao Wang, et al. Tighter information-theoretic generalization bounds from supersamples. ICML 2023.

---

> > ### Comment · Reviewer_s7qd · 2023-08-14
> > **Answer to rebuttal**
> >
> > Thank you for your rebuttal (both the general one and this one). Let me continue the discussion on some still unclear topics.
> >
> > 1. *Regarding the definitions in Definition 2.1.* \
> > Thank you for the explanations. Apart of including them in the text, could you give some examples of reasonable generality, where $\gamma\_2 \leq \gamma\_4$ and where $\gamma\_2, \gamma\_3,$ and $\gamma_4$ are smaller or equal to $\beta\_1$?
> >
> > 2. *Regarding the examples where Theorems 3.3 and 4.3 improve upon known results.*
> >     * *Regarding the first comparison between Theorem 3.1s and  3.3.* \
> > It is clear that if the constants decay faster than the terms in Theorem 3.1, then it is tighter. However, when would that happen? Do you have some examples where you can state that this is the case?
> >     * *Regarding the second comparison between Theorems 3.1 and 3.3.* \
> > If $\mathcal{A}$ is has a bounded domain $|\mathcal{W}| = K$ it is known with classical information-theoretic techniques that $\mathcal{E}\_\mu \in \mathcal{O}(\frac{\log K}{\sqrt{n}})$. Also, if $\mathcal{A}$ is an interpolating algorithm we know that $\mathcal{E}\_\mu \in \mathcal{O}(\frac{\log K}{n})$ already from [58, Theorem 5.7] or [24,25].  \
> > Still, this could be added to the main text to help understand these results.
> >
> > 3. *Regarding situations where presented bounds improve situations where neither stability nor classical information-theoretic can achieve the desired expected generalization error rate (also asked by the AC).* \
> > In Example 2, the bound is obtainable with previous known information-theoretic bounds, e.g. those from [21] as mentioned by the authors. I think that finding an example where neither stability nor classical information-theoretic bounds achieve a desired rate, but their combination does would improve substantially the paper.
> >
> > 4. *Regarding examples where the bounds with the conditioning can be employed to gain an understanding of some problem beyond the fact that it is stable.* \
> > I agree with the authors that this has been studied for other notions of CMI. My question is, what do we gain from using your particular conditioning instead of those in [19,20,23,25,58]?
> >
> > 5. *Regarding the suggestions*. \
> > Thank you for revising the paper to include the suggestions. Could you specify which examples where these mutual information can be calculated are you including in the revised version of the paper?

---

> > > ### Author Response · Authors · 2023-08-15
> > >
> > > Thanks for your reply. Our response follows.
> > >
> > > >- could you give some examples of reasonable generality, where $\gamma_2\leq\gamma_4$ and where $\gamma_2$, $\gamma_3$ and $\gamma_4$ are smaller or equal to $\beta_1$?
> > >
> > > For $\gamma_2\leq\gamma_4$, first, we again highlight that $Z'$, which is used in $\gamma_2$, represents an independent testing instance for both $W$ and $W^i$, while $Z_i$ used in $\gamma_4$ is a training instance for $W$, it serves as a testing instance for $W^i$. To compare them, To compare them, we begin by applying Jensen's inequality in $\gamma_2$, then the remaining intuition is motivated by |test_loss - test_loss|$\leq$ |train_loss - test_loss|. As a concrete example, let $\ell$ be zero-one loss and assume $\mathcal{A}$ is an interpolating algorithm and and randomly makes predictions for unseen data. By Jensen's inequality, $\gamma\_2^2\leq\mathbb{E}\_{W,W^i,Z'}{\left[\ell(W,Z')-{\ell(W^i,Z')}\right]^2}=\mathbb{E}\_{W,Z'}\left[\ell(W,Z')\right]-2\mathbb{E}\_{W,W^i,Z'}\left[(\ell(W,Z')\ell(W^i,Z'))\right]+\mathbb{E}\_{W^i,Z'}\left[\ell(W^i,Z')\right]^2$,
> > > where we use $\ell^2=\ell$ for zero-one loss.
> > > Since $Z'$ is an unseen data for both $W$ and $W^i$, we have
> > > $\gamma_2^2\leq\mathbb{E}\_{W^i,Z'}\left[\ell(W^i,Z')\right]^2+\frac{1}{2}-\frac{1}{2}=\mathbb{E}\_{W^i,Z'}\left[\ell(W^i,Z')\right]^2$. While in this case $\gamma_4^2=\mathbb{E}\_{W^i,Z_i}{\left[{\ell(W^i,Z_i)}\right]^2}$ so $\gamma_2\leq\gamma_4$.
> > >
> > > For $\gamma_2,\gamma_3,\gamma_4$ vs. $\beta_1$, please notice that $\gamma\_2\leq\beta\_1$ and $\gamma\_4\leq\beta\_1$ can be rigorously proved by $\mathbb{E}\leq\sup$. Consider a Gaussian location estimation problem, let $\ell(w,z)=||w-z||\_2$ and let $Z\sim\mathcal{N}(0,\sigma^2)$. Note that the ERM solution is the sample mean, $W=\frac{1}{n}\sum_{i=1}^n Z_i$, and the Euclidean distance is 1-Lipschitz. Thus, for $\gamma_2$, we have $\gamma^2\_2\leq \mathbb{E}\_{S,Z'\_i}||W-W^i||^2=\mathbb{E}\_{Z\_i,Z'\_i}\frac{1}{n^2}||Z\_i-Z'\_i||^2=\frac{2\sigma^2}{n^2}$. Notice that $\gamma_3^2$ and $\gamma_4^2$ share the same upper bound in this case. For $\beta_1$, $\sup\_{s\simeq s^i,z}| \ell(w,z)-\ell(w^i,z)|\leq \sup\_{z\_i,z'\_i}\frac{1}{n}||z'\_i-z\_i||\to\infty$, which implies that $\beta\_1\to\infty$. Thus, we conclude that $\gamma\_2,\gamma\_3,\gamma\_4\leq\beta\_1$.
> > >
> > > We have a sense that the reviewer may still have some confusion regarding our Definition 2.1, please do let us know if any specific definitions remain unclear.
> > >
> > > >- However, when would that happen? Do you have some examples where you can state that this is the case?
> > >
> > > The previous Gaussian location estimation problem is such a case. First, $\gamma_2^2/\gamma_1\leq\gamma_2\leq\mathcal{O}{(\frac{1}{n})}$ ( $\gamma_2\leq\gamma_1$ was explained in the previous response), namely the second term in Theorem 3.3 decays with $\mathcal{O}{(\frac{1}{n})}$. Additionally, $I(W;Z_i)\leq\mathcal{O}{(\frac{1}{n})}$ as proved in [1, Example 1] so $\frac{1}{n}\sum_{i=1}^n\sqrt{I(W;Z_i)}\leq\mathcal{O}{(\frac{1}{\sqrt{n}})}$. Hence, $\gamma_2$ decays faster than $\frac{1}{n}\sum_{i=1}^n\sqrt{I(W;Z_i)}$.
> > >
> > > >- Regarding the second comparison.
> > >
> > > We acknowledge that a similar conclusion is already established, but this specific example validates that Theorems 3.3 can result in a tighter bound compared to Theorems 3.1.
> > >
> > > >- Regarding situations where presented bounds improve situations where neither stability nor classical information-theoretic can achieve the desired expected generalization error rate.
> > >
> > > We provide such an example; please refer to the response to AC.
> > >
> > > >- Regarding examples where the bounds with the conditioning can be employed to gain an understanding of some problem beyond the fact that it is stable.
> > >
> > > About comparing our novel CMI notions with the classical CMI notions, we do provide a brief discussion in Section G and acknowledge that this might not be a straightforward task.  In our attempt, we treat the classical CMI notions as the forward channel ($P_{W|\widetilde{Z}}$) while addressing our CMI notions in Section 4 as the backward channel ($P_{\hat{Z}|\widetilde{W}}$). However, we currently cannot offer any new insights in this regard.
> > > At present, as done in this paper, we focus on demonstrating that our new CMI notions preserve the favorable properties of the original CMI notions, including the boundedness, , being upper-bounded by the unconditional individual bound (Theorem 4.2), establishing a connection with VC theory (Theorem F.1), and exhibiting  $f$-CMI, e-CMI, and ld-CMI counterparts (Theorem 6.1).
> > >
> > > >-  Could you specify which examples where these mutual information can be calculated are you including in the revised version of the paper?
> > >
> > > We believe that Example 3, as presented in our response to AC, demonstrates that our bounds are not merely a reduction of the known bound derived from uniform stability.
> > >
> > >
> > > [1] Rodríguez Gálvez, et al. Tighter expected generalization error bounds via Wasserstein distance. NeurIPS 2021.

---

> > > > ### Comment · Reviewer_s7qd · 2023-08-15
> > > > **Answer to further clarifications**
> > > >
> > > > Thank you for the clarifications, they were very useful. Please, make sure to include them in the main text. I will reiterate that below just for completeness.
> > > >
> > > > 1. Thank you. The examples helped me understand better the definitions and your expectations about their relationships. If possible, please include them in the main text. It makes the reading of the text later easier.
> > > >
> > > > 2. Thanks. Please add them to the main text as well, I think they make the contributions clearer.
> > > >
> > > > 3. Thank you. Again, please add this to the main text. I believe that this example is great to see a situation where one make the most out of the proposed bounds.
> > > >
> > > > 4. I understand. I hope this can be done in a journal extension of the paper or in future works.
> > > >
> > > > 5. Thanks.

---

### Official Review · Reviewer_LJ4V · 2023-07-06

**Soundness:** 4 excellent
**Presentation:** 3 good
**Contribution:** 3 good
**Rating:** 7
**Confidence:** 4

**Summary:**

This work improves information-theoretic generalization gap bounds by doing more careful derivations. As a result, the derived bounds get a multiplicative factors that capture some notions of hypothesis stability, while the existing bounds usually have a multiplicative constant factors that depend on the loss function and the data distribution. For example, the bound of Bu et al. [9] that states that $\text{expected gen. gap} \le \frac{1}{n}\sum_i \sqrt{2 I(W;Z_i)}$, with $W$ being the output of the learning algorithm on the dataset $(Z_1,\ldots,Z_n)$, gets improved by a multiplicative stability term $\gamma_1 \le 1$ that captures how much (in the worst-case) the loss on an example $z'$ can change when one replaces a single training example. Similar improvements are derived for conditional mutual information (CMI) bounds, both with standard CMI terms and a novel CMI term. In the random-subsample setting of Steinke and Zakynthinou [58], this novel CMI term measures the conditional mutual information between the $i$-th example that contributed to the training and the selection variable of the $i$-th pair, given two hypotheses: one corresponding to training with the first example of the $i$-th pair, the other corresponding to training with the second example, keeping everything else the same.

The authors show that in some known cases when existing information-theoretic bounds fail to vanish with $n$, adding the stability factor result in a vanishing bound. They also show that there are cases when the stability term alone does not vanish, but when the mutual information part is also considered, the resulting bound vanishes.

**Strengths:**

**Strength #1: Significance & Originality.** Information-theoretic generalization bounds have gained a lot of attention recently, partly because they are algorithm and distribution-dependent and some variants of them are nonvacuous in practical setting for deep learning. Recently, Haghifam et al [21] uncovered some limitations of information-theoretic bounds, constructing cases when the existing bounds do not even vanish with $n$. This submission addresses these limitations, and is therefore a significant contribution. While the derivations mostly follow common techniques, the resulting unification of stability-based and information-theoretic bounds is a novel (to my best knowledge) and significant contribution.

**Strength #2: Quality.**  The submission is technically sound.  I have checked the proofs. The related work is cited adequately.

**Strength #3:  Clarity & Presentation.** This work is well-written and presented.

**Weaknesses:**

**Weakness #1: Limitations [minor].** As the authors acknowledge, the proposed stability terms are hard to evaluate or estimate in practical deep learning settings. Therefore, the extent of improvements outside of the considered synthetic settings is unclear.

**Questions:**

- Line 153: "are identically but not independent distributed" --> "are identically distributed but not independent".
- Line 169: It should be $\gamma_2 \le \gamma_1$ and $\gamma_4 \le \gamma_3$.
- Line 219: there might be no uniform convergence if the conditional mutual information terms vanish with different rates for different $\tilde{w}_i$ such that their supremum does not vanish.
- Equation (14): $W_i$ and $\bar{W}_i$ need to be defined in the Lemma statement.
- Theorem 4.4: Is the assumption of $\mathcal{A}$ being symmetric with respect to $S$ necessary?
- Line 280: Are $\hat{\mu}$ and $\hat{\mu}^i$ treated like constants here?
- Lines 713-720: If $E_0$ denotes the event of two elements of $\tilde{Z}$ sharing the same non-zero coordinate, then we are interested in upper bounding $p(E_0)$ (in contrast to what is written in line 717). On line 720, "the first inequality holds because when event $E_0$ occurs, one can determine the value of $U_i$ completely", it should be when $E_0$ *does not* occur.

**Limitations:**

The limitations are adequately addressed.

---

> ### Author Rebuttal · Authors · 2023-08-08
>
> Thank you very much for your positive comments. Our responses follow.
>
> >- Line 219: there might be no uniform convergence if the conditional mutual information terms vanish with different rates for different $\tilde{w}_i$ such that their supremum does not vanish.
>
> **Response.** We agree that if the CMI with the worst $\tilde{w}_i$ does not vanish, the current bound cannot be considered a uniform convergence bound. We will provide clarification in the revised version.
>
> >- Theorem 4.4: Is the assumption of $\mathcal{A}$ being symmetric with respect to $S$ necessary?
>
> **Response.** It is necessary in our proof of Theorem 4.4, otherwise we may not be able to obtain the inequalities in Line 678.
>
>
> >- Line 280: Are $\hat{\mu}$ and $\hat{\mu}^i$ treated like constants here?
>
> **Response.** In our main text, yes, we let $\hat{\mu}$ and $\hat{\mu}^i$ be fixed. However, it is worth noting that all these developments still apply when $\hat{\mu}$ and $\hat{\mu}^i$ are treated as random variables, namely $||\eta t\hat{\mu}-\eta t\hat{\mu}^i||=\eta t||\frac{1}{n}\sum_{j=1}^n Z_j-\frac{1}{n}(\sum_{j\neq i} Z_j+Z'_i)||=\frac{\eta t}{n}||Z_i-Z'_i||\leq\mathcal{O}({\eta t}/{n})$.
>
> >- Lines 713-720: If $E_0$ denotes the event of two elements of $\widetilde{Z}$ sharing the same non-zero coordinate, then we are interested in upper bounding $p(E_0)$ (in contrast to what is written in line 717). On line 720, "the first inequality holds because when event $E_0$ occurs, one can determine the value of $U_i$ completely", it should be when $E_0$ does not occur.
>
> **Response.** Thank you very much for your careful reading and pointing out these. There indeed exists a typo, $E_0$ should be the event that **no** pair of instances in $\widetilde{Z}$ share the same non-zero coordinate. We apologize for this oversight, and we have corrected this in the revision.
>
> >- Line 153:...; Line 169:...;Equation (14):....
>
> **Response.** Thanks for the suggestions, we have revised our paper according to your comments.

---

> > ### Comment · Reviewer_LJ4V · 2023-08-12
> > **Reviewer response**
> >
> > Thank you for the clarifications.

---

### Author Rebuttal · Authors · 2023-08-08

# To all reviewers, particularly to Reviewer s7qd:

>- Reviewer s7qd has pointed out that our Definition 2.1 lacks clarity. We would like to address this concern by offering the following commentary:

We first note that the reason we introduce SCH stabilities in Definition 2.1 is that solely using $\beta_2$ (as mentioned in Line 132) in our bounds might be too loose for the randomized setting (but it is still weaker than the bounded loss assumption), as it considers the supremum over all sources of randomness. By incorporating SCH stabilities, we aim to demonstrate that theoretically, we can achieve significantly tighter stability parameters.

The basic set up is as follows. Assume a random sample $S$ gives rise to $W$. For each $Z_i\in S$, we construct $S^i$ by replacing $Z_i$ with another independently drawn instance; call the training result $W^i$, the neighbor of $W$.

In (a), $\gamma_1$-SCH-A stability measures the difference between the loss of $w$ and the expected loss of its neighbor $W^i$ at a worst $z$ and the worst possible $w$. While in (b), $\gamma_2$-SCH-B stability measures the square of this difference, not in the worst case, but in an average case, where the average is over an independently $Z'$ for the loss evaluation, the training sample, and the algorithm randomness. Since "average is smaller than worst", $\gamma_2\leq\gamma_1$ can be rigorously proved.

In (c),  we consider the difference between the loss of $W$ and the loss of its neighbor when evaluated at the worst possible $z_i$ that when included in $S$ gives rise to $W$. The expected value of this difference is $\gamma_3$-SCH-C stability. While in (d), $\gamma_4$-SCH-D stability measures the expected squared difference between the loss of $W$ and the loss of its neighbor when evaluated at $Z_i$ (a member of $S$). For a similar "average smaller worst" reason, one expects that $\gamma_4\leq\gamma_3$. However, this result can not be rigourously proved. We will revise this in Remark 2.2. Note that this relationship has never been used in this work.

We expect that $\gamma_2$, $\gamma_3$, and $\gamma_4$ are all smaller than $\beta_1$. This is because in $\beta_1$, we consider the worst evaluated instance, whereas in the other cases, we take the expectation over all instances. In Line 217-218, we also expect that ${\mathbb{E}\_{\widetilde{W}_i}{\Delta_1(\widetilde{W}_i)^2}}\leq\beta^2_1$, similarly, this is because $\beta_1$-stability holds for all the possible $s$ and $s^i$, namely it holds for all the $(w,w^i)$ pair (that shares the same randomness) while in ${\mathbb{E}\_{\widetilde{W}_i}{\Delta_1(\widetilde{W}_i)^2}}$, we take the expectation of these pairs.

We expect $\gamma_2\leq\gamma_4$ due to the following reason: first by Jensen's inequality, we have $\mathbb{E}\_{S,R,Z'}{\left[\ell(W,Z')-\mathbb{E}\_{W^i|W}{\ell(W^i,Z')}\right]^2}\leq\mathbb{E}\_{W,W^i,Z'}{\left[\ell(W,Z')-{\ell(W^i,Z')}\right]^2}$, then since $Z'$ is independent of both $W$ and $W'$, $Z'$ can be regarded as a testing data point for both $W$ and $W'$, we could expect that the expectation of $\ell(W,Z')-{\ell(W^i,Z')}$ is small. While in $\mathbb{E}\_{S,Z'_i,R}{\left[\ell(W,Z_i)-\ell(W^i,Z_i)\right]^2}$, $Z_i$ is a training data point for obtaining $W$, so $\ell(W,Z_i)$ could be small in general, and $Z_i$ is a testing point for $W^i$. Therefore, it is reasonable to expect that $\mathbb{E}\_{W,W^i,Z'}{\left[\ell(W,Z')-{\ell(W^i,Z')}\right]^2}\leq\mathbb{E}\_{S,Z'_i,R}{\left[\ell(W,Z_i)-\ell(W^i,Z_i)\right]^2}$, namely $\gamma_2\leq\gamma_4$.

We will include these explainations in the revision.

---

### Comment · Area_Chair_7BFw · 2023-08-14
**Further clarification from AC**

Dear authors, I would like the following further clarification from you in order to be more focused when assessing this paper.

I am looking at your example1 and example2. together with the statement "These two examples demonstrate that our bounds can improve both the stability-based bound and information-theoretic bounds in some learning scenarios."

At a quick glance on both examples, it seems you employ a generalization bound of the form:

$$ \mathcal{E}_\mu (A) \le \left(\textrm{stability bound}\right)\cdot \left(\textrm{information-theoretic bound}\right)$$

Clearly this bound can improve over one bound when the other is meaningful. But one could argue that in both your examples the bound

$$ \mathcal{E}_\mu (A) \le \min\left( \left(\textrm{stability bound}\right), \left(\textrm{information-theoretic bound}\right)  \right)$$

improves over both bounds, exactly the same.

Do you have an example where both stability bound **and** information theoretic bounds are vacuous or insufficient, yet you can obtain a novel generalization bound?

---

> ### Author Response · Authors · 2023-08-15
>
> Dear AC, thanks for your comment. Our response follows.
>
> >- Do you have an example where both stability bound and information theoretic bounds are vacuous or insufficient, yet you can obtain a novel generalization bound?
>
> **Response.** Thank you for pushing us to find such an example. Here is one:
>
> **Example 3.** Assume that $\mathcal{W}\in\mathbb{R}^d$ is a bounded ball with radius $\frac{1}{d}$, and let the input space be $\mathcal{Z}=\\{z_0d, -z_0d\\}$ where $z_0\in\mathcal{W}$ such that $||z_0||=\frac{1}{d}$. Let $\mu={\rm Unif}(\mathcal{Z})$. Now consider a  convex and L-Lipschiz loss function $\ell(w,z)=-L\langle w, z \rangle$. In addition, let $d=\sqrt{n}$ and let $\mathcal{A}$ be any empirical risk minimization (ERM) algorithm.
>
> In this example, let $S=(Z_1,\dots,Z_n)\sim \mu$ be the training dataset. Let the Rademacher random variable $\epsilon_i = 1$ if $Z_i=z_0d$ and $\epsilon_i = -1$ if $Z_i=-z_0d$. Hence, we can represent the training set as $S=(z_0d\epsilon_1,\dots,z_0d\epsilon_n)$. The empirical risk for $w$ is $\frac{-Ld}{n} \big \langle w,z_0\sum\nolimits_{i}\epsilon_i \big \rangle$. Note that the ERM solution will be $z_0$ if $\text{sign}(\sum_{i=1}^{n}\epsilon_i)=1$ and will be $-z_0$ if $\text{sign}(\sum_{i=1}^{n}\epsilon_i)=-1$.
>
>
> Notice that the population risk in this case is zero. Recall that $d=\sqrt{n}$, then the expected generalization error of ERM is
> $$
> \mathcal{E}=\mathbb{E}\left[\frac{Ld}{n} \big \langle w,z_0\sum\nolimits_{i}\epsilon_i \big \rangle\right]=\frac{L}{n\sqrt{n}}\mathbb{E} \left[ \left|\sum_{i=1}^{n} \epsilon_i \right|\right]
>     \geq \frac{L}{\sqrt{2}n}=\mathcal{O}(\frac{1}{n}),
> $$
> where the last inequality is by Khintchine–Kahane inequality [1,Theorem D.9].
>
> Now, for the stability bound alone, $\sup_{s\simeq s^i,z}| \ell(w,z)-\ell(w^i,z)|\leq L||w-w^i||\leq 2L||z_0||=\frac{2L}{\sqrt{n}}$. Thus, $\beta_2=\frac{2L}{\sqrt{n}}$ achieves a slower rate.
>
> Now, for MI alone, note that the following Markov chain holds:
> $S - \text{sign}(\sum_{i=1}^{n}\epsilon_i) - W$. Then,
> $$
> \frac{1}{n}\sum_{i=1}^n \sqrt{I(W;Z_i)}\leq \sqrt{\frac{I(W;S)}{n}}\leq \sqrt{\frac{I(W;\text{sign}(\sum_{i=1}^{n}\epsilon_i))}{n}}\leq\sqrt{\frac{H(\text{sign}(\sum_{i=1}^{n}\epsilon_i))}{n}}\leq\frac{1}{\sqrt{n}},
> $$
> where the last inequality is due to the fact that $\text{sign}(\sum_{i=1}^{n}\epsilon_i)$ is a binary random variable.
>
> Thus, the MI part itself also has a slower rate.
>
> Now, for the new bound in our paper, we have
> $$
> \frac{\beta_2}{n}\sum_{i=1}^n \sqrt{I(W;Z_i)}\leq \frac{2L}{n}=\mathcal{O}(\frac{1}{n}).
> $$
>
> This bound then matches the rate of the generalization lower bound, so it obtains the optimal rate.
>
>
> [1] Mohri, Mehryar, Afshin Rostamizadeh, and Ameet Talwalkar. Foundations of machine learning. MIT press, 2018.

---

> > ### Author Response · Authors · 2023-08-15
> >
> > >- At a quick glance on both examples, it seems you employ a generalization bound of the form:
> > $$
> > \mathcal{E}\_\mu(\mathcal{A})\leq(\text{stability bound})\cdot(\text{information-theoretic bound})
> > $$
> > Clearly this bound can improve over one bound when the other is meaningful. But one could argue that in both your examples the bound
> > $$
> > \mathcal{E}\_\mu(\mathcal{A})\leq\min((\text{stability bound}), (\text{information-theoretic bound}))
> > $$
> > improves over both bounds, exactly the same.
> >
> > **Response.** While we are fully on board what AC's perspective is, the statement you make is slightly inaccurate. Thus, we want to clarify this in addition to providing the example above.
> >
> > In $\mathcal{E}\_\mu(\mathcal{A})\leq(\text{stability bound})\cdot(\text{information-theoretic bound})$, note that this should be more precisely written as $\mathcal{E}\_\mu(\mathcal{A})\leq(\text{stability param.})\cdot(\text{information-theoretic quantity})$. The entire RHS component is referred to as the information-theoretic bound within our paper. We agree that $\text{stability param.}$ is the same thing with $\text{stability bound}$, but information-theoretic quantity, either MI or CMI, itself is not a valid bound. The choice of the parameter accompanying MI or CMI depends on the techniques and assumptions used in bounding CGF, and we indeed want to highlight that this step is significant in the information-theoretic framework. In addition, as information-theoretic bounds are known to be algorithm-dependent, making full use of the algorithm properties isn't deceptive; it's essential. If the algorithmic stability holds the key to achieving precise generalization, then it's reasonable to anticipate that information-theoretic bounds would reduce to stability bounds.
> >
> > For $\mathcal{E}\_\mu(\mathcal{A})\leq\min((\text{stability bound}), (\text{information-theoretic bound}))$, agreeably this bound is sufficient for both Example 1 and Example 2. However, if doing so is satisfied, then we would have no chance to provide a defense for the information-theoretic bounds in Example 1.
> >
> > At this point, we would like to re-emphasize our main motivation and main contribution: We hope to demonstrate that information-theoretic framework is able to explain the learnablity of examples in [2].
> > To achieve this, we use the stability property to bound the CGF in the Donsker-Varadhan (DV) variational representation of the KL divergence, and construct a novel $\widetilde{W}$ matrix. That is, we advance the techniques for proving information-theoretic bounds, and they typically take the form of $(\text{stability param.})\cdot(\text{information-theoretic quantity})$, thus overcoming the limitations explored in [2].
> >
> > Furthermore, we remark that the fundamental reason behind the failure of classical information-theoretic bounds in explaining the learnability of SCO with GD/SGD is because MI and CMI are dimension-dependent, while the generalization error in SCO with GD/SGD is dimension-independent. It's hard to believe at the moment that there exist new MI or CMI notions that could be completely dimension-independent in these cases. Hence, using a better $\text{param}.$ associated with MI or CMI emerges as a potential solution.
> >
> >
> > [2] Haghifam, Mahdi, et al. Limitations of information-theoretic generalization bounds for gradient descent methods in stochastic convex optimization. ALT 2023.

---

> > > ### Comment · Area_Chair_7BFw · 2023-08-16
> > > **Thanks**
> > >
> > > Thanks, this is an informative example, and will help assess the paper in the discussion.
> > >
> > >
> > > I do question your fundamental motivation. MI bounds are indeed dimension dependent but I fail to see how a bound of the type $O(\textrm{stability}\cdot \textrm{Information quantity})$ could remedy this. Unless you hope to demonstrate stability terms that are **inverse** in the dimension, the aforementioned bound inherits the dimension dependence of the MI term.

---

> > > > ### Author Response · Authors · 2023-08-16
> > > >
> > > > Thank you for your reply. Our response follows.
> > > >
> > > > >- MI bounds are indeed dimension dependent but I fail to see how a bound of the type $\mathcal{O}(\text{stability}\cdot\text{Information quantity})$ could remedy this. Unless you hope to demonstrate stability terms that are inverse in the dimension, the aforementioned bound inherits the dimension dependence of the MI term.
> > > >
> > > > **Response.** Firstly, we note that the dimension-dependent property of information quantity doesn't necessarily imply unbounded growth with the dimension. Let's delve into CMI and MI separately.
> > > >
> > > > Regarding CMI, by the construction of $\widetilde{W}$ in this paper, our new CMI preserves the nice bounded nature, i.e. $\text{individual CMI}\leq H(U_i)=\log{2}$. Thus, $\mathcal{O}(\text{stability}\cdot\text{CMI})\leq \mathcal{O}(\text{stability}\cdot\log{2})$. As mention in Line 292-293, this suggests that our CMI bound can explain generalization as long as the stability-based bound is sufficient. In other words, there exist some dimension settings where CMI becomes a constant, and will not grow with the dimension in an unbounded manner. In addition, $\text{CMI}\leq\text{MI}$ (see Theorem 4.2), so  $\mathcal{O}(\text{stability}\cdot\text{CMI})$ also gives the optimal rate in Example 3 above. We hope this can address AC's concern.
> > > >
> > > > For MI, there is no strict guarantee that we can prevent it grow with the dimension, and this is still an ongoing challenge. The prevalent method to mitigate this is to add noise perturbation to the algorthm output [3]. For example, let $\hat{W}=W+N$ where $N\sim\mathcal{N}(0,\sigma^2I_d)$, and let $\mathcal{A}\_W$ be the original algorithm and let $\mathcal{A}\_{\hat{W}}$ be the perturbed one. Then, $\mathcal{E}\_\mu(\mathcal{A}\_W)= \mathcal{E}\_\mu(\mathcal{A}_W)+\mathcal{E}\_\mu(\mathcal{A}\_{\hat{W}})-\mathcal{E}\_\mu(\mathcal{A}\_{\hat{W}})\leq \mathcal{O}(I(\hat{W};S))+|\mathcal{E}\_\mu(\mathcal{A}_W)-\mathcal{E}\_\mu(\mathcal{A}\_{\hat{W}})|$. Note that the noise distribution can be selected arbitrarily. To neutralize the dependence between $I(\hat{W};S)$ and $d$, one can select $\sigma^2=\mathcal{O}(d)$, (i.e. large peturbation reduces the dependence between $\hat{W}$ and $S$). However, let $\sigma^2=\mathcal{O}(d)$ will make the second term $|\mathcal{E}\_\mu(\mathcal{A}_W)-\mathcal{E}\_\mu(\mathcal{A}\_{\hat{W}})|$ grow with $d$. If we further have some boundedness assumption for the loss function or for the loss difference, it is possible that this bound will not grow with the dimension in an unbounded manner. Again, this might not have a guarantee, and the bound in [3] fails to explain th learnability of the SCO example as shown in [2].
> > > >
> > > > [3] Neu, Gergely, et al. "Information-theoretic generalization bounds for stochastic gradient descent." COLT 2021.

---

> > > > > ### Comment · Area_Chair_7BFw · 2023-08-16
> > > > > **But it has nothing to do with stability**
> > > > >
> > > > > - MI, there is no strict guarantee that we can prevent it grow with the dimension, and this is still an ongoing challenge.
> > > > >
> > > > > First, I fail to see how your discussion has anything to do with stability factor. My question was, **if** the information is dimension dependent, how do you expect your bound to be dimension independent?
> > > > >
> > > > > Second, regarding hope that the information is dimension independent. Indeed, as you previously stated, there is no hope for dimension independent bound in the general case (see Livni "information theoretic lower bounds for information theoretic upper bounds).

---

> > > > > > ### Author Response · Authors · 2023-08-16
> > > > > >
> > > > > > Thank you for clarifying your question.
> > > > > >
> > > > > > >- Unless you hope to demonstrate stability terms that are inverse in the dimension
> > > > > >
> > > > > > This is true in Example 3, where $d=\sqrt{n}$ and $\beta_2=\frac{2L}{\sqrt{n}}=\frac{2L}{d}$. This is also true in Example 1, $d=2n^2$ and $\beta_2=\mathcal{O}(\frac{1}{\sqrt{n}})=\mathcal{O}(\frac{1}{d^{1/4}})$.
> > > > > >
> > > > > > >- My question was, if the information is dimension dependent, how do you expect your bound to be dimension independent?
> > > > > >
> > > > > > When information quantity like CMI and MI becomes constant (e.g., MI$\leq 1$ in Example 2), you can regard it as dimension-independent.
> > > > > >  Then $\mathcal{O}(\text{stability}\cdot\text{information quantity})\leq \mathcal{O}(\text{stability}\cdot \text{Constant})= \mathcal{O}(\text{stability})$ is completely dimension independent if $\text{stability}$ is also dimension independent.
> > > > > >
> > > > > > In general, $\mathcal{O}(\text{stability}\cdot\text{information quantity})$ is dimension dependent if we don't consider the relationship between $d$ and $n$ as Livni's work.

---

> > > > > > > ### Comment · Area_Chair_7BFw · 2023-08-16
> > > > > > > **The relationship between d and n**
> > > > > > >
> > > > > > > Well, the whole point in having dimension independent bounds is to achieve dimension independent sample size  so it doesn't really make sense to claim the stability term is inverse in the dimension because you consider a dimension dependent sample size (one could argue then that every bound is dimension independent if the sample is linear in the dimension)

---

> > > > > > > > ### Author Response · Authors · 2023-08-16
> > > > > > > >
> > > > > > > > We would like to thanks AC very much for fostering this discussion, which also helps us to understand the problem better.
> > > > > > > > >- the whole point in having dimension independent bounds is to achieve dimension independent sample size so it doesn't really make sense to claim the stability term is inverse in the dimension
> > > > > > > >
> > > > > > > > We completely agree with this, and if stability only depends on $n$ without any further relationship between $n$ and $d$, we will just say stability is dimension independent. Yet, notice that $\beta=\frac{2L}{d}$ still holds in Example 3 even without the relationship between $n$ and $d$, so stability could be dimension dependent.
> > > > > > > >
> > > > > > > > We now focus on where stability is dimension independent while CMI is dimension dependent (SCO with GD/SGD). It's easy to show
> > > > > > > > $$
> > > > > > > > \mathcal{O}(\text{stability}\cdot\text{CMI})\leq \sup_{d}\\mathcal{O}(\text{stability}\cdot \text{CMI})= \mathcal{O}(\text{stability}\cdot\sup_{d}\text{CMI})\leq \mathcal{O}(\text{stability}\cdot \text{constant})=\mathcal{O}(\text{stability})
> > > > > > > > $$
> > > > > > > > That is, **the dimension dependent CMI bound in this paper has a dimension independent upper bound**, and is able to explain generalization as long as the stability bound can.
> > > > > > > >
> > > > > > > > In general, we cannot prove $\mathcal{O}(\text{stability}\cdot\text{MI})\leq\mathcal{O}(\text{stability})$ (note  that we don't make such claim in our paper and we only mention the combination of the supersample/CMI technique and stability can solve Example 1), so we agree with AC this bound is still dimension dependent.
> > > > > > > >
> > > > > > > > >- because you consider a dimension dependent sample size (one could argue then that every bound is dimension independent if the sample is linear in the dimension)
> > > > > > > >
> > > > > > > > Regarding this, we understand your point, and we only want to remark that to explain the learnability of a class of learning problems, we hope to see the bound vanishes with $n$. For a  generalization bound with the form of $\mathcal{O}(\frac{d^{a}}{n^{b}})$, if $d$ is  independent with $n$, namely not changing with $n$, this bound always vanishes with $n$ and dimension dependency is no longer an issue, namely such bound is always able to explain the learnability.
> > > > > > > >
> > > > > > > > Thus, to demonstrate the failure of $\mathcal{O}(\frac{d^{a}}{n^{b}})$,  building the explicit relationship between $n$ and $d$, e.g., let $d^a$ grow faster than $n^b$, is necessary. Then, this will implicitly let stability has the relationship with $d$, as shown in our previous response.
> > > > > > > >
> > > > > > > > There are two ways to think about the generalization bound, how it changes with $n$ and how it changes with $d$. We think AC is talking about the second one, regardless of $n$, in this sense, we agree every comments that AC made.  **While in order to discuss about the learnability,  $d$ has to depend on $n$.**
> > > > > > > >
> > > > > > > > We are happy to discuss this further if AC has more comments.

---

> > ### Comment · Reviewer_Ptk3 · 2023-08-17
> > **This example does show improvement w.r.t typic MI bounds**
> >
> > Thanks for constructing this example. However, I would like to remind that the conventional MI bound in this case has the following form:
> > $\|\mathcal{E}\_{\mu}(\mathcal{A})\| \leq LR \sqrt{\frac{I(W;S)}{n}} \leq \frac{LR}{\sqrt{n}} = \frac{L\|\|z\_0\|\|}{\sqrt{n}} = \frac{L/d}{\sqrt{n}} \in \mathcal{O}(\frac{1}{n})$.
> >
> > The proposed bound does not show improvements w.r.t MI bounds.

---

> > > ### Author Response · Authors · 2023-08-18
> > >
> > > Thank you for your comments.
> > >
> > > Indeed you are correct in pointing out that the classical MI bound in our Example 3 is in fact tight, and our new MI bound incorporating stability does not improve over the classical MI bound.
> > >
> > > We however wish to note that the construction of Example 3 was to show that our bound (in the form of $\text{stability}\times \text{MI quantity}$) is better than both the MI quantity and the stability parameter (when it is also vanishing). This example supports our rationale that bounding the CGF appropriate is important for obtaining tighter bounds. Specifically, using the loss boundedness in Example 3 in the classical MI bound matches the stability bound, whereas the boundedness condition in Example 1 produces a much worse bound than the stability bound.
> > >
> > > In light of your comments and to show the advantage of our bound over both the stability bound and the classical MI bound, we here provide another example.
> > >
> > >
> > > **Example 4.** Let $Z\sim\text{Bern}\left(\theta\right)$ be a Bernoulli random variable, where we let $\theta=\frac{1}{\sqrt{n}}$. Now, let the hypothesis space be
> > > $\mathcal{W}=[0,1]$. Let the loss function be $\ell(w,z)=(w-z)^2$. Assume we use some ERM algorithm to estimate the population mean of  $Z$, namely $\theta$.
> > >
> > > Let $S=\\{Z_i\\}\_{i=1}^n$ be the training dataset, then the ERM will return the hypothesis $W\_{\text{ERM}}=\frac{1}{n}\sum_{i=1}^n Z_i$.
> > >
> > > In this example, the upper bound of the loss function is $\sup_{w,z}(w-z)^2= 1$.
> > >
> > > The stability parameter is
> > > $$
> > > \begin{aligned}
> > > \beta_2=\sup_{s\simeq s^i,z}| (w-z)^2-(w^i-z)^2|
> > > =&\sup_{s\simeq s^i,z}|w^2-{w^i}^2+2(w^i-w)z|\\\\
> > > \leq& \sup_{s,s^i}4|w-w^i|\\\\
> > > =& \sup_{z_i,z_i'}\frac{4}{n}|z_i-z'_i|\\\\
> > > =&\frac{4}{n}.
> > > \end{aligned}
> > > $$
> > >
> > > The MI in this example is $I(W;S)=H(W)=H(\frac{1}{n}\sum_{i=1}^n Z_i)=H(\sum_{i=1}^n Z_i)\approx \frac{1}{2}\log(2\pi ne\theta(1-\theta))$, where we use the approximation of Binomial distribution's entropy.
> > >
> > > Hence, classical MI bound gives us: $1\cdot\sqrt{\frac{H(W)}{n}}\leq\mathcal{O}(\sqrt{\frac{\log{n}}{n}})$.
> > >
> > >
> > > While the MI bound in this paper: $\beta_2\cdot\sqrt{\frac{H(W)}{n}}\leq\mathcal{O}(\frac{\sqrt{\log{n}}}{{n\sqrt{n}}})$.
> > > This improves both the stability and original MI bounds.
> > >
> > > A direct analysis of the exact generalization error of this example will give us $\mathcal{E}_\mu(\mathcal{A})\leq\mathcal{O}(\frac{1}{n\sqrt{n}})$. The MI bound in this paper is slightly slower.
> > >
> > > We would like thank again these review comments, which have enabled us to fully reveal the advantage of this work. Specifically, at this point, we see that to truly demonstrate that our bounds are superior to both the classical MI bound and the stability bound, the learning problem needs to simultaneously satisfied the following conditions:
> > >
> > > 1. the stability parameter should decay faster than the loss upper bound,
> > > 2. the MI quantity should also decay with sample size.
> > >
> > > When Condition 1 is satisfied, our bound will improve over the classical MI bound. When Condition 2 is satisfied, our bound will be tighter than the stability bound. In practice, e.g., deep learning scenarios, we anticipate both conditions are likely to hold.

---

> > > > ### Comment · Area_Chair_7BFw · 2023-08-19
> > > > **Elaborate on the direct analysis**
> > > >
> > > > Can you please elaborate of the direct analysis here? The best known generalization error I am aware of is $O(1/n)$ but I might be missing something here. Can you show how you obtain $O(1/n\sqrt{n})$ here directly?

---

> > > > > ### Author Response · Authors · 2023-08-19
> > > > > **Thanks for your question.**
> > > > >
> > > > > This is due to our uncommon choice of $\theta=\frac{1}{\sqrt{n}}$, that is, the variance $\theta(1-\theta)=\frac{1}{\sqrt{n}}-\frac{1}{n}$ also decays with $n$. For a fixed $\theta$, the generalization error remains within $\mathcal{O}(1/n)$. To elaborate,
> > > > > $$
> > > > > \begin{aligned}
> > > > > \mathcal{E}\_\mu(\mathcal{A})=&\frac{1}{n}\sum\_{i=1}^n\mathbb{E}\_{W,Z',Z\_i}[(W-Z')^2-(W-Z_i)^2]\\\\
> > > > > =&\frac{2}{n}\sum\_{i=1}^n\mathbb{E}\_{W,Z\_i}[WZ_i]-\frac{2}{n}\sum\_{i=1}^n\mathbb{E}\_{W,Z'}[WZ']\\\\
> > > > > =&\frac{2}{n}\sum\_{i=1}^n\mathbb{E}\_{W,Z\_i}[(\frac{1}{n}\sum_{j=1}^nZ_j)\cdot Z_i]-2\mathbb{E}\_{W}[W]\mathbb{E}\_{Z'}[Z']\\\\
> > > > > =&2\frac{n-1}{n}\theta^2+\frac{2}{n}\cdot\left(\theta(1-\theta)+\theta^2\right)-2\theta^2\\\\
> > > > > =&2\frac{1}{n}\left(\frac{1}{\sqrt{n}}-\frac{1}{n}\right).
> > > > > \end{aligned}
> > > > > $$
> > > > >
> > > > > The reason we chose $\theta=\frac{1}{\sqrt{n}}$ is to reveal the distribution-independent property of uniform stability. For a fixed $\theta$, we remark that, the variance of the binomial random variable $\sum_{i=1}^nZ_i$, namely $n\theta(1-\theta)$, linearly grows with $n$. When $n\to\infty$, using the Gaussian approximation as done before, might not be valid for the infinite variance case. In this case, we use the result in [4], which shows $\frac{H(W)}{n}\geq\Omega(H(Z))$. Hence, if $\theta$ is fixed, then $H(Z)$ is fixed, we have $\frac{H(W)}{n}\geq\Omega(1)$. In this setting, stability parameter itself can achieve the optimal rate of $\mathcal{O}(\frac{1}{n})$. When $H(Z)$ decreases with $n$, e.g., $\theta=\frac{1}{\sqrt{n}}$, the MI part can vanish, and the stability parameter alone has a slower rate.
> > > > >
> > > > > [4] Cheraghchi, Mahdi. "Expressions for the entropy of binomial-type distributions." 2018 IEEE International Symposium on Information Theory (ISIT). IEEE, 2018.

---

### Decision · Program_Chairs · 2023-09-21

**Decision:**

Accept (poster)

**Comment:**

The paper introduced a novel generalization bound of the form $\textrm{stability} \cdot \textrm{information}$ which can potentially improve over both stability and information-theoretic generalization bounds.

- The motivation of the paper, to remedy the lower bound in  Haghifam et al. was not found to be convincing. In particular, it doesn't seem that the above bound can be dimension dependent if any of the bounds (stability or MI) are dimension dependent.

- Nevertheless the authors could demonstrate a proof of concept where the above bound can be superior to both the stability bound and the information bound. It is true that this can happen only under certain distributional assumptions (as opposed to pure algorithmic assumptions) but this is the best that can be done given L'22 lower bound.

- The AC as well as reviewer strongly recommend to rewrite the introduction and reframe their result in the right light. It seems the authors acknowledged this need.